# Logit Distance Bounds Representational Similarity

**Beatrix M. G. Nielsen** [* 1] **Emanuele Marconato** [* 2] **Luigi Gresele** [† 3] **Andrea Dittadi** [† 4] **Simon Buchholz** [† 5]

## Abstract

For a broad family of discriminative models that includes autoregressive language models, identifiability results imply that if two models induce the same conditional distributions, then their internal representations are equal up to an invertible linear transformation. We ask whether an analogous conclusion holds approximately when the distributions are close instead of equal. Building on the observation of Nielsen et al. (2025) that closeness in KL divergence need not imply high linear representational similarity, we study a distributional distance based on logit differences and show that closeness in this distance does yield linear similarity guarantees. Specifically, we define a representational dissimilarity measure based on the models' identifiability class and prove that it is bounded by the logit distance. We further show that, when model probabilities are bounded away from zero, KL divergence upper-bounds logit distance; yet the resulting bound fails to provide nontrivial control in practice. As a consequence, KL-based distillation can match a teacher's predictions while failing to preserve linear representational properties, such as linear-probe recoverability of human-interpretable concepts. In distillation experiments on synthetic and image datasets, logit-distance distillation yields students with higher linear representational similarity and better preservation of the teacher's linearly recoverable concepts.

## 1. Introduction

It is widely believed that the success of deep learning models depends on the data representation they learn (Bengio et al., 2013); yet it is unclear what properties representations of "good" models have in common (Bansal et al., 2021). Accordingly, prior work studies when models with comparable performance exhibit similar internal representations (Morcos et al., 2018; Kornblith et al., 2019; Klabunde et al., 2025) and how human-interpretable concepts are encoded within them (Bricken et al., 2023; Gurnee & Tegmark, 2024). Empirically, many such concepts can be predicted from internal representations by simple linear probes (Alain & Bengio, 2017; Kim et al., 2018), suggesting substantial linear structure in the representations of successful models (Mikolov et al., 2013; Park et al., 2024). However, this regularity is not universal (Engels et al., 2025; Li et al., 2026), and it remains unclear to what extent linear representational properties are consistently shared across models that perform comparably well on the same task.

We study this question for a broad family of discriminative models, including autoregressive next-token prediction. Prior identifiability results show that, under a suitable diversity assumption, if two such models induce the same conditional distribution, then their representations are equal up to an invertible linear transformation (Khemakhem et al., 2020b; Roeder et al., 2021; Lachapelle et al., 2023), and one can characterize which linear properties are shared across the equivalence class (Marconato et al., 2025). A key question is whether these conclusions hold approximately when two models induce distributions that are close but not equal (Buchholz & Schölkopf, 2024). Nielsen et al. (2025) show that the answer depends on how distributional closeness is measured: in particular, models can be arbitrarily close in KL divergence while their representations remain far from linearly related. On the other hand, robustness can be recovered under other suitable divergences for which distributional closeness does imply representational similarity.

In this work, we ask: when two models in this family induce similar conditional distributions, to what extent do their representations agree up to an invertible linear map? We introduce a distributional distance based on logit differences and prove quantitative representation-level guarantees: small logit distance (i) implies high linear representational

---
[*]Equal contribution [†]Shared last author [1]IT University of Copenhagen, Denmark [2]University of Trento, Italy [3]University of Copenhagen, Denmark [4]Helmholtz Munich & TUM [5]MPI IS, Tübingen, Germany. Correspondence to: Beatrix M. G. Nielsen <beat@itu.dk>, Emanuele Marconato <emanuele.marconato@unitn.it>.

*Proceedings of the 43rd International Conference on Machine Learning*, Seoul, South Korea. PMLR 306, 2026. Copyright 2026 by the author(s).

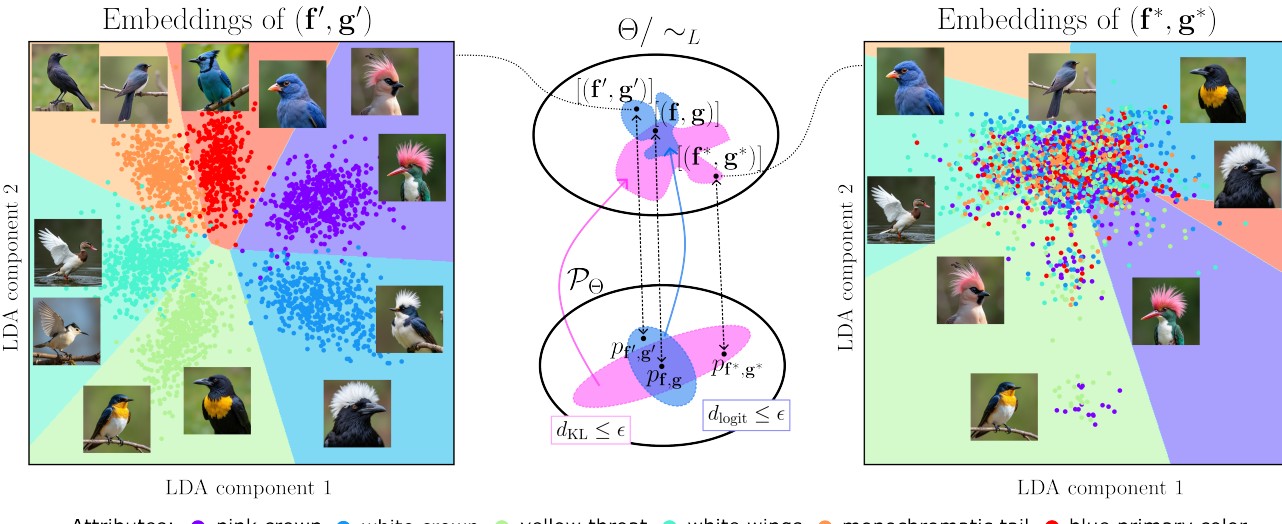

Figure 1. **In the center**: The intuition of bounding representational similarity using distributional distance. $\mathcal{P}_\Theta$ is the set of probability distributions parametrized by models in $\Theta$ (Eq. (1)) which satisfy Asm. 2.1. These distributions are one-to-one with identifiability classes $[(\mathbf{f}, \mathbf{g})]$ in the quotient space $\Theta/\sim_L$ (Khemakhem et al., 2020a). The colored areas in $\mathcal{P}_\Theta$ contain the distributions which are $\epsilon$-close to a reference $p_{\mathbf{f},\mathbf{g}}$, as measured by $d_{\text{logit}}$ (Def. 3.1, **blue area**) or by $d_{\text{KL}}$ (Eq. (10), **pink area**). Our Thm. 3.4 lower-bounds representational similarity (in terms of $m_{\text{CCA}}$, **blue arrow**) using the logit distance $d_{\text{logit}}$; similarly, Thm. 3.9 upper-bounds dissimilarity in terms of our $d_{\text{rep}}$ (Def. 3.7). In Thm. 3.3 (**pink arrow**) we prove that $d_{\text{KL}}$ yields weak bounds on $d_{\text{logit}}$. We illustrate this with representations of two student models distilled from a teacher on the SUB dataset (Bader et al., 2025), see §5.2 for details. **On the left**, a student model trained to minimize a variant of $d_{\text{logit}}$ (Eq. (21)) to the teacher distribution $p_{\mathbf{f},\mathbf{g}}$ preserves linearly encoded concepts (Thm. 4.3): for 6 attributes, we visualize their linear separability in the embeddings by projecting them to two dimensions through LDA (Bishop, 2006, Ch. 4.1). Distinct concept attributes can be well separated linearly in this 2d subspace. **On the right**, for a model trained to minimize $d_{\text{KL}}$ the LDA reduction shows that different concept attributes are not linearly separable, as reflected by the extremely low accuracy in Tab. 2.

similarity, yielding an explicit lower bound on mCCA between representation spaces (Raghu et al., 2017; Morcos et al., 2018) and (ii) upper-bounds a representational dissimilarity measure that we design to respect the model family's equivalence class. Finally, we clarify the (limited) extent to which KL control can recover such guarantees: if model probabilities are bounded away from zero, then the KL divergence upper-bounds the logit distance, but the resulting tight robustness bounds are insufficient for practical settings.

Our results have an immediate implication for knowledge distillation. Standard approaches match student predictions to a teacher's by minimizing a KL divergence between output distributions (Hinton et al., 2015); yet our theory implies that a student can be very close to its teacher in KL while still learning representations far from being linearly aligned. This motivates alternative objectives, and our theory suggests that minimizing logit differences—an approach considered in prior distillation work (Ba & Caruana, 2014; Menon et al., 2021; Kim et al., 2021)— should yield representations that are closer to being linearly equivalent. Empirically, we compare vanilla (KL-based) distillation to logit-distance distillation and find that the latter preserves the teacher's linear representational structure substantially better, both in terms of representational similarity measured by mCCA and our measure (on a synthetic dataset and CIFAR 100

(Krizhevsky et al., 2009)) and in terms of downstream linear properties recoverable by probing (on the SUB dataset (Bader et al., 2025)). Overall, our work clarifies when linear properties are approximately shared across models whose distributions are close, and suggests a practical distillation takeaway: if representational similarity matters, KL is the wrong loss; a logit-distance objective is a better one.

Our contributions can be summarized as follows:

- We analyze a logit-difference distance between conditional distributions and show it is a proper metric for the model family we consider (§3).
- We show KL divergence can upper-bound logit distance under strong assumptions (§3.1), but the resulting bounds do not yield meaningful representation guarantees in typical regimes.
- We propose a representation dissimilarity measure which is zero if and only if models are in the same linear equivalence class (§3.3).
- We prove that small logit distance implies high linear representational similarity, giving quantitative guarantees both for mCCA (§3.2), our dissimilarity measure (§3.4) and for linearly encoded concepts (§4).
- In distillation experiments, we show empirically that logit matching preserves the teacher's representational structure better than KL-based distillation (§5).

## 2. Preliminaries

In this section, we introduce the notation and model family, recall its linear identifiability under a diversity assumption, and formulate the question of approximate identifiability that motivates our analysis.

Let $\mathcal{X}$ denote the input space, either continuous or discrete. We assume a data distribution $p_{\mathbf{x}}$ supported on $\mathrm{supp}(p_{\mathbf{x}}) \subseteq \mathcal{X}$, and a finite label set $\mathcal{Y}$ of size $k$, such as tokens in a language model or class labels in a classification problem. We focus on a broad model family which encompasses autoregressive language models (Radford et al., 2019), methods for self-supervised pretraining for image classification (Oord et al., 2018), and several supervised classifiers (Khemakhem et al., 2020b; Ibrahim et al., 2024). We denote this model family by $\Theta$. Each element $(\mathbf{f}, \mathbf{g}) \in \Theta$ induces a conditional distribution $p_{\mathbf{f}, \mathbf{g}}(y \mid \mathbf{x})$, where $\mathbf{f} : \mathcal{X} \to \mathbb{R}^m$ is the *embedding* function and $\mathbf{g} : \mathcal{Y} \to \mathbb{R}^m$ the *unembedding* function. The conditional distribution takes the form

$$p_{\mathbf{f}, \mathbf{g}}(y \mid \mathbf{x}) = \frac{\exp(\mathbf{f}(\mathbf{x})^\top \mathbf{g}(y))}{\sum_{y' \in \mathcal{Y}} \exp(\mathbf{f}(\mathbf{x})^\top \mathbf{g}(y'))} \,. \quad (1)$$

These embeddings, $\mathbf{f}(\mathbf{x})$, and unembeddings, $\mathbf{g}(y)$, are the "last" representations before the final softmax, and they are the only representations we will consider in this article, since they are amenable to theoretical analysis based on existing identifiability results. We also indicate the logits generated by the model as

$$\mathbf{u}(\mathbf{x}) := \big(\mathbf{f}(\mathbf{x})^\top \mathbf{g}(y_1), \dots, \mathbf{f}(\mathbf{x})^\top \mathbf{g}(y_k)\big)^\top, \quad (2)$$

so the probability distribution can be equivalently described as a categorical distribution with probability vector $\mathrm{softmax}(\mathbf{u}(\mathbf{x}))$. Without loss of generality, we assume the unembeddings to be centered with zero mean,[1] which also results in the logits being centered:

$$\sum_{y \in \mathcal{Y}} \mathbf{g}(y) = \mathbf{0} \implies \sum_{i=1}^k u(\mathbf{x})_i = 0 \,. \quad (3)$$

When studying this model family, it is convenient to introduce the *shifted unembeddings* of a model $(\mathbf{f}, \mathbf{g}) \in \Theta$. For any pivot label $\tilde{y} \in \mathcal{Y}$, these are given by defining $\tilde{\mathbf{g}}(y) := \mathbf{g}(y) - \mathbf{g}(\tilde{y})$. We then introduce the matrix of shifted unembeddings constructed from $\tilde{\mathbf{g}}$ only considering a subset of $m$ labels $\mathcal{J} = \{y_1, \dots, y_m\} \subseteq \mathcal{Y} \setminus \{\tilde{y}\}$:

$$\tilde{\mathbf{L}}_{\mathcal{J}} := \big(\tilde{\mathbf{g}}(y_1) \quad \cdots \quad \tilde{\mathbf{g}}(y_m)\big) \in \mathbb{R}^{m \times m} \,. \quad (4)$$

**Identifiability of the model class.** Several works (Khemakhem et al., 2020b; Roeder et al., 2021; Lachapelle et al., 2023; Marconato et al., 2025; Reizinger et al., 2025; Nielsen

---

[1]Applying a fixed translation to all unembeddings does not change the probability distribution $p_{\mathbf{f}, \mathbf{g}}$ in Eq. (1).

et al., 2025) have investigated the symmetries belonging to the model class for which two different choices of embeddings and unembeddings, say $(\mathbf{f}, \mathbf{g}), (\mathbf{f}', \mathbf{g}') \in \Theta$, generate the same conditional probability distribution. In our setting, similar to (Nielsen et al., 2025), we study the case where labels are abundant and their number exceeds that of the representations, i.e., $k > m$. Below, we report the linear identifiability result from (Roeder et al., 2021). This requires an extra condition, known as diversity, which (following Marconato et al. (2025)) can be interpreted as a restriction to models that span the whole representation space.

**Assumption 2.1** (Diversity). *A model $(\mathbf{f}, \mathbf{g}) \in \Theta$ satisfies the diversity condition if both embeddings and the shifted unembeddings span the representation space. That is, if:*

$$\mathrm{span}\{\mathbf{f}(\mathbf{x}) : \mathbf{x} \in \mathrm{supp}(p_{\mathbf{x}})\} = \mathbb{R}^m \,, \quad (5)$$
$$\mathrm{span}\{\tilde{\mathbf{g}}(y) : y \in \mathcal{Y}\} = \mathbb{R}^m \,. \quad (6)$$

*In particular, there exists a choice of $\tilde{y} \in \mathcal{Y}$ and $\mathcal{J} \subseteq \mathcal{Y} \setminus \{\tilde{y}\}$, for which $\tilde{\mathbf{L}}_{\mathcal{J}}$ (Eq. (4)) is invertible.*

This condition gives the following result (proof in App. C.1).

**Theorem 2.2** (Linear Identifiability). *Let $(\mathbf{f}, \mathbf{g}), (\mathbf{f}', \mathbf{g}') \in \Theta$ and let $(\mathbf{f}, \mathbf{g})$ satisfy the diversity condition (Asm. 2.1). Let $\tilde{y} \in \mathcal{Y}$ and $\mathcal{J} \subseteq \mathcal{Y} \setminus \{\tilde{y}\}$ be a choice of pivot point and $m$ labels such that $\tilde{\mathbf{L}}_{\mathcal{J}}$ is invertible. Let $\tilde{\mathbf{A}}_{\mathcal{J}} := \tilde{\mathbf{L}}_{\mathcal{J}}^{-\top} \tilde{\mathbf{L}}_{\mathcal{J}}'^{\top} \in \mathbb{R}^{m \times m}$. Let $\sim_L$ be the equivalence relation in $\Theta$, such that $(\mathbf{f}, \mathbf{g}) \sim_L (\mathbf{f}', \mathbf{g}')$ if and only if there exists a matrix $\mathbf{A}$ such that*

$$\mathbf{f}(\mathbf{x}) = \mathbf{A}\mathbf{f}'(\mathbf{x}), \quad \tilde{\mathbf{g}}(y) = \mathbf{A}^{-\top}\tilde{\mathbf{g}}'(y) \,. \quad (7)$$

*Then we have that*

$$p_{\mathbf{f}, \mathbf{g}}(y \mid \mathbf{x}) = p_{\mathbf{f}', \mathbf{g}'}(y \mid \mathbf{x}), \ \forall (\mathbf{x}, y) \in \mathcal{X} \times \mathcal{Y}$$
$$\iff (\mathbf{f}, \mathbf{g}) \sim_L (\mathbf{f}', \mathbf{g}') \,. \quad (8)$$

*In particular, we can set $\mathbf{A} = \tilde{\mathbf{A}}_{\mathcal{J}}$.*

This result shows that when the distributions generated by two models are equal and one satisfies the diversity condition, their representations are equal up to an invertible linear transformation given by $\tilde{\mathbf{A}}_{\mathcal{J}} \in \mathbb{R}^{m \times m}$. We show (Proposition C.1) that the form of $\tilde{\mathbf{A}}_{\mathcal{J}}$ does not depend on the choice of $\tilde{y}$ and $\mathcal{J} \subseteq \mathcal{Y} \setminus \{\tilde{y}\}$ as long as the matrix $\tilde{\mathbf{L}}_{\mathcal{J}}$ is invertible.

Note that to make this comparison of models from our model family, we need them to have the same number of labels, $k$, and have the same representational dimension, $m$. However, the embedding, $\mathbf{f}, \mathbf{f}'$, and unembedding, $\mathbf{g}, \mathbf{g}'$, functions can have any number of hidden layers and do not need to have the same architecture. Therefore, there is still a large amount of freedom for which models we can compare.

**Closeness in distribution and representational similarity.** The result in Thm. 2.2 assumes *exact* equality of conditional

distributions. In practice, however, independently trained models will almost never satisfy this condition exactly, even when trained on the same data. This raises the question of whether the identifiability conclusion is *robust*: namely, whether models whose distributions are merely close, rather than equal, must also have similar internal representations up to the model class's intrinsic symmetry (an invertible linear change of basis).

This question was first systematically investigated by Nielsen et al. (2025). Addressing it requires specifying (i) a notion of distance between conditional distributions and (ii) a notion of dissimilarity between representations. For the latter, we require the dissimilarity to vanish for $\sim_L$-linearly equivalent models. This is because, for any $p_{\mathbf{f},\mathbf{g}}(y \mid \mathbf{x})$, different reparametrizations of $(\mathbf{f}, \mathbf{g})$ exists that do not change the probability distribution (by Thm. 2.2). With this requirement in place, our goal is to identify distributional distances for which small discrepancies in conditional distributions provably imply small representational dissimilarity.

# 3. The Logit Distance and Connection to Representational Similarity

We now define a metric between distributions from our model family.

**Definition 3.1.** *Let $p_{\mathbf{f},\mathbf{g}}, p'_{\mathbf{f}',\mathbf{g}'}$ be two sets of conditional distributions arising from two models from our model class. We denote by $d_{\mathrm{logit}}$ the following logit distance*

$$d^2_{\mathrm{logit}}(p_{\mathbf{f},\mathbf{g}}, p'_{\mathbf{f}',\mathbf{g}'}) = \mathbb{E}_{\mathbf{x} \sim p_{\mathbf{x}}} \|\mathbf{u}(\mathbf{x}) - \mathbf{u}'(\mathbf{x})\|_2^2 \ . \quad (9)$$

Notice that we are here defining the squared distance, $d^2_{\mathrm{logit}}$, so $d_{\mathrm{logit}}$ coincides with the square root of the mean squared error between models' logits. This can be written in terms of log-probabilities (see Appendix D.1 for details), and the expression $\|\mathbf{u}(\mathbf{x}) - \mathbf{u}'(\mathbf{x})\|_2$ corresponds to the Aitchison distance (see App. D.2) of the conditional distributions $p_{\mathbf{f},\mathbf{g}}(\cdot|\mathbf{x})$ and $p_{\mathbf{f}',\mathbf{g}'}(\cdot|\mathbf{x})$, which is used in compositional data analysis (Aitchison et al., 2000; Pawlowsky-Glahn et al., 2007). We show in App. D.3 that Def. 3.1 is a metric (potentially with infinite value) between distributions from models in our model class. A sufficient condition for $d_{\mathrm{logit}}(p, p') < \infty$, is that the probabilities are lower bounded by some $\tau > 0$ for all $y \in \mathcal{Y}$ and $\mathbf{x} \in \mathcal{X}$.

## 3.1. KL Divergence Bounds Logit Distance

We consider the relationship between the *logit distance* and the Kullback–Leibler (KL) divergence which both measure the closeness between two distributions. Following Nielsen et al. (2025), given $(\mathbf{f}, \mathbf{g}), (\mathbf{f}', \mathbf{g}') \in \Theta$, we consider

$$d_{\mathrm{KL}}(p_{\mathbf{f},\mathbf{g}}, p_{\mathbf{f},\mathbf{g}'}) := \mathbb{E}_{\mathbf{x} \sim p_{\mathbf{x}}}[\mathrm{KL}(p_{\mathbf{f},\mathbf{g}}(\cdot \mid \mathbf{x}) \| p_{\mathbf{f}',\mathbf{g}'}(\cdot \mid \mathbf{x}))], \quad (10)$$

which is related to the cross-entropy loss

$$\mathcal{L}_p(\mathbf{f}', \mathbf{g}') = \mathbb{E}_{\mathbf{x} \sim p_{\mathbf{x}}, y \sim p(y|\mathbf{x})}[-\log p_{\mathbf{f}',\mathbf{g}'}(y|\mathbf{x})] \quad (11)$$

for a well-specified model through

$$d_{\mathrm{KL}}(p_{\mathbf{f},\mathbf{g}}, p_{\mathbf{f},\mathbf{g}'}) = \mathcal{L}_{p_{\mathbf{f},\mathbf{g}}}(\mathbf{f}', \mathbf{g}') - \mathcal{L}_{p_{\mathbf{f},\mathbf{g}}}(\mathbf{f}, \mathbf{g}) \ . \quad (12)$$

This means that learning a model $(\mathbf{f}', \mathbf{g}') \in \Theta$ on data sampled from a distribution $p_{\mathbf{f},\mathbf{g}}$ in the model class, will be equivalent in expectation to minimizing $d_{\mathrm{KL}}(p_{\mathbf{f},\mathbf{g}}, p_{\mathbf{f}',\mathbf{g}'})$. However, Nielsen et al. (2025, Theorem 3.1) show that the equivalence relation $\sim_L$ is not robust to small changes in the probability distribution as measured by $d_{\mathrm{KL}}$. Specifically, they construct pairs $(\mathbf{f}, \mathbf{g})$ and $(\mathbf{f}', \mathbf{g}')$ with arbitrarily small KL divergence, yet whose representations cannot be made close by any invertible linear map ($m_{\mathrm{CCA}}$ is low). This can happen because, without additional regularity assumptions on $\Theta$, models may assign extremely small probabilities to some events. Changes concentrated on such near-zero-probability events can correspond to large (potentially non-linear) changes in representation geometry while incurring only a negligible increase in $d_{\mathrm{KL}}$. To rule out these corner cases, we restrict attention to models $(\mathbf{f}, \mathbf{g}) \in \Theta$ whose probability assignments are uniformly lower bounded by some constant $\tau > 0$.

**Assumption 3.2** ($\tau$-lower bounded). *For $\tau > 0$, we say that a model $(\mathbf{f}, \mathbf{g}) \in \Theta$ is $\tau$-lower bounded if, for $p_{\mathbf{x}}$-almost every $\mathbf{x} \in \mathcal{X}$, we have*

$$\min_{y \in \mathcal{Y}} p_{\mathbf{f},\mathbf{g}}(y \mid \mathbf{x}) \geq \tau \ . \quad (13)$$

We also show in App. E.1 that this is equivalent to having bounded embeddings and unembeddings. We can now prove the next result (full proof in App. E.2).

**Theorem 3.3.** *For two models $(\mathbf{f}, \mathbf{g}), (\mathbf{f}', \mathbf{g}') \in \Theta$, we have*

$$d^2_{\mathrm{logit}}(p_{\mathbf{f},\mathbf{g}}, p'_{\mathbf{f}',\mathbf{g}'}) \geq 2d_{\mathrm{KL}}(p_{\mathbf{f},\mathbf{g}}, p_{\mathbf{f}',\mathbf{g}'}) \ . \quad (14)$$

*If there exists $0 < \tau < 1/3$ such that both $(\mathbf{f}, \mathbf{g})$ and $(\mathbf{f}', \mathbf{g}')$ are $\tau$-lower bounded (Asm. 3.2), then*

$$d^2_{\mathrm{logit}}(p_{\mathbf{f},\mathbf{g}}, p'_{\mathbf{f}',\mathbf{g}'}) \leq \frac{4 \log(\tau)^2}{\tau} d_{\mathrm{KL}}(p_{\mathbf{f},\mathbf{g}}, p_{\mathbf{f}',\mathbf{g}'}) \ . \quad (15)$$

In contrast to the general setting discussed above—where arbitrarily small $d_{\mathrm{KL}}$ may still correspond to highly dissimilar representations—Thm. 3.3 shows that, under the additional $\tau$-boundedness assumption, the KL divergence *does* bound the logit distance. As we show below, this in turn implies that $d_{\mathrm{KL}}$ also bounds representation dissimilarity. However, the upper bound in (15) is only quantitatively meaningful when $d_{\mathrm{KL}} = O(\tau)$, since the bound scales as $\tau^{-1}$ up to logarithmic factors. As $\tau \leq 1/k$ in a $k$-class setting and

is typically much smaller in practice, this condition can be restrictive. We remark that a similar but weaker bound holds when only one of the models (e.g., a constrained student) is $\tau$-bounded (see Lemma E.4). Moreover, the dependence on $\tau$ is tight up to logarithmic terms and cannot be improved when the KL-divergence is replaced by the Jensen-Shannon divergence (see Remark E.3).

## 3.2. Logit Distance Bounds Mean CCA

We next show that the representations are approximately linearly related when the distance between the logits of two models is small. Canonical Correlation Analysis (CCA) (Hotelling, 1936) is a common way of measuring whether sets of vectors are linearly related (see App. B.1). We denote by $m_{\mathrm{CCA}}$ the mean canonical correlation which is 1 when two vector-valued variables are almost surely linear transformations of each other. In our setting, the pair of variables will be either a pair of embeddings or unembeddings from two different models.

**Theorem 3.4.** *Let $m$ be the dimension of the representations. Let $\mu_m$ be the $m$-th eigenvalue of the matrix $\boldsymbol{\Sigma}_{\mathbf{u},\mathbf{u}} := \mathbb{E}_{\mathbf{x} \sim p_{\mathbf{x}}} \left[ \mathbf{u}(\mathbf{x}) \mathbf{u}(\mathbf{x})^\top \right] - \mathbb{E}_{\mathbf{x} \sim p_{\mathbf{x}}} [\mathbf{u}(\mathbf{x})] \mathbb{E}_{\mathbf{x} \sim p_{\mathbf{x}}} [\mathbf{u}(\mathbf{x})^\top]$. Assume $\mu_m > 0$. Then we have*

$$m_{\mathrm{CCA}}(\mathbf{f}(\mathbf{x}), \mathbf{f}'(\mathbf{x})) \geq 1 - \frac{d_{\mathrm{logit}}^2(p_{\mathbf{f},\mathbf{g}}, p'_{\mathbf{f}',\mathbf{g}'})}{m \mu_m} \quad (16)$$

*and the same bound holds for $m_{\mathrm{CCA}}(\mathbf{g}(y), \mathbf{g}'(y))$.*

Full proof in App. F.4. We emphasize that this result is invariant to linear transformations $(\mathbf{f}, \mathbf{g}) \mapsto (\mathbf{A}\mathbf{f}, \mathbf{A}^{-\top}\mathbf{g})$. The theorem shows that, with a small difference in distributions in terms of $d_{\mathrm{logit}}$ of two models, we get similar representations in terms of $m_{\mathrm{CCA}}$. In particular, if the distributions of $p_{\mathbf{f},\mathbf{g}}$ and $p_{\mathbf{f}',\mathbf{g}'}$ agree, we conclude that $m_{\mathrm{CCA}}(\mathbf{f}(\mathbf{x}), \mathbf{f}'(\mathbf{x})) = 1$, which means that there exists a matrix $\mathbf{B}$ such that $\mathbf{f}(\mathbf{x}) = \mathbf{B}\mathbf{f}'(\mathbf{x})$ for all $\mathbf{x}$ and similarly $\mathbf{g}(y) = \tilde{\mathbf{B}}\mathbf{g}(y)$ for another matrix $\tilde{\mathbf{B}}$. Therefore this result generalizes the linear identifiability of $\mathbf{f}$ and $\mathbf{g}$ in Thm. 2.2 to models that are similar in terms of $d_{\mathrm{logit}}$ or $d_{\mathrm{KL}}$ (via Theorem 3.3). However, using $m_{\mathrm{CCA}}$ to measure representational similarity has some limitations. Having $m_{\mathrm{CCA}}(\mathbf{f}(\mathbf{x}), \mathbf{f}'(\mathbf{x})) = 1$ and small KL divergence is not sufficient to conclude that the joint representations $(\mathbf{f}, \mathbf{g})$ and $(\mathbf{f}', \mathbf{g}')$ are similar (see App. G for an example), and having both $m_{\mathrm{CCA}}(\mathbf{f}(\mathbf{x}), \mathbf{f}'(\mathbf{x})) = 1$ and $m_{\mathrm{CCA}}(\mathbf{g}(y), \mathbf{g}'(y)) = 1$ does not ensure that the distributions of the models are equal (see App. H for an example). Also, while we can conclude that $\mathbf{B} = \tilde{\mathbf{B}}^{-\top}$ holds for $d_{\mathrm{logit}} = 0$ and thus recover the identifiability result, no rates for the error can be obtained when $d_{\mathrm{logit}}$ is positive (see Appendix F.3). This motivates the study of further dissimilarity measures which naturally incorporate the relation of embeddings and unembeddings and therefore better match the linear identifiability result.

## 3.3. Linear Identifiability Dissimilarity

In this section we define a dissimilarity measure, $d_{\mathrm{rep}}$, between representations inspired by Thm. 2.2, in the sense that it is zero if and only if the representations are linearly equivalent. We first notice that linear equivalence in Thm. 2.2, mediated by $\tilde{\mathbf{A}}_{\mathcal{J}}$, necessitates a choice of pivot point and $m$ other labels such that $\tilde{\mathbf{L}}_{\mathcal{J}}$ is invertible[2]. Since we do not want the invertibility of $\tilde{\mathbf{L}}_{\mathcal{J}}$ to depend on the choice, we introduce the following assumption:

**Assumption 3.5.** *A model $(\mathbf{f}, \mathbf{g}) \in \Theta$ is in general position if, for every $\tilde{y} \in \mathcal{Y}$ and $\mathcal{J} \subseteq \mathcal{Y} \setminus \{\tilde{y}\}$, the matrix $\tilde{\mathbf{L}}_{\mathcal{J}} \in \mathbb{R}^{m \times m}$ (Eq. (4)) is invertible, and the embeddings span the representation space:* $\mathrm{span}\{\mathbf{f}(\mathbf{x}) : \mathbf{x} \in \mathrm{supp}(p_{\mathbf{x}})\} = \mathbb{R}^m$.

This assumption naturally requires that for any choice of pivot point, $\tilde{y}$, the shifted unembeddings $\tilde{\mathbf{g}}(y)$ are in *general position* (Cover, 1965).[3] It also ensures that the *diversity condition* (Asm. 2.1) is satisfied, since the assumption is the same for the embeddings and slightly stronger for the shifted unembeddings. For models satisfying Asm. 3.5, Thm. 2.2 gives us the following corollary:

**Corollary 3.6.** *For two models $(\mathbf{f}, \mathbf{g}), (\mathbf{f}', \mathbf{g}') \in \Theta$ that satisfy Asm. 3.5. If $p_{\mathbf{f},\mathbf{g}}(y \mid \mathbf{x}) = p_{\mathbf{f}',\mathbf{g}'}(y \mid \mathbf{x})$, $\forall (\mathbf{x}, y) \in \mathcal{X} \times \mathcal{Y}$ then* **for every choice of pivot point and $m$ labels,** *we have*

$$\mathbf{f}(\mathbf{x}) = \tilde{\mathbf{A}}_{\mathcal{J}} \mathbf{f}'(\mathbf{x}) \quad and \quad \tilde{\mathbf{g}}(y) = \tilde{\mathbf{A}}_{\mathcal{J}}^{-\top} \tilde{\mathbf{g}}'(y). \quad (17)$$

*Equivalently, if there exists a choice of pivot and labels such that the linear relation does not hold, then $\exists \mathbf{x}, y$ s.t. $p_{\mathbf{f},\mathbf{g}}(y \mid \mathbf{x}) \neq p_{\mathbf{f}',\mathbf{g}'}(y \mid \mathbf{x})$.*

Therefore, if we want to make a dissimilarity measure which measures how far the representations are from being in the same equivalence class —independent of the choice of pivot points and labels— we have to consider the error for all possible choices of labels which can be used to construct the $\tilde{\mathbf{A}}_{\mathcal{J}}$ matrix. We capture this in the definition below.

**Definition 3.7.** *Let $J = \binom{k-1}{m}$. For any two choices of models $(\mathbf{f}, \mathbf{g}), (\mathbf{f}', \mathbf{g}') \in \Theta$ satisfying Asm. 3.5, the squared linear identifiability dissimilarity is given by*

$$d_{\mathrm{rep}}^2((\mathbf{f}, \mathbf{g}), (\mathbf{f}', \mathbf{g}')) := \quad (18)$$
$$\frac{1}{kJ} \sum_{\tilde{y} \in \mathcal{Y}} \sum_{\mathcal{J} \subseteq \mathcal{Y} \setminus \{\tilde{y}\}} \mathbb{E}_{\mathbf{x} \sim p_{\mathbf{x}}} \left\| \mathbf{f}(\mathbf{x}) - \tilde{\mathbf{A}}_{\mathcal{J}} \mathbf{f}'(\mathbf{x}) \right\|_2^2.$$

The *linear identifiability dissimilarity* in Def. 3.7 averages the squared distance between $\mathbf{f}(\mathbf{x})$ and $\tilde{\mathbf{A}}_{\mathcal{J}} \mathbf{f}'(\mathbf{x})$ over every

---

[2]Although in the case where distributions are equal, the $\tilde{\mathbf{A}}_{\mathcal{J}}$ will be the same for any invertible choice (see Proposition C.1).

[3]We say $n \geq m$ vectors are in *general position* in $\mathbb{R}^m$ if, for any choice of $m$ or fewer vectors, these are linearly independent.

choice of pivot and labels and every input. Note that because of Asm. 3.5 all choices lead to invertible matrices. We see that by Corollary 3.6, if the distributions of $(\mathbf{f}, \mathbf{g})$ and $(\mathbf{f}', \mathbf{g}')$ are equal, then $d_{\text{rep}}$ is zero. Next, we will show that using the logit distance between distributions, this can be bounded. We will also show that if $d_{\text{rep}}((\mathbf{f}, \mathbf{g}), (\mathbf{f}', \mathbf{g}')) = 0$ then distributions are also equal, so equal representations in this sense implies equal distributions. Note that this differs from for example $m_{\text{CCA}}$, since $m_{\text{CCA}}$ can be maximal between both embeddings and unembeddings without the distributions being equal (see App. H). In other words, the *linear identifiability dissimilarity* gives us a tighter connection between the functional, $d_{\text{logit}}$, and representational, $d_{\text{rep}}$, similarity measures (see Klabunde et al. (2025) for details on this distinction).

### 3.4. Logit Distance Bounds Representational Distance

In this section, we show that, for our model family, the logit distance is zero if and only if the linear identifiability dissimilarity vanishes and that the former bounds the latter. As a corollary, we show that this bound extends also to KL. Full proofs for this section are in App. I.

We note that, for two models $(\mathbf{f}, \mathbf{g}), (\mathbf{f}', \mathbf{g}') \in \Theta$ that satisfy Asm. 3.5, the squared logit distance can be written as a sum of terms of the form $\left\| \mathbf{B}^{-\top} \left( \mathbf{f}(\mathbf{x}) - \tilde{\mathbf{A}}_{\mathcal{J}} \mathbf{f}'(\mathbf{x}) \right) \right\|_2^2$ for a matrix $\mathbf{B}$ related to the SVD decomposition of $\tilde{\mathbf{L}}_{\mathcal{J}}$ (see Thm. I.1 for the precise statement). This shows that the logit difference is naturally connected to the linear identifiability dissimilarity and gives us the following corollary:

**Corollary 3.8.** *For $(\mathbf{f}, \mathbf{g}), (\mathbf{f}', \mathbf{g}') \in \Theta$ satisfying Asm. 3.5, we have*

$$d_{\text{rep}}((\mathbf{f}, \mathbf{g}), (\mathbf{f}', \mathbf{g}')) = 0 \iff d_{\text{logit}}(p_{\mathbf{f}, \mathbf{g}}, p_{\mathbf{f}', \mathbf{g}'}) = 0.$$

Moreover, we can upper (and lower, see App. I.2) bound $d_{\text{rep}}$ using $d_{\text{logit}}$ and the singular values of $\tilde{\mathbf{L}}_{\mathcal{J}}$.

**Theorem 3.9.** *Let $p_{\mathbf{f}, \mathbf{g}}, p_{\mathbf{f}', \mathbf{g}'}$ be as before, and $m$ the dimension of the representations. Let $C = \sqrt{\frac{2m}{k-1}}$. Let $\sigma_{\min}$ be the smallest singular value[4] of all $\tilde{\mathbf{L}}_{\mathcal{J}}$. Then $d_{\text{logit}}$, bounds the linear identifiability dissimilarity $d_{\text{rep}}$ as follows:*

$$d_{\text{rep}}((\mathbf{f}, \mathbf{g}), (\mathbf{f}', \mathbf{g}')) \leq C \, \frac{d_{\text{logit}}(p_{\mathbf{f}, \mathbf{g}}, p_{\mathbf{f}', \mathbf{g}'})}{\sigma_{\min}} \,. \quad (19)$$

Interestingly, this result highlights that the connection between distributional distance and representational dissimilarity is tightly connected to both embeddings and unembeddings. In fact, the bound depends on the minimum singular value among the unembeddings matrices $\tilde{\mathbf{L}}_{\mathcal{J}}$, i.e., $\sigma_{\min}$, accounting for the direction in embedding space of minimum

---

[4]Note that since all the $\tilde{\mathbf{L}}_{\mathcal{J}}$ are invertible, $\sigma_{\min} > 0$.

variation of the shifted unembeddings. In particular, small values of $d_{\text{logit}}$ will ensure that the embeddings of the two models are close to being a linear invertible transformation of each other, where the transformation is mediated by the unembeddings. This also links to a bound on $d_{\text{rep}}$ through $d_{\text{KL}}$ when models have lower bounded probabilities:

**Corollary 3.10.** *Let $(\mathbf{f}, \mathbf{g}), (\mathbf{f}', \mathbf{g}') \in \Theta$ be two models from our model class (1) which satisfy Asm. 3.2 and Asm. 3.5 . Let $C = \sqrt{\frac{2m}{k-1}}$ and $C_{KL} = \frac{2C|\log(\tau)|}{\sqrt{\tau}}$. Then*

$$d_{\text{rep}}((\mathbf{f}, \mathbf{g}), (\mathbf{f}', \mathbf{g}')) \leq C_{KL} \frac{\sqrt{d_{\text{KL}}(p_{\mathbf{f}, \mathbf{g}}, p_{\mathbf{f}', \mathbf{g}'})}}{\sigma_{\min}} \,. \quad (20)$$

## 4. Applications: Distillation and Interpretability

**Knowledge distillation** (KD) aims to train a compact student model to match the predictive distribution of a larger teacher. In its original formulation (Hinton et al., 2015), KD combines a supervised cross-entropy loss, $\mathcal{L}_{p_{\mathcal{D}}}$, on the data distribution $p_{\mathcal{D}}(y \mid \mathbf{x})$ with a KL divergence term that aligns the student distribution with that of a fixed teacher $(\mathbf{f}, \mathbf{g})$ with distribution $p_{\mathbf{f}, \mathbf{g}}$. The student $(\mathbf{f}', \mathbf{g}')$ with distribution $p_{\mathbf{f}' \mathbf{g}'}$ is obtained by solving

$$(\mathbf{f}', \mathbf{g}') \in \operatorname*{argmin}_{(\mathbf{f}^*, \mathbf{g}^*) \in \Theta} \left( \mathcal{L}_{p_{\mathcal{D}}}(\mathbf{f}^*, \mathbf{g}^*) + d_{\text{KL}}(p_{\mathbf{f}, \mathbf{g}}, p_{\mathbf{f}^*, \mathbf{g}^*}) \right)$$

typically using only a small subset of training samples. A large body of work explores refinements of this objective to better match the teacher distribution (Mansourian et al., 2025), including alternative formulations of the KL term (Zhao et al., 2022) or by considering an equivalent loss to $d_{\text{logit}}$ (Navaneet et al., 2021; Kim et al., 2021). When available, the teacher's intermediate activations can further guide training (Romero et al., 2015; Huang et al., 2023), with regression on embeddings or unembeddings encouraging closer representational alignment.

Our analysis highlights that, when only teacher probabilities are available, the original $d_{\text{KL}}$ minimization does not significantly bound representational similarity (as measured with $d_{\text{rep}}$) between teacher and student. When only teacher probabilities or logits are available, a better learning target is to minimize an equivalent loss to $d_{\text{logit}}$. Apart from using $d_{\text{logit}}$ as a loss directly, we consider training with a $L_1$ variant of the logit distance between teacher logits $\mathbf{u}$ and student $\mathbf{u}'$, that is

$$\mathcal{L}^1_{\text{logit}}(p_{\mathbf{f}, \mathbf{g}}, p_{\mathbf{f}', \mathbf{g}'}) \coloneqq \mathbb{E}_{\mathbf{x} \sim p_{\mathbf{x}}} \left\| \mathbf{u}(\mathbf{x}) - \mathbf{u}'(\mathbf{x}) \right\|_1 \,, \quad (21)$$

Using $\mathcal{L}^1_{\text{logit}}$ improves training stability (Qi et al., 2020) and, importantly, still leads to decreasing $d_{\text{logit}}(p_{\mathbf{f}, \mathbf{g}}, p_{\mathbf{f}', \mathbf{g}'})$:

**Proposition 4.1.** *For two models $(\mathbf{f}, \mathbf{g}), (\mathbf{f}', \mathbf{g}') \in \Theta$ which both satisfy Assumption 3.2 for some $\tau > 0$, then*

$$d^2_{\text{logit}}(p_{\mathbf{f}, \mathbf{g}}, p_{\mathbf{f}', \mathbf{g}'}) \leq 2|\log(\tau)|\mathcal{L}^1_{\text{logit}}(p_{\mathbf{f}, \mathbf{g}}, p_{\mathbf{f}', \mathbf{g}'}) \,. \quad (22)$$

In particular, with this result we have that minimizing $\mathcal{L}^1_{\text{logit}}$ will naturally reduce $d_{\text{logit}}$ between the teacher and student distributions, independently of the representation dimensionality of the two. Full proof in App. K.

**Interpretable properties.** Prior works have investigated whether model embeddings linearly encode latent, interpretable concepts (Mikolov et al., 2013; Kim et al., 2018; Engels et al., 2025). For example, Marks & Tegmark (2024) showed that the truth value of a statement $\mathbf{x}$ can be approximated by a linear function of its language model embedding $\mathbf{f}(\mathbf{x})$. Similar forms of linear separability have also been studied in pretrained neural networks for visual concepts in image classification (Alain & Bengio, 2017).

Formally, we define a *categorical concept* as a function $\mathbf{h} : \mathcal{X} \to \Delta^{C-1}$, where $\Delta^{C-1}$ is the simplex over $C \geq 2$ values, giving a distribution over concept values

$$p_{\mathbf{h}}(c \mid \mathbf{x}) := \mathbf{h}(\mathbf{x})_c, \ \forall c \in \{1, \ldots, C\}. \qquad (23)$$

Inspired by previous works (Alain & Bengio, 2017; Rajendran et al., 2024), we propose a definition that captures linear separability of concepts from model embeddings:

**Definition 4.2** (Linearly encoded concept $\mathbf{h}$ in $\mathbf{f}$). *For a concept* $\mathbf{h} : \mathcal{X} \to \Delta^{C-1}$, *with* $C \geq 2$, *giving a distribution* $p_{\mathbf{h}}$ *as in Eq.* (23), *we say that that* $\mathbf{h}$ *is* linearly encoded *in* $\mathbf{f}$ *if there exist* $C$ *linear weights* $\mathbf{w}_c \in \mathbb{R}^m$ *and biases* $b_c \in \mathbb{R}$ *such that, for all* $c \in \{1, \ldots, C\}$,

$$\frac{\exp\left(\mathbf{w}_c^\top \mathbf{f}(\mathbf{x}) + b_c\right)}{\sum_{j=1}^C \exp\left(\mathbf{w}_j^\top \mathbf{f}(\mathbf{x}) + b_j\right)} = p_{\mathbf{h}}(c \mid \mathbf{x}). \qquad (24)$$

*Then, we indicate with* $\mathbf{W} \in \mathbb{R}^{m \times C}$ *the matrix of weights* $\mathbf{w}_c$ *and with* $\mathbf{b} \in \mathbb{R}^C$ *the vector of biases* $b_c$.

In words, Def. 4.2 states that the probability assignments over concept values defined by $p_{\mathbf{h}}$ can be recovered through a linear function of the model embeddings. This also implies that the embedding space is partitioned linearly into convex regions, each corresponding to the most likely concept label (Snell et al., 2017), as illustrated in Fig. 1 (left). Crucially, we show that $d_{\text{logit}}$ bounds the degree to which concepts are shared between models that are close:

**Theorem 4.3.** *For two models* $(\mathbf{f}, \mathbf{g}), (\mathbf{f}', \mathbf{g}') \in \Theta$ *and a concept* $\mathbf{h} : \mathcal{X} \to \Delta^{C-1}$ *with* $C \geq 2$, *if* $\mathbf{h}$ *is linearly encoded in* $\mathbf{f}$, *then*

$$\min_{\mathbf{W}', \mathbf{b}'} d_{\text{KL}}\left(p_{\mathbf{h}}, \text{softmax}\left(\mathbf{W}'^\top \mathbf{f}'(\cdot) + \mathbf{b}'\right)\right)$$
$$\leq \frac{1}{2}\|\mathbf{A}\|_{op}^2 d_{\text{logit}}^2(p_{\mathbf{f},\mathbf{g}}, p_{\mathbf{f}',\mathbf{g}'}), \quad (25)$$

*where* $\|\cdot\|_{\text{op}}$ *denotes the operator norm,* $\mathbf{A}$ *is determined by* $\mathbf{W} = \mathbf{L}\mathbf{A}$, $\mathbf{L}$ *is the unembedding matrix, and* $\mathbf{W}$, $\mathbf{b}$ *are obtained from Def. 4.2.*

This result (proof in App. J) also shows that equality between model distributions preserves linearly encoded concepts between models that are linearly equivalent, mirroring previous results (Park et al., 2024; Marconato et al., 2025).

**Implications for KD and interpretability**. The analysis we present here highlights two issues that arise when aiming to obtain student representations similar to those of a teacher model. First, standard KD strategies that leverage (symmetric variants of) KL divergence may fail to guarantee linear similarity to teacher representations. While not affecting the accuracy of the students, this can impact other aspects of the model distribution, e.g., rank orderings (Grivas et al., 2024) or uncertainty attributions (Wang, 2023). Second, linear properties of the teacher representations may be less faithfully preserved in KL-distilled students. This is undesirable when students are expected to preserve interpretable properties of the teacher such as linear steerability or parallel vectors (Park et al., 2024; Wu et al., 2025).

# 5. Experiments

**Datasets.** We evaluate all methods on three datasets: (1) Synth is a synthetically generated dataset with two-dimensional inputs and 7 labels. The inputs are designed so that the labels are not linearly separable (see Fig. 2 for an illustration). We train models with $(m = 2)$-dimensional representations. (2) CIFAR 100 (Krizhevsky et al., 2009) consists of natural images from 100 classes. We use representations of dimension $m = 50$. (3) SUB (Bader et al., 2025) is a high-quality synthetic variant of CUB200 (Wah et al., 2011) containing bird images from 33 classes. Each instance is also annotated with a human-interpretable attribute used during generation (e.g., crest color), yielding 33 distinct values. We train the teacher using both class labels and concept annotations, and use $m = 10$.

**Methods.** We fix the gauge on unembeddings by centering them as in Eq. (3). In all experiments, we first train a reference teacher model by minimizing the cross-entropy loss on labeled data. We then compare three types of student models: KL-students, trained to minimize $d_{\text{KL}}$ to the teacher distribution, $L_2$-students and $L_1$-students, trained to minimize $d_{\text{logit}}^2$ and $\mathcal{L}^1_{\text{logit}}$, respectively. We use ResNet (He et al., 2016) and DINOv2 (Oquab et al., 2024) as feature extractors for CIFAR and SUB, respectively. All architectural details, hyperparameter choices, and train–validation–test splits are reported in App. L. Teachers and students share the same architectures and representational dimensionality, except on CIFAR, where we use different ResNet sizes.

**Metrics.** We measure the representational similarity between students and teachers using $m_{\text{CCA}}$ on their embeddings and $d_{\text{rep}}$, and distributional distance with $d_{\text{KL}}$ and $d_{\text{logit}}$. We report classification accuracy on ground-truth

| Synth | Teacher | KL-student | $L_1$-student |
|---|---|---|---|
| 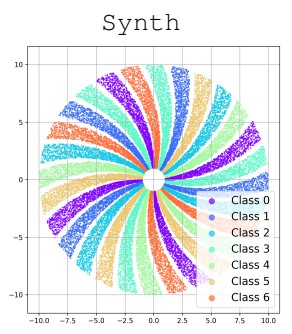 | 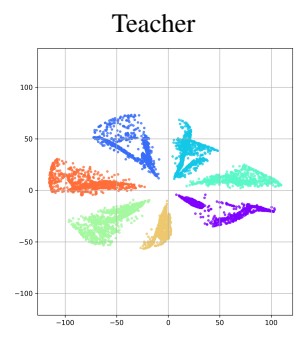 | 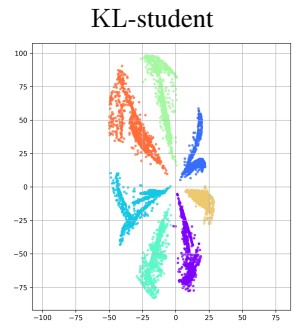 | 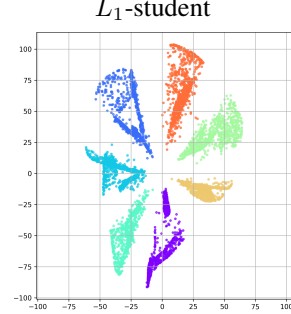 |

*Figure 2.* On the left, input data of the Synth dataset (§5), where inputs are colored based on their labels. The remaining plots display the model embeddings of the teacher and student models. We notice that embeddings of points belonging to **class 1** are nearest neighbors to those of **class 6** and **class 2** for the teacher and similarly for the $L_1$-student, but not for the KL student. Here, the KL-student has low linear similarity to the teacher ($d_{\mathrm{rep}} \approx 6.9$ and $m_{\mathrm{CCA}} \approx 0.62$), while the $L_1$ student has higher similarity ($d_{\mathrm{rep}} \approx 1.3$ and $m_{\mathrm{CCA}} \approx 0.99$).

labels, denoted $\mathrm{Acc}(Y)$. On SUB, we additionally evaluate how well linear classifiers recover human-annotated concepts from the embeddings, reported as $\mathrm{Acc}(C)$. We evaluate all metrics on the test set and average the results over 5 teacher seeds and 5 student seeds per teacher, reporting means and standard deviations. More details and results are provided in App. L.

### 5.1. Distillation with $\mathcal{L}_{\mathrm{logit}}^1$ or $d_{\mathrm{logit}}^2$ yields more similar representations than $d_{\mathrm{KL}}$

All teachers excel in label prediction on Synth, attaining the maximum $\mathrm{Acc}(Y)$ (Tab. 1). On SUB, they exhibit high accuracy for both labels and concepts (Tab. 2). Performance on CIFAR is moderate. The students, compatibly, score close to teachers in terms of $\mathrm{Acc}(Y)$ on all datasets, with students improving over the teachers on SUB. For Synth and SUB, we observe that $L_1$ and $L_2$ students display smaller $d_{\mathrm{KL}}$ than the KL student, and for all datasets $L_1$ and $L_2$ students have lower $d_{\mathrm{logit}}$ (even one order of magnitude below KL students in Synth and SUB, see App. L.5). In all datasets, we observe a drastic increase in similarity when minimizing $\mathcal{L}_{\mathrm{logit}}^1$ and $d_{\mathrm{logit}}$ rather than $d_{\mathrm{KL}}$. $L_1$ and $L_2$ students reduce $d_{\mathrm{rep}}$ by an order of magnitude compared to KL students in Synth, while $L_2$ reduces by two orders in CIFAR, and three orders of magnitude on SUB. Also, $m_{\mathrm{CCA}}$ is always higher for $L_1$ and $L_2$, scoring (almost) perfectly on Synth and SUB and improving by more than 20 pp on CIFAR. This results in more similar representations, as qualitatively displayed in Fig. 2 for the Synth dataset.

### 5.2. Distillation with $\mathcal{L}_{\mathrm{logit}}^1$ or $d_{\mathrm{logit}}^2$ recovers linear concepts of the teacher, while $d_{\mathrm{KL}}$ does not

We experiment on SUB by training teachers to be (linearly) predictive of both labels and concepts from the embeddings. We observe that KL students, have worse $m_{\mathrm{CCA}}$ and $d_{\mathrm{rep}}$ than $L_1$ and $L_2$ students and fare extremely poorly on con-

*Table 1.* **Results on Synth and CIFAR.** Best in bold.

| | | Acc(Y)($\uparrow$) | $d_{\mathrm{rep}}(\downarrow)$ | $m_{\mathrm{CCA}}(\uparrow)$ |
|---|---|---|---|---|
| **Synth** | teach | $0.999 \pm 0.001$ | $-$ | $-$ |
| | KL | $0.999 \pm 0.001$ | $49.5 \pm 2.9$ | $0.580 \pm 0.048$ |
| | $L_1$ | $0.999 \pm 0.001$ | $1.44 \pm 0.04$ | $\mathbf{0.999 \pm 0.001}$ |
| | $L_2$ | $0.999 \pm 0.001$ | $\mathbf{0.20 \pm 0.01}$ | $\mathbf{0.999 \pm 0.001}$ |
| **CIFAR** | teach | $0.540 \pm 0.010$ | $-$ | $-$ |
| | KL | $0.480 \pm 0.001$ | $87.6 \pm 13.7$ | $0.515 \pm 0.001$ |
| | $L_1$ | $0.492 \pm 0.001$ | $4.56 \pm 0.60$ | $\mathbf{0.767 \pm 0.001}$ |
| | $L_2$ | $\mathbf{0.493 \pm 0.001}$ | $\mathbf{0.66 \pm 0.17}$ | $0.763 \pm 0.001$ |

cept classification, scoring slightly above a random classifier ($1/33 \approx 0.03$). On the other hand, $L_1$ and $L_2$ students, while showing lower $\mathrm{Acc}(Y)$, improve over all other metrics. Optimal $m_{\mathrm{CCA}}$ and lower $d_{\mathrm{rep}}$ reflect also in higher $\mathrm{Acc}(C)$ (below teacher by 17 p.p. for $L_1$). This is also qualitatively visible in Fig. 1, where the $2d$ LDA projection on 6 concept attributes shows high linear separability for the $L_1$ student (on the left) but not for KL student (on the right). We further experiment on SUB, varying the representation dimension of students from $m' = 2$ to $m' = 10 = m$. Results in Fig. 8 display an upward trend for $L_1$ and $L_2$ students in all metrics, with always high mCCA. KL students, instead, while rapidly matching (and even surpassing) teacher $\mathrm{Acc}(Y)$, always fare poorly in $\mathrm{Acc}(C)$ and mCCA.

## 6. Discussion and Conclusion

**Limitations. (i)** As in prior works (Khemakhem et al., 2020b; Roeder et al., 2021; Nielsen et al., 2025), we assume that the number of labels exceeds the representation dimension plus one. This is satisfied for most modern language models with large vocabularies (e.g. GPT-3 (Brown et al., 2020), Llama 3 (Meta, 2024), Mistral 7B (Jiang et al., 2023)), but may be restrictive in standard image-classification settings with few classes. Also, systematic experiments are needed to see how using logit distance will interact with and affect optimization in distillation of

larger models. In future work, we would therefore like to do experiments on language models, where our assumptions are likely to be met. Extending our results to regimes where the number of labels is equal to, or smaller than, the representation dimension (Marconato et al., 2025) is another key next step. **(ii)** In our Def. 3.7, we assumed our models to be in general position (Asm. 3.5). This resembles the diversity assumption (Asm. 2.1), but it is stronger. Similar to diversity, it is satisfied with high probability for random vectors in high dimension ($m$ large) under mild conditions. **(iii)** Our results suggest a trade-off between capturing the structure of teacher representations and ignoring information about unlikely labels which might be noisy. If we ignore the unlikely labels, as KL does, then we are not necessarily capturing the structure of teacher representations. On the other hand, if the structure of the teacher representations is important, e.g. we know the teacher has interpretable concepts we want to preserve, then we must also consider the unlikely labels, as the logit distance does. In the case where the structure of unlikely labels of the teacher is not meaningful, the logit distance may steer students to learn a "noisy" behavior. **(iv)** Most of our results assume that the dimensions of the representations of the compared models are the same. In Thm. F.1, we have an initial theoretical result bounding canonical correlations, which allows different dimensions. This bound is illustrated in Fig. 3.

**Notions of representational similarity.** Our results show that having models whose distributions are close in KL divergence does not provide strong guarantees for *linear* representational similarity. If closeness in KL induces a robust notion of representational similarity, it is therefore likely to be of a different kind. Relatedly, Huh et al. (2024) report that standard linear similarity metrics reveal limited convergence, whereas a mutual $k$-nearest-neighbor metric on representation space uncovers stronger alignment across models. Other similarity notions are discussed in App. A.

**Drivers of representational similarity.** Our Thm. 3.3 implies weak linear representational similarity guarantees for KL, which is connected to the cross entropy loss (12); nonetheless, some works report empirical observations of linear representational similarity under the cross entropy loss—e.g., (Reizinger et al., 2025). There may thus be additional factors that drive the observed similarity, for example, learning dynamics in early phases of training (e.g., Frankle et al. (2020); Kapoor et al. (2026)), see also App. A.

**Distillation and representational similarity.** There is also a line of work on similarity-based knowledge distillation that explicitly matches hidden states between teacher and student, for example by aligning embedding spaces or internal features with auxiliary losses (Singh & Wang, 2024; Wang et al., 2024). Our analysis is complementary: rather than adding representational regularizers, we study how

*Table 2.* **Results on SUB**. Teachers are in gray, KL-students in pink, $L_1$-students in light blue, and $L_2$-students in blue.

| $\mathrm{Acc}(Y)(\uparrow)$ | $\mathrm{Acc}(C)(\uparrow)$ | $d_{\mathrm{rep}}(\downarrow)$ | $m_{\mathrm{CCA}}(\uparrow)$ |
|---|---|---|---|
| $0.91 \pm 0.01$ | $0.92 \pm 0.01$ | — | — |
| $\mathbf{0.93 \pm 0.01}$ | $0.06 \pm 0.01$ | $3100 \pm 170$ | $0.42 \pm 0.01$ |
| $0.92 \pm 0.01$ | $\mathbf{0.75 \pm 0.01}$ | $10.0 \pm 0.9$ | $\mathbf{0.99 \pm 0.01}$ |
| $0.92 \pm 0.01$ | $0.72 \pm 0.01$ | $\mathbf{1.6 \pm 0.2}$ | $\mathbf{0.99 \pm 0.01}$ |

purely distributional objectives—KL divergence and our logit-based distance—control (or fail to control) linear representational similarity.

**Conclusion.** We showed that the logit distance is a metric between model distributions and, under a lower bound on conditional probabilities, can be controlled by the KL divergence. Building on this, we introduced the linear identifiability dissimilarity, tightly linked to the identifiability of the model family, and proved that logit distance upper-bounds both this dissimilarity and the mCCA between representations. Empirically, minimizing the logit distance yields student models whose representations are substantially more linearly similar to their teachers' than when training with standard KL-based distillation. An important question for future research is whether these results extend to large-scale experiments, such as the distillation of LLMs.

# Impact Statement

The focus of this paper is to improve our understanding of a model class often used in machine learning (e.g. autoregressive language models). Specifically we analyze the connection between probability distributions of such models and their representations. Although we hope this will lead to some impact in the future, for now this is mostly a theoretical work and there are no immediate societal consequences, which would make sense to highlight here.

# Acknowledgments

We thank Anton Rask Lundborg for pointers to the compositional data analysis literature and David Klindt for interesting discussions. B. M. G. N. was supported by the Danish Pioneer Centre for AI, DNRF grant number P1 and partially by the Novo Nordisk Foundation grant NNF24OC0092612. E.M. acknowledges support from TANGO, Grant Agreement No. 101120763, funded by the European Union. Views and opinions expressed are however those of the author(s) only and do not necessarily reflect those of the European Union or the European Health and Digital Executive Agency (HaDEA). Neither the European Union nor the granting authority can be held responsible for them. S.B. was supported by the Tübingen AI Center. L.G. was supported by the Danish Data Science Academy, which is funded by the Novo Nordisk Foundation (NNF21SA0069429), and by the Pioneer Centre for AI,

DNRF grant number P1.

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

# A. Additional Related Work

In this section we reference additional related work.

**Measures of representational similarity.** In our work, we take an identifiability perspective on the study of representational similarity, under which similarity measures that are invariant to linear transformations are the most natural choice. Our point is not that these are the right similarity measures in general, but rather that they are aligned with the identifiability perspective; those for which we can prove bounds; and they are directly connected to the preservation of linear properties such as those discussed in §4. Our use of $m_{\mathrm{CCA}}$ is also in line with prior work on representational similarity based on CCA, such as SVCCA (Raghu et al., 2017) and PWCCA (Morcos et al., 2018), which likewise measure similarity between embeddings. Other works consider different notions of similarity or dissimilarity, reflecting different objectives and invariance requirements. For instance, Kornblith et al. (2019) introduce CKA, which is invariant only to orthogonal transformations and isotropic scaling, a choice motivated by considerations different from identifiability, such as invariance of gradient-descent dynamics (LeCun et al., 1990); orthogonal Procrustes alignment (Schönemann, 1966) is similarly based on invariance up to orthogonal transformations. For RSA (Kriegeskorte et al., 2008), the precise invariances depend on the choice of similarity functions. Finally, Nielsen et al. (2025) introduce a dissimilarity measure for our model class, $d_{\mathbf{f},\mathbf{g}}$. Whether theory analogous to ours can, under suitable assumptions, be established for such similarity measures is an open question. More broadly, what constitutes the most appropriate measure of representational similarity remains unsettled (Bansal et al., 2021), and existing approaches reflect different objectives and associated trade-offs (Klabunde et al., 2025).

**Other drivers of representational similarity.** Ciernik et al. (2024) show that the training objective strongly impacts how consistently representational similarity generalizes across datasets. Li et al. (2025) find that dataset and task overlap both correlate strongly with increased representational similarity, especially in combination. The role of early stages of training in shaping shared representational structure was investigated by (Frankle et al., 2020). Braun et al. (2025) showed that enforcing robustness to parameter noise during training can force more aligned representations across runs.

**Extent of the theory**. We remark that our theory is not specific to supervised classifiers, but it also encompasses autoregressive language models (Yang et al., 2019; Brown et al., 2020), self-supervised classifiers (Oord et al., 2018; Henaff, 2020), and deep metric learning (Sohn, 2016), see also (Roeder et al., 2021; Marconato et al., 2025; Nielsen et al., 2025).This also implies that, upon meeting our theoretical conditions, the same theoretical results hold for a wide variety of models beyond those we experiment with. Comparing models that define representations on distinct data modalities (Huh et al., 2024), however, is currently out of the scope of our analysis, but future work may consider different views of the same latent, as in (Gresele et al., 2020), as a starting point.

**Interpretable properties in model representations.** A wealth of works studies learning concepts in weakly-supervised settings (Taeb et al., 2022; Marconato et al., 2023; Rajendran et al., 2024; Zheng et al., 2025; Bortolotti et al., 2025; Goyal et al., 2025). Our work differs in spirit from these, as we show which distillation schemes preserve already existing linearly encoded concepts in the teacher representations, especially when distributional equality cannot be guaranteed. The notion we propose of linearly encoded concepts shares some similarities to that of Rajendran et al. (2024), but we focus on probability distributions over categorical values instead of conceptual subspaces. Other interpretable properties can be linearly encoded in model representations, such as parallel directions in model unembeddings (Mikolov et al., 2013; Park et al., 2024) and relational linearity in model embeddings of text sentences (Hernandez et al., 2024; Marconato et al., 2025). Both have also been shown to allowing steering model behavior (Hase et al., 2023; Stolfo et al., 2025). We expect small values of $d_{\mathrm{logit}}$ to preserve these properties as well, but we leave an in-depth analysis to future work.

# B. Definitions and background material

In this section, we recall useful mathematical definitions and results for later proofs of main paper claims.

## B.1. Background on Canonical Correlation Analysis (CCA)

Consider two random vectors $\mathbf{X} \in \mathbb{R}^m$ and $\mathbf{Y} \in \mathbb{R}^{m'}$. The first canonical variables are given by

$$(\mathbf{a}_1, \mathbf{b}_1) = \underset{\mathbf{a}\in\mathbb{R}^m, \mathbf{b}\in\mathbb{R}^{m'}}{\mathrm{argmax}} \ \mathrm{corr}(\mathbf{a}\cdot\mathbf{X}, \mathbf{b}\cdot\mathbf{Y}) \tag{26}$$

where corr denotes the correlation coefficient. And we denote by

$$\rho_1 = \max_{\mathbf{a} \in \mathbb{R}^m, \mathbf{b} \in \mathbb{R}^{m'}} \mathrm{corr}(\mathbf{a} \cdot \mathbf{X}, \mathbf{b} \cdot \mathbf{Y}) \tag{27}$$

the first canonical correlation. The second and all further canonical variables and canonical correlations are defined similarly except that we restrict the maximum to the orthogonal complements of all previous canonical variables. There are a total of $\min(m, m')$ canonical correlations. It can be shown that these correspond to the left and right singular vectors and the singular values of the matrix

$$\mathbf{\Sigma}_{\mathbf{XX}}^{-1/2} \mathbf{\Sigma}_{\mathbf{XY}} \mathbf{\Sigma}_{\mathbf{YY}}^{-1/2} \tag{28}$$

where $\mathbf{\Sigma}_{\mathbf{XX}}$ and $\mathbf{\Sigma}_{\mathbf{YY}}$ are the covariance matrices of $\mathbf{X}$ and $\mathbf{Y}$ and

$$\mathbf{\Sigma}_{\mathbf{XY}} = \mathbb{E}\left((\mathbf{X} - \mathbb{E}(\mathbf{X}))(\mathbf{Y} - \mathbb{E}(\mathbf{Y}))^\top\right) \tag{29}$$

the cross correlation. Clearly, this is invariant to linear transformations of the variables. For $m = m'$ We consider the mean CCA

$$m_{\mathrm{CCA}} = \frac{1}{m} \sum_{i=1}^{m} \rho_i. \tag{30}$$

Variables $\mathbf{X}$ and $\mathbf{Y}$ are linearly related iff $m_{\mathrm{CCA}} = 1$. Sums of canonical correlations can be related to matrix norms through (28), e.g., by

$$\sum_{i=1}^{\min(m,m')} \rho_i^2 = \mathrm{Tr}\left(\mathbf{\Sigma}_{\mathbf{XX}}^{-1/2} \mathbf{\Sigma}_{\mathbf{XY}} \mathbf{\Sigma}_{\mathbf{YY}}^{-1} \mathbf{\Sigma}_{\mathbf{YX}} \mathbf{\Sigma}_{\mathbf{XX}}^{-1/2}\right). \tag{31}$$

### B.2. Von Neumann's trace inequality

For reference we state von Neumann's trace inequality which is a bound for the trace of the product of two positive semi-definite matrices.

**Theorem B.1** (Von Neumann's trace inequality). *For positive semi-definite symmetric (or hermitian) matrices* $\mathbf{A}, \mathbf{B} \in \mathbb{R}^{m \times m}$ *with eigenvalues* $a_1 \geq a_2 \geq \ldots \geq a_m$ *and* $b_1 \geq b_2 \geq \ldots \geq b_m$ *the following bound holds*

$$\sum_{i=1}^{m} a_i b_{m+1-i} \leq \mathrm{Tr}(\mathbf{AB}) \leq \sum_{i=1}^{m} a_i b_i. \tag{32}$$

## C. Theoretical Material for Section 2

### C.1. Proof of Theorem 2.2

*Proof.* Let $(\mathbf{f}, \mathbf{g}), (\mathbf{f}', \mathbf{g}') \in \Theta$ and let $(\mathbf{f}, \mathbf{g})$ satisfy the diversity condition (Asm. 2.1). The theorem then says that

$$p_{\mathbf{f},\mathbf{g}}(y \mid \mathbf{x}) = p_{\mathbf{f}',\mathbf{g}'}(y \mid \mathbf{x}), \ \forall (\mathbf{x}, y) \in \mathcal{X} \times \mathcal{Y} \iff (\mathbf{f}, \mathbf{g}) \sim_L (\mathbf{f}', \mathbf{g}'), \tag{33}$$

and in particular, we can set $\mathbf{A} = \tilde{\mathbf{A}}_{\mathcal{J}}$. We will prove each implication separately.

" $\Longleftarrow$ ": For this direction, we have that $(\mathbf{f}, \mathbf{g}) \sim_L (\mathbf{f}', \mathbf{g}')$. This means that for an invertible matrix $\mathbf{A}$ we have that

$$\mathbf{f}(\mathbf{x}) = \mathbf{A}\mathbf{f}'(\mathbf{x}), \quad \tilde{\mathbf{g}}(y) = \mathbf{A}^{-\top}\tilde{\mathbf{g}}'(y). \tag{34}$$

If we consider the probability the model $(\mathbf{f}, \mathbf{g})$ assigns to a label $y$, we see that for a pivot, $\tilde{y} \in \mathcal{Y}$, we get:

$$p_{\mathbf{f},\mathbf{g}}(y \mid \mathbf{x}) = \frac{\exp(\mathbf{f}(\mathbf{x})^\top \mathbf{g}(y))}{\sum_{y' \in \mathcal{Y}} \exp(\mathbf{f}(\mathbf{x})^\top \mathbf{g}(y'))} \tag{35}$$

$$= \frac{\exp(\mathbf{f}(\mathbf{x})^\top (\mathbf{g}(y) - \mathbf{g}(\tilde{y})))}{\sum_{y' \in \mathcal{Y}} \exp(\mathbf{f}(\mathbf{x})^\top (\mathbf{g}(y') - \mathbf{g}(\tilde{y})))} \tag{36}$$

$$= \frac{\exp((\mathbf{A}\mathbf{f}'(\mathbf{x}))^\top \mathbf{A}^{-\top} (\mathbf{g}'(y) - \mathbf{g}'(\tilde{y})))}{\sum_{y' \in \mathcal{Y}} \exp((\mathbf{A}\mathbf{f}'(\mathbf{x}))^\top \mathbf{A}^{-\top} (\mathbf{g}'(y') - \mathbf{g}'(\tilde{y})))} \tag{37}$$

$$= \frac{\exp(\mathbf{f}'(\mathbf{x})^\top \mathbf{A}^\top \mathbf{A}^{-\top} (\mathbf{g}'(y) - \mathbf{g}'(\tilde{y})))}{\sum_{y' \in \mathcal{Y}} \exp(\mathbf{f}'(\mathbf{x})^\top \mathbf{A}^\top \mathbf{A}^{-\top} (\mathbf{g}'(y') - \mathbf{g}'(\tilde{y})))} \tag{38}$$

$$= \frac{\exp(\mathbf{f}'(\mathbf{x})^\top (\mathbf{g}'(y) - \mathbf{g}'(\tilde{y})))}{\sum_{y' \in \mathcal{Y}} \exp(\mathbf{f}'(\mathbf{x})^\top (\mathbf{g}'(y') - \mathbf{g}'(\tilde{y})))} \tag{39}$$

$$= \frac{\exp(\mathbf{f}'(\mathbf{x})^\top \mathbf{g}'(y))}{\sum_{y' \in \mathcal{Y}} \exp(\mathbf{f}'(\mathbf{x})^\top \mathbf{g}'(y'))} \tag{40}$$

$$= p_{\mathbf{f}',\mathbf{g}'}(y \mid \mathbf{x}) . \tag{41}$$

Where Eqs. (36) and (40) are because translation with a constant vector does not change the probability and Eq. (37) comes from the initial assumption and recalling that $\tilde{\mathbf{g}}(y) := \mathbf{g}(y) - \mathbf{g}(\tilde{y})$.

" $\implies$ ": This proof is very similar to the one in Nielsen et al. (2025), Appendix B, but we include it here for completeness:

We first prove that $p_{\mathbf{f},\mathbf{g}}(y \mid \mathbf{x}) = p_{\mathbf{f}',\mathbf{g}'}(y \mid \mathbf{x})$, $\forall (\mathbf{x}, y) \in \mathcal{X} \times \mathcal{Y} \implies \mathbf{f}(\mathbf{x}) = \mathbf{A}\mathbf{f}'(\mathbf{x})$ for $\mathbf{A} = \mathbf{A}_{\mathcal{J}} = \tilde{\mathbf{L}}_{\mathcal{J}}^{-\top} \tilde{\mathbf{L}}_{\mathcal{J}}^{'\top}$ and $\mathbf{A}_{\mathcal{J}}$ is invertible.

By assumption, the models $(\mathbf{f}, \mathbf{g}), (\mathbf{f}', \mathbf{g}')$ have equal likelihoods. Moreover, by construction, the two models have representations in $\mathbb{R}^m$. Let $Z(\mathbf{x}, \mathcal{Y}) = \sum_{y' \in \mathcal{Y}} \exp(\mathbf{f}(\mathbf{x})^\top \mathbf{g}(y'))$, and similarly for $Z'(\mathbf{x}, \mathcal{Y})$. Then

$$p_{\mathbf{f},\mathbf{g}}(y \mid \mathbf{x}) = p_{\mathbf{f}',\mathbf{g}'}(y \mid \mathbf{x}) \tag{42}$$

$$\implies \mathbf{f}(\mathbf{x})^\top \mathbf{g}(y) - \log(Z(\mathbf{x}, \mathcal{Y})) = \mathbf{f}'(\mathbf{x})^\top \mathbf{g}'(y) - \log(Z'(\mathbf{x}, \mathcal{Y})) \tag{43}$$

for all $\mathbf{x} \in \mathcal{X}$ and all $y \in \mathcal{Y}$. In particular, this equation holds for any fixed $y$. From the diversity condition (Asm. 2.1), we can choose a pivot, $\tilde{y}$ and $m$ $y$'s, $y_1, \ldots, y_m$, such that the displaced unembeddings $\{\tilde{\mathbf{g}}(y_i)\}_{i=1}^m$ are linearly independent. By subtracting the log-conditional distribution for the pivot $\tilde{y} \in \mathcal{Y}$, we obtain for each $y_i \in \mathcal{Y}$ the following:

$$\mathbf{f}(\mathbf{x})^\top \mathbf{g}(y_i) - \mathbf{f}(\mathbf{x})^\top \mathbf{g}(\tilde{y}) + \log(Z(\mathbf{x}, \mathcal{Y})) - \log(Z(\mathbf{x}, \mathcal{Y}))$$
$$= \mathbf{f}'(\mathbf{x})^\top \mathbf{g}'(y_i) - \mathbf{f}'(\mathbf{x})^\top \mathbf{g}'(\tilde{y}) + \log(Z'(\mathbf{x}, \mathcal{Y})) - \log(Z'(\mathbf{x}, \mathcal{Y})) \tag{44}$$

$$\mathbf{f}(\mathbf{x})^\top \mathbf{g}(y_i) - \mathbf{f}(\mathbf{x})^\top \mathbf{g}(\tilde{y}) = \mathbf{f}'(\mathbf{x})^\top \mathbf{g}'(y_i) - \mathbf{f}'(\mathbf{x})^\top \mathbf{g}'(\tilde{y}) \tag{45}$$

$$\mathbf{f}(\mathbf{x})^\top \tilde{\mathbf{g}}(y_i) = \mathbf{f}'(\mathbf{x})^\top \tilde{\mathbf{g}}'(y_i) , \tag{46}$$

where the last passage holds by definition of $\tilde{\mathbf{g}}(y) := \mathbf{g}(y) - \mathbf{g}(\tilde{y})$, see §2. Let $\tilde{\mathbf{L}}_{\mathcal{J}}$ be the matrix which has $\tilde{\mathbf{g}}(y_i)$ as columns and $\tilde{\mathbf{L}}_{\mathcal{J}}^{'}$ be the matrix which has $\tilde{\mathbf{g}}'(y_i)$ as columns. Notice that the diversity condition (Asm. 2.1) implies that $\tilde{\mathbf{L}}_{\mathcal{J}}$ is an invertible matrix. We can then stack the equations to get

$$\tilde{\mathbf{L}}_{\mathcal{J}}^\top \mathbf{f}(\mathbf{x}) = \tilde{\mathbf{L}}_{\mathcal{J}}^{'\top} \mathbf{f}'(\mathbf{x}) \tag{47}$$

and since $\tilde{\mathbf{L}}_{\mathcal{J}}$ is invertible,

$$\mathbf{f}(\mathbf{x}) = \tilde{\mathbf{L}}_{\mathcal{J}}^{-\top} \tilde{\mathbf{L}}_{\mathcal{J}}^{'\top} \mathbf{f}'(\mathbf{x}) . \tag{48}$$

If we set $\mathbf{A} := (\tilde{\mathbf{L}}'_{\mathcal{J}} \tilde{\mathbf{L}}_{\mathcal{J}}^{-1})^\top$, we only need to show that $\mathbf{A}$ is invertible. Using the diversity condition, we pick points $\mathbf{x}_1, ..., \mathbf{x}_m \in \mathcal{X}$ such that the embeddings $\{\mathbf{f}(\mathbf{x}_i)\}_{i=1}^m$ are linearly independent. Let $\mathbf{N}$ be the matrix with $\mathbf{f}(\mathbf{x}_i)$ as columns and $\mathbf{N}'$ be the matrix with $\mathbf{f}'(\mathbf{x}_i)$ as columns. Notice that, from the diversity condition $\mathbf{N}$ is an invertible matrix. Then

$$\mathbf{N} = \mathbf{A}\mathbf{N}'. \tag{49}$$

Since any two matrices, $\mathbf{B}, \mathbf{C} \in \mathbb{R}^{m \times m}$, have the property $\mathrm{rank}(\mathbf{B}\mathbf{C}) \leq \min(\mathrm{rank}(\mathbf{B}), \mathrm{rank}(\mathbf{C}))$, and $\mathbf{N}$ has rank equal to $m$, we have that necessarily also $\mathbf{A}$ and $\mathbf{N}'$ have rank $m$. In particular, $\mathbf{A}$ is an invertible matrix.

Next we prove that $p_{\mathbf{f},\mathbf{g}}(y \mid \mathbf{x}) = p_{\mathbf{f}',\mathbf{g}'}(y \mid \mathbf{x})$, $\forall (\mathbf{x}, y) \in \mathcal{X} \times \mathcal{Y} \implies \tilde{\mathbf{g}}(y) = \mathbf{A}^{-\top}\tilde{\mathbf{g}}'(y)$ for $\mathbf{A} = \mathbf{A}_{\mathcal{J}} = \tilde{\mathbf{L}}_{\mathcal{J}}^{-\top}\tilde{\mathbf{L}}_{\mathcal{J}}^{'\top}$, and $\mathbf{A}_{\mathcal{J}}$ is invertible.

As before, we have that

$$\mathbf{f}(\mathbf{x})^\top \mathbf{g}(y) - \log(Z(\mathbf{x}, \mathcal{Y})) = \mathbf{f}'(\mathbf{x})^\top \mathbf{g}'(y) - \log(Z'(\mathbf{x}, \mathcal{Y})) \tag{50}$$

holds for all $\mathbf{x} \in \mathcal{X}$ and all $y \in \mathcal{Y}$. In particular, this equation holds for any specific $\mathbf{x}$. From the diversity condition (Asm. 2.1), we can choose $m$ $\mathbf{x}$'s, $\mathbf{x}_1, \dots, \mathbf{x}_m$, such that the embeddings $\{\mathbf{f}(\mathbf{x}_i)\}_{i=1}^m$ are linearly independent. By using these values of $\mathbf{x}$ in the equation from above, we get the following:

$$\mathbf{f}(\mathbf{x}_i)^\top \mathbf{g}(y) - \log(Z(\mathbf{x}_i, \mathcal{Y})) = \mathbf{f}'(\mathbf{x}_i)^\top \mathbf{g}'(y) - \log(Z'(\mathbf{x}_i, \mathcal{Y})) \tag{51}$$
$$\mathbf{f}(\mathbf{x}_i)^\top \mathbf{g}(y) = \mathbf{f}'(\mathbf{x}_i)^\top \mathbf{g}'(y) + c_i, \tag{52}$$

where

$$c_i = \log\left(\frac{Z(\mathbf{x}_i, \mathcal{Y})}{Z'(\mathbf{x}_i, \mathcal{Y})}\right). \tag{53}$$

Let $\mathbf{N}$ be the matrix with $\mathbf{f}(\mathbf{x}_i)$ as columns, let $\mathbf{N}'$ be the matrix with $\mathbf{f}'(\mathbf{x}_i)$ as columns and let $\mathbf{c}$ be the vector with $c_i$ as entries. Then, since $\mathbf{N}$ is invertible

$$\mathbf{N}^\top \mathbf{g}(y) = \mathbf{N}'^\top \mathbf{g}'(y) + \mathbf{c} \tag{54}$$
$$\mathbf{g}(y) = \mathbf{N}^{-\top} \mathbf{N}'^\top \mathbf{g}'(y) + \mathbf{N}^{-\top} \mathbf{c}. \tag{55}$$

Since we found in Eq. (49) that $\mathbf{A}$ is invertible, we have that

$$\mathbf{N}' = \mathbf{A}^{-1}\mathbf{N}. \tag{56}$$

Therefore, we get

$$\mathbf{g}(y) = \mathbf{N}^{-\top} \mathbf{N}'^\top \mathbf{g}'(y) + \mathbf{N}^{-\top} \mathbf{c} \tag{57}$$
$$\mathbf{g}(y) = \mathbf{N}^{-\top} (\mathbf{A}^{-1}\mathbf{N})^\top \mathbf{g}'(y) + \mathbf{N}^{-\top} \mathbf{c} \tag{58}$$
$$\mathbf{g}(y) = \mathbf{N}^{-\top} \mathbf{N}^\top \mathbf{A}^{-\top} \mathbf{g}'(y) + \mathbf{N}^{-\top} \mathbf{c} \tag{59}$$
$$\mathbf{g}(y) = \mathbf{A}^{-\top} \mathbf{g}'(y) + \mathbf{b}, \tag{60}$$

where $\mathbf{b} = \mathbf{N}^{-\top}\mathbf{c}$. So, considering the displaced unembedding vectors, we get:

$$\tilde{\mathbf{g}}(y) = \mathbf{g}(y) - \mathbf{g}(\tilde{y}) \tag{61}$$
$$= \mathbf{A}^{-\top} \mathbf{g}'(y) + \mathbf{b} - \mathbf{A}^{-\top} \mathbf{g}'(\tilde{y}) - \mathbf{b} \tag{62}$$
$$= \mathbf{A}^{-\top} \tilde{\mathbf{g}}'(y) \tag{63}$$

This proves the claim.

$\square$

## C.2. The Choice of Pivot Point does not Change the Transformation Matrix

**Proposition C.1.** *For any two models* $(\mathbf{f}, \mathbf{g}), (\mathbf{f}', \mathbf{g}') \in \Theta$ *that satisfy the diversity condition (Asm. 2.1) and such that*

$$p_{\mathbf{f},\mathbf{g}}(y \mid \mathbf{x}) = p_{\mathbf{f}',\mathbf{g}'}(y \mid \mathbf{x}), \quad \forall (\mathbf{x}, y) \in \mathcal{X} \times \mathcal{Y}, \tag{64}$$

*any choice of* $\tilde{y}$ *and* $\tilde{\mathcal{J}} \subseteq \mathcal{Y} \setminus \{\tilde{y}\}$ *such that* $\tilde{\mathbf{L}}_{\mathcal{J}}$ *is invertible, gives the same matrix* $\mathbf{A} = \tilde{\mathbf{A}}_{\mathcal{J}} = \tilde{\mathbf{L}}_{\mathcal{J}}^{-\top} \tilde{\mathbf{L}}_{\mathcal{J}}^{'\top}$.

*Proof.* Consider two difference choices of pivots $\tilde{y}, \hat{y} \in \mathcal{Y}$ and corresponding $m$-dimensional subsets $\mathcal{J} = \{\tilde{y}_1, \ldots, \tilde{y}_m\} \subseteq \mathcal{Y} \setminus \{\tilde{y}\}$ and $\mathcal{K} = \{\hat{y}_1, \ldots, \hat{y}_m\} \subseteq \mathcal{Y} \setminus \{\hat{y}\}$ for which, both

$$\tilde{\mathbf{L}}_{\mathcal{J}} = \left(\mathbf{g}(\tilde{y}_1) - \mathbf{g}(\tilde{y}) \quad \cdots \quad \mathbf{g}(\tilde{y}_m) - \mathbf{g}(\tilde{y})\right), \quad \hat{\mathbf{L}}_{\mathcal{K}} = \left(\mathbf{g}(\hat{y}_1) - \mathbf{g}(\hat{y}) \quad \cdots \quad \mathbf{g}(\hat{y}_m) - \mathbf{g}(\hat{y})\right) \tag{65}$$

are $m \times m$ invertible matrices. Because they are invertible, there exists an invertible matrix $\mathbf{O} \in \mathbb{R}^{m \times m}$ such that

$$\hat{\mathbf{L}}_{\mathcal{K}} = \tilde{\mathbf{L}}_{\mathcal{J}} \mathbf{O}. \tag{66}$$

We now make use of Eq. (64) to rewrite $\mathbf{f}$ in terms of $\mathbf{f}'$ by considering shifted unembeddings $\tilde{\mathbf{g}}(y) := \mathbf{g}(y) - \mathbf{g}(\tilde{y})$

$$\log \frac{p_{\mathbf{f},\mathbf{g}}(y \mid \mathbf{x})}{p_{\mathbf{f},\mathbf{g}}(\tilde{y} \mid \mathbf{x})} = \log \frac{p_{\mathbf{f}',\mathbf{g}'}(y \mid \mathbf{x})}{p_{\mathbf{f}',\mathbf{g}'}(\tilde{y} \mid \mathbf{x})} \tag{67}$$

$$\implies \mathbf{f}(\mathbf{x})^\top \tilde{\mathbf{g}}(y) = \mathbf{f}'(\mathbf{x})^\top \tilde{\mathbf{g}}'(y). \tag{68}$$

In particular, this is true also for $\hat{\mathbf{g}}(y) := \mathbf{g}(y) - \mathbf{g}(\hat{y})$, giving

$$\mathbf{f}(\mathbf{x})^\top \hat{\mathbf{g}}(y) = \mathbf{f}'(\mathbf{x})^\top \hat{\mathbf{g}}'(y). \tag{69}$$

Considering the subsets $\mathcal{J}, \mathcal{K}$ of labels, we can write

$$\tilde{\mathbf{L}}_{\mathcal{J}}^\top \mathbf{f}(\mathbf{x}) = \tilde{\mathbf{L}}_{\mathcal{J}}^{'\top} \mathbf{f}'(\mathbf{x}) \tag{70}$$

$$\hat{\mathbf{L}}_{\mathcal{K}}^\top \mathbf{f}(\mathbf{x}) = \hat{\mathbf{L}}_{\mathcal{K}}^{'\top} \mathbf{f}'(\mathbf{x}), \tag{71}$$

and taking the inverse of $\tilde{\mathbf{L}}_{\mathcal{J}}^\top$ and of $\hat{\mathbf{L}}_{\mathcal{K}}^\top$, we have

$$\mathbf{f}(\mathbf{x}) = \tilde{\mathbf{L}}_{\mathcal{J}}^{-\top} \tilde{\mathbf{L}}_{\mathcal{J}}^{'\top} \mathbf{f}'(\mathbf{x}), \tag{72}$$

$$\mathbf{f}(\mathbf{x}) = \hat{\mathbf{L}}_{\mathcal{K}}^{-\top} \hat{\mathbf{L}}_{\mathcal{K}}^{'\top} \mathbf{f}'(\mathbf{x}). \tag{73}$$

We subtract the two expressions and obtain

$$\left(\tilde{\mathbf{L}}_{\mathcal{J}}^{-\top} \tilde{\mathbf{L}}_{\mathcal{J}}^{'\top} - \hat{\mathbf{L}}_{\mathcal{K}}^{-\top} \hat{\mathbf{L}}_{\mathcal{K}}^{'\top}\right) \mathbf{f}'(\mathbf{x}) = \mathbf{0} \tag{74}$$

$$\left(\tilde{\mathbf{L}}_{\mathcal{K}}^{-\top} \mathbf{O}^\top \tilde{\mathbf{L}}_{\mathcal{J}}^{'\top} - \hat{\mathbf{L}}_{\mathcal{K}}^{-\top} \hat{\mathbf{L}}_{\mathcal{K}}^{'\top}\right) \mathbf{f}'(\mathbf{x}) = \mathbf{0} \tag{75}$$

$$\tilde{\mathbf{L}}_{\mathcal{K}}^{-\top} \left(\mathbf{O}^\top \tilde{\mathbf{L}}_{\mathcal{J}}^{'\top} - \hat{\mathbf{L}}_{\mathcal{K}}^{'\top}\right) \mathbf{f}'(\mathbf{x}) = \mathbf{0}. \tag{76}$$

Now, the diversity condition for the two models gives that $\hat{\mathbf{L}}_{\mathcal{K}}$ is an invertible matrix and that $\mathbf{f}'$ spans the whole $\mathbb{R}^m$. This means that

$$\mathbf{O}^\top \tilde{\mathbf{L}}_{\mathcal{J}}^{'\top} - \hat{\mathbf{L}}_{\mathcal{K}}^{'\top} = \mathbf{0} \tag{77}$$

and so we get

$$\hat{\mathbf{L}}_{\mathcal{K}}^{'\top} = \mathbf{O}^\top \tilde{\mathbf{L}}_{\mathcal{J}}^{'\top}. \tag{78}$$

Combining Eq. (66) and Eq. (78), we get:

$$\mathbf{A} = \hat{\mathbf{L}}_{\mathcal{K}}^{-\top} \hat{\mathbf{L}}_{\mathcal{K}}^{'\top} \tag{79}$$

$$= \tilde{\mathbf{L}}_{\mathcal{J}}^{-\top} \mathbf{O}^{-\top} \mathbf{O}^\top \tilde{\mathbf{L}}_{\mathcal{J}}^{'\top} \tag{80}$$

$$= \tilde{\mathbf{L}}_{\mathcal{J}}^{-\top} \tilde{\mathbf{L}}_{\mathcal{J}}^{'\top} \tag{81}$$

which shows the claim. $\qquad \square$

# D. Theoretical Material for first part of Section 3: How Logit Distance is Connected to Log Probabilities and Proof that it is a Metric

In this section we first show how the logit distance, $d_{\text{logit}}(p, p')$, from Def. 3.1 can be written as a difference between log probabilities of sets of distributions with only non-zero probabilities, App. D.1. We then show that logit distance is a proper metric, App. D.3.

## D.1. Logit Distance in Terms of log Probabilities

We recall that we defined the logits of models from our model class as:

$$\mathbf{u}(\mathbf{x}) = \begin{pmatrix} \mathbf{g}(y_1)^\top \mathbf{f}(\mathbf{x}) \\ \vdots \\ \mathbf{g}(y_k)^\top \mathbf{f}(\mathbf{x}) \end{pmatrix} \tag{82}$$

We will show that the logit distance is equal to a difference of log probabilities as in the following result:

**Lemma D.1.** *Let $p, p'$, be distributions generated by the model class Eq. (1), where $m$ is the dimension of the representations. Let $N = \binom{k-2}{m-1}$. When evaluating $d_{\text{logit}}(p, p')$, we have that*

$$\|\mathbf{u}(\mathbf{x}) - \mathbf{u}'(\mathbf{x})\|_2^2 = \frac{1}{2kN} \sum_{\tilde{y} \in \mathcal{Y}} \sum_{\mathcal{J} \subseteq \mathcal{Y} \setminus \{\tilde{y}\}} \left\| \left( \log \left( \frac{p(y_j|\mathbf{x})}{p'(y_j|\mathbf{x})} \right) - \log \left( \frac{p(\tilde{y}|\mathbf{x})}{p'(\tilde{y}|\mathbf{x})} \right) \right)_{j \in \mathcal{J}} \right\|_2^2. \tag{83}$$

*where on the right we have a vector indexed with $j$ running over the set $\mathcal{J}$.*

We show the result in two steps: First we show that the logit distance can be written in terms of shifted logits, defined in the following way:

$$\mathbf{u}_i(\mathbf{x}) = \begin{pmatrix} \mathbf{g}(y_1)^\top \mathbf{f}(\mathbf{x}) \\ \vdots \\ \mathbf{g}(y_k)^\top \mathbf{f}(\mathbf{x}) \end{pmatrix} - \mathbf{g}(y_i)^\top \mathbf{f}(\mathbf{x}) \begin{pmatrix} 1 \\ \vdots \\ 1 \end{pmatrix} \tag{84}$$

Then we will show that the shifted logits are the same as a difference of log probabilities.

**Proposition D.2.** *The following relation holds*

$$\|\mathbf{u}(\mathbf{x}) - \mathbf{u}'(\mathbf{x})\|_2^2 = \frac{1}{2k} \sum_{y_i \in \mathcal{Y}} \|\mathbf{u}_i(\mathbf{x}) - \mathbf{u}_i'(\mathbf{x})\|_2^2 = \frac{1}{2k\binom{k-2}{m-1}} \sum_{y_i \in \mathcal{Y}} \sum_{\mathcal{J} \subseteq \mathcal{Y} \setminus \{y_i\}, |\mathcal{J}|=m} \sum_{j \in \mathcal{J}} (u_i(\mathbf{x})_j - u_i'(\mathbf{x})_j)^2. \tag{85}$$

*Remark* D.3. The first identity is essentially a version of the well-known identity

$$\mathbb{E}(\mathbf{X} - \mathbf{X}')^2 = 2\text{Var}(\mathbf{X}) \tag{86}$$

holding true for two independent copies $\mathbf{X}$ and $\mathbf{X}'$ of a random variable.

*Proof.* We observe that since we assume that the unembeddings $\mathbf{g}(y_i)$ and $\mathbf{g}'(y_i)$ are centred we get

$$(\mathbf{u}(\mathbf{x}) - \mathbf{u}'(\mathbf{x})) \cdot \begin{pmatrix} 1 \\ \vdots \\ 1 \end{pmatrix} = \sum_{i=1}^{k} \mathbf{f}(\mathbf{x})^\top \mathbf{g}(y_i) - \mathbf{f}'(\mathbf{x})^\top \mathbf{g}'(y_i) = \mathbf{f}(\mathbf{x})^\top \left( \sum_{i=1}^{k} \mathbf{g}(y_i) \right) - \mathbf{f}'(\mathbf{x})^\top \left( \sum_{i=1}^{k} \mathbf{g}'(y_i) \right) = 0. \tag{87}$$

We then note that using the definition of $\mathbf{u}_i$ and $\mathbf{u}'_i$ and then expand the squared norm we get

$$
\begin{aligned}
\|\mathbf{u}_i(\mathbf{x}) - \mathbf{u}'_i(\mathbf{x})\|^2 &= \|\mathbf{u}(\mathbf{x}) - \mathbf{u}'(\mathbf{x})\|^2 - 2(\mathbf{u}(\mathbf{x}) - \mathbf{u}'(\mathbf{x})) \cdot \begin{pmatrix} 1 \\ \vdots \\ 1 \end{pmatrix} (\mathbf{g}(y_i)^\top \mathbf{f}(\mathbf{x}) - \mathbf{g}'(y_i)^\top \mathbf{f}'(\mathbf{x})) \\
&\quad + (\mathbf{g}(y_i)^\top \mathbf{f}(\mathbf{x}) - \mathbf{g}'(y_i)^\top \mathbf{f}'(\mathbf{x}))^2 \left\| \begin{pmatrix} 1 \\ \vdots \\ 1 \end{pmatrix} \right\|^2 \\
&= \|\mathbf{u}(\mathbf{x}) - \mathbf{u}'(\mathbf{x})\|^2 + k((u(\mathbf{x}))_i - (u'(\mathbf{x}))_i)^2.
\end{aligned}
\tag{88}
$$

Summing this over $i$ we get

$$
\sum_{i=1}^{k} \|\mathbf{u}_i(\mathbf{x}) - \mathbf{u}'_i(\mathbf{x})\|^2 = k\|\mathbf{u}(\mathbf{x}) - \mathbf{u}'(\mathbf{x})\|^2 + k\sum_{i=1}^{k}((u(\mathbf{x}))_i - (u'(\mathbf{x}))_i)^2 = 2k\|\mathbf{u}(\mathbf{x}) - \mathbf{u}'(\mathbf{x})\|^2
\tag{89}
$$

and therefore the first relation. For the second relation we first note that by definition

$$
u_i(\mathbf{x})_i - u'_i(\mathbf{x})_i = (\mathbf{f}(\mathbf{x})^\top \mathbf{g}(y_i) - \mathbf{f}(\mathbf{x})^\top \mathbf{g}(y_i)) - (\mathbf{f}'(\mathbf{x})^\top \mathbf{g}'(y_i) - \mathbf{f}'(\mathbf{x})^\top \mathbf{g}'(y_i)) = 0.
\tag{90}
$$

Then we use that each index $j \neq i$ appears in $\binom{k-2}{m-1}$ subsets of size $m$ not containing index $i$. Indeed, we have to select $m-1$ remaining points of the set of size $m$ from $k-2$ options ($i$ and $j$ are removed from the $k$ options). Therefore, for each pivot $y_i$

$$
\begin{aligned}
\sum_{\mathcal{J} \subseteq \mathcal{Y}\setminus\{y_i\}, |\mathcal{J}|=m} \sum_{j \in \mathcal{J}} (u_i(\mathbf{x})_j - u'_i(\mathbf{x})_j)^2 &= \binom{k-2}{m-1} \sum_{j \neq i} (u_i(\mathbf{x})_j - u'_i(\mathbf{x})_j)^2 \\
&= \binom{k-2}{m-1} \sum_{j=1}^{k} (u_i(\mathbf{x})_j - u'_i(\mathbf{x})_j)^2 \\
&= \binom{k-2}{m-1} \|\mathbf{u}_i(\mathbf{x}) - \mathbf{u}'_i(\mathbf{x})\|^2.
\end{aligned}
\tag{91}
$$

This implies the second part of (85) by summing over $y_i$.

$\square$

We recall that the matrix of shifted unembeddings constructed from $\tilde{\mathbf{g}}$ using a subset of $m$ labels $\mathcal{J} = \{y_1, \ldots, y_m\} \subseteq \mathcal{Y} \setminus \{\tilde{y}\}$:

$$
\tilde{\mathbf{g}}(y) := \mathbf{g}(y) - \mathbf{g}(\tilde{y}), \quad \tilde{\mathbf{L}}_{\mathcal{J}} := \begin{pmatrix} \tilde{\mathbf{g}}(y_1) & \cdots & \tilde{\mathbf{g}}(y_m) \end{pmatrix} \in \mathbb{R}^{m \times m}.
\tag{92}
$$

With this definition we can also write the following result.

**Lemma D.4.** *Let $N = \binom{k-2}{m-1}$. When evaluating $d_{\text{logit}}(p, p')$ on distributions, $p, p'$, generated by the model class Eq. (1), where $m$ is the dimension of the representations, we have that:*

$$
\|\mathbf{u}(\mathbf{x}) - \mathbf{u}'(\mathbf{x})\|^2 = \frac{1}{2kN} \sum_{\tilde{y} \in \mathcal{Y}} \sum_{\mathcal{J} \subseteq \mathcal{Y}\setminus\{\tilde{y}\}} \|\tilde{\mathbf{L}}_{\mathcal{J}}^\top \mathbf{f}(\mathbf{x}) - \tilde{\mathbf{L}}_{\mathcal{J}}'^\top \mathbf{f}'(\mathbf{x})\|_2^2
\tag{93}
$$

*Proof.* This is since taking the entries of the shifted logits $\mathbf{u}_i(\mathbf{x})$ corresponding to the $m$ labels from $\mathcal{J}$, is the same as $\tilde{\mathbf{L}}_{\mathcal{J}}^\top \mathbf{f}(\mathbf{x})$. $\square$

**Lemma D.5.** *When evaluating $d_{\text{logit}}(p, p')$ on distributions, $p, p'$, generated by the model class Eq. (1), where $m$ is the dimension of the representations, we have that:*

$$
\|\tilde{\mathbf{L}}_{\mathcal{J}}^\top \mathbf{f}(\mathbf{x}) - \tilde{\mathbf{L}}_{\mathcal{J}}'^\top \mathbf{f}'(\mathbf{x})\|_2 = \left\| \left( \log\left(\frac{p(y_j|\mathbf{x})}{p'(y_j|\mathbf{x})}\right) - \log\left(\frac{p(\tilde{y}|\mathbf{x})}{p'(\tilde{y}|\mathbf{x})}\right) \right)_{j \in \mathcal{J}} \right\|_2
\tag{94}
$$

*where on the right we have a vector with entries indexed by $j$ running over the set $\mathcal{J}$.*

*Proof.* We consider first the definition of the norm:

$$\|\tilde{\mathbf{L}}_{\mathcal{J}}^{\top}\mathbf{f}(\mathbf{x}) - \tilde{\mathbf{L}}_{\mathcal{J}}^{'\top}\mathbf{f}'(\mathbf{x})\|_2^2 = \sum_{j=1}^{m}(\mathbf{f}(\mathbf{x})^{\top}\mathbf{g}(y_j) - \mathbf{f}'(\mathbf{x})^{\top}\mathbf{g}'(y_j) - \mathbf{f}(\mathbf{x})^{\top}\mathbf{g}(\tilde{y}) + \mathbf{f}'(\mathbf{x})^{\top}\mathbf{g}'(\tilde{y}))^2 \tag{95}$$

We then consider each term of the sum, and see that if we both add and subtract a normalization term for each model, then the value will be unchanged. That is, we have

$$\mathbf{f}(\mathbf{x})^{\top}\mathbf{g}(y_j) - \mathbf{f}(\mathbf{x})^{\top}\mathbf{g}(\tilde{y}) = \log(\exp(\mathbf{f}(\mathbf{x})^{\top}\mathbf{g}(y_j))) - \log(\exp(\mathbf{f}(\mathbf{x})^{\top}\mathbf{g}(\tilde{y})))$$
$$- \log\left(\sum_{y_n}\exp(\mathbf{f}(\mathbf{x})^{\top}\mathbf{g}(y_n))\right) + \log\left(\sum_{y_n}\exp(\mathbf{f}(\mathbf{x})^{\top}\mathbf{g}(y_n))\right) \tag{96}$$
$$= \log(p(y_j|\mathbf{x})) - \log(p(\tilde{y}|\mathbf{x}))$$

And thus we see that

$$\|\tilde{\mathbf{L}}_{\mathcal{J}}^{\top}\mathbf{f}(\mathbf{x}) - \tilde{\mathbf{L}}_{\mathcal{J}}^{'\top}\mathbf{f}'(\mathbf{x})\|_2^2 = \sum_{j=1}^{m}(\log(p(y_j|\mathbf{x})) - \log(p(\tilde{y}|\mathbf{x})) - \log(p'(y_j|\mathbf{x})) + \log(p'(\tilde{y}|\mathbf{x})))^2 \tag{97}$$

$$= \sum_{j=1}^{m}\left(\log\left(\frac{p(y_j|\mathbf{x})}{p'(y_j|\mathbf{x})}\right) - \log\left(\frac{p(\tilde{y}|\mathbf{x})}{p'(\tilde{y}|\mathbf{x})}\right)\right)^2 \tag{98}$$

$$= \left\|\left(\log\left(\frac{p(y_j|\mathbf{x})}{p'(y_j|\mathbf{x})}\right) - \log\left(\frac{p(\tilde{y}|\mathbf{x})}{p'(\tilde{y}|\mathbf{x})}\right)\right)_{j\in\mathcal{J}}\right\|_2^2. \tag{99}$$

$\square$

### D.2. Connection between the Logit Distance and the Aitchison Distance

In this section we show how the Logit Distance is related to the Aitchison distance. The Aitchison distance takes inputs from the hull of the simplex for $C$ values defined as,

$$\Delta^{C-1} = \left\{\mathbf{z} = \{z_1, ..., z_C\}|z_i > 0, i = 1, ..., C, \sum_{i=1}^{C}z_i = 1\right\}, \tag{100}$$

and the distance is defined for $\mathbf{z}, \mathbf{w} \in \Delta^{C-1}$ as

$$d_a(\mathbf{z}, \mathbf{w}) = \sqrt{\frac{1}{2C}\sum_{i=1}^{C}\sum_{j=1}^{C}\left(\log\left(\frac{z_j}{z_i}\right) - \log\left(\frac{w_j}{w_i}\right)\right)^2}. \tag{101}$$

Since our models always assign non-zero probabilities and sum to 1, the probabilities are in the hull of the simplex. This means that the squared distance between logits is the same as the Aitchison distance between the probabilities for a given input, $\mathbf{x}$.

**Proposition D.6.** *Let $p, p'$ be sets of probabilties from our model class and let $\mathbf{u}(\mathbf{x}), \mathbf{u}'(\mathbf{x})$ be the logits. Then for each $\mathbf{x}$ we have*

$$\|\mathbf{u}(\mathbf{x}) - \mathbf{u}'(\mathbf{x})\|_2^2 = d_a^2(p(\cdot|\mathbf{x}), p'(\cdot|\mathbf{x})). \tag{102}$$

*Proof.* We see that

$$(\mathbf{u}_i(\mathbf{x}) - \mathbf{u}_i'(\mathbf{x}))_j = \mathbf{f}(\mathbf{x})^{\top}\mathbf{g}(y_j) - \mathbf{f}(\mathbf{x})^{\top}\mathbf{g}(y_i) - \mathbf{f}'(\mathbf{x})^{\top}\mathbf{g}'(y_j) + \mathbf{f}'(\mathbf{x})^{\top}\mathbf{g}'(y_i) = \frac{\log(p(y_j|\mathbf{x}))}{\log(p(y_i|\mathbf{x}))} - \frac{\log(p'(y_j|\mathbf{x}))}{\log(p'(y_i|\mathbf{x}))}. \tag{103}$$

Which means that

$$\|\mathbf{u}(\mathbf{x}) - \mathbf{u}'(\mathbf{x})\|_2^2 = \frac{1}{2k} \sum_{y_i \in \mathcal{Y}} \|\mathbf{u}_i(\mathbf{x}) - \mathbf{u}_i'(\mathbf{x})\|_2^2 \tag{104}$$

$$= \frac{1}{2k} \sum_{i=1}^{k} \sum_{j=1}^{k} \left( \frac{\log(p(y_j|\mathbf{x}))}{\log(p(y_i|\mathbf{x}))} - \frac{\log(p'(y_j|\mathbf{x}))}{\log(p'(y_i|\mathbf{x}))} \right)^2 \tag{105}$$

$$= d_a^2(p(\cdot|\mathbf{x}), p'(\cdot|\mathbf{x})) \tag{106}$$

$$\square$$

### D.3. The Logit Distance is a Metric

In this section we provide a full proof that the logit distance is a metric. We will begin by defining a distance between sets of probability distributions which are not necessarily from our model class:

**Definition D.7.** *Let $p, p'$ be two sets of conditional distributions with $k$ labels and only non-zero probabilities. Let $m \in \mathbb{N}$, $N = \binom{k-2}{m-1}$. The shifted log-likelihood distance for $p, p'$ is defined as:*

$$d_{\text{LLD}}^m(p, p') = \frac{1}{2kN} \sum_{\tilde{y} \in \mathcal{Y}} \sum_{\mathcal{J} \subseteq \mathcal{Y} \setminus \{\tilde{y}\}} \mathbb{E}_{\mathbf{x} \sim p_{\mathbf{x}}} \left\| \left( \log \left( \frac{p(y_j|\mathbf{x})}{p'(y_j|\mathbf{x})} \right) - \log \left( \frac{p(\tilde{y}|\mathbf{x})}{p'(\tilde{y}|\mathbf{x})} \right) \right)_j \right\|_2 \tag{107}$$

**Theorem D.8.** *For any $m \in \mathbb{N}$, the function from Def. D.7 is a metric (which can have infinite value) between sets of conditional distributions, $p, p'$, over $k$ labels with non-zero probabilities.*

*Proof.* We will show that Def. D.7 is a metric. That is,

   i  it is non-negative, $d_{\text{LLD}}^m(p, p') \geq 0$, and zero only when all distributions are equal, $d_{\text{LLD}}^m(p, p') = 0 \iff p = p'$.

  ii  It is symmetric, $d_{\text{LLD}}^m(p, p') = d_{\text{LLD}}^m(p', p)$.

 iii  It satisfies the triangle inequality, $d_{\text{LLD}}^m(p_1, p_2) \leq d_{\text{LLD}}^m(p_1, p_3) + d_{\text{LLD}}^m(p_3, p_2)$.

W.r.t. i) we see that Eq. (107) is non-negative, since it is a norm. We will show that if $p = p'$, then $d_{\text{LLD}}^m(p, p') = 0$: When $p = p'$, $p(y_j|\mathbf{x}) = p'(y_j|\mathbf{x})$ for any choice of $\{y_j\}_{j=0}^m$, therefore $d_{\text{LLD}}^m(p, p') = 0$. We will now show that if $d_{\text{LLD}}^m(p, p') = 0$ then $p = p'$: Assume $d_{\text{LLD}}^m(p, p') = 0$, that is:

$$\frac{1}{2kN} \sum_{\tilde{y} \in \mathcal{Y}} \sum_{\mathcal{J} \subseteq \mathcal{Y} \setminus \{\tilde{y}\}} \mathbb{E}_{\mathbf{x} \sim p_{\mathbf{x}}} \left\| \left( \log \left( \frac{p(y_j|\mathbf{x})}{p'(y_j|\mathbf{x})} \right) - \log \left( \frac{p(\tilde{y}|\mathbf{x})}{p'(\tilde{y}|\mathbf{x})} \right) \right)_j \right\| = 0 \tag{108}$$

Since the norm is always non-negative, this means that

$$\mathbb{E}_{\mathbf{x} \sim p_{\mathbf{x}}} \left\| \left( \log \left( \frac{p(y_j|\mathbf{x})}{p'(y_j|\mathbf{x})} \right) - \log \left( \frac{p(\tilde{y}|\mathbf{x})}{p'(\tilde{y}|\mathbf{x})} \right) \right)_j \right\| = 0, \ \forall \{y_j\}_{j=0}^m \subseteq \mathcal{Y} \tag{109}$$

Which means that for all choices of $\{y_j\}_{j=0}^m \subseteq \mathcal{Y}$, we have that each entry in the vector $\left( \log \left( \frac{p(y_j|\mathbf{x})}{p'(y_j|\mathbf{x})} \right) - \log \left( \frac{p(\tilde{y}|\mathbf{x})}{p'(\tilde{y}|\mathbf{x})} \right) \right)_j$ is zero almost everywhere[5].

That is, we have

$$\log(p(y_j|\mathbf{x})) - \log(p(\tilde{y}|\mathbf{x})) - \log(p'(y_j|\mathbf{x})) + \log(p'(\tilde{y}|\mathbf{x})) = 0 \tag{110}$$

$$\log(p(y_j|\mathbf{x})) - \log(p(\tilde{y}|\mathbf{x})) = \log(p'(y_j|\mathbf{x})) - \log(p'(\tilde{y}|\mathbf{x})) \tag{111}$$

$$\log \left( \frac{p(y_j|\mathbf{x})}{p(\tilde{y}|\mathbf{x})} \right) = \log \left( \frac{p'(y_j|\mathbf{x})}{p'(\tilde{y}|\mathbf{x})} \right) \tag{112}$$

$$\frac{p(y_j|\mathbf{x})}{p(\tilde{y}|\mathbf{x})} = \frac{p'(y_j|\mathbf{x})}{p'(\tilde{y}|\mathbf{x})} \tag{113}$$

---

[5]Almost everywhere, also abbreviated a.e.: Everywhere except on some null-set, that is except on a set with measure zero.

for all $y_j, \tilde{y} \in \mathcal{Y}$. Now since $p(y_i|\mathbf{x})$ and $p'(y_i|\mathbf{x})$ are probability distributions, we have that

$$\frac{1}{p(\tilde{y}|\mathbf{x})} = \sum_{j=0}^{c} \frac{p(y_j|\mathbf{x})}{p(\tilde{y}|\mathbf{x})} = \sum_{j=0}^{c} \frac{p'(y_j|\mathbf{x})}{p'(\tilde{y}|\mathbf{x})} = \frac{1}{p'(\tilde{y}|\mathbf{x})} \tag{114}$$

which means that Eq. (113) gives us

$$p(y_j|\mathbf{x}) = p'(y_j|\mathbf{x}), \forall y_j \in \mathcal{Y} \tag{115}$$

and almost everywhere w.r.t. $\mathbf{x}$ thus, $p = p'$ a.e..

W.r.t. ii) Eq. (107) is symmetric, since the norm is symmetric.

W.r.t. iii): Assume we have three sets of distributions, $p_1, p_2, p_3$ and let us consider $d_{\text{LLD}}^m(p_1, p_2)$.

$$\frac{1}{2kN} \sum_{\tilde{y}\in\mathcal{Y}} \sum_{\mathcal{J}\subseteq\mathcal{Y}\setminus\{\tilde{y}\}} \mathbb{E}_{\mathbf{x}\sim p_\mathbf{x}} \left\| \left( \log\left(\frac{p(y_j|\mathbf{x})}{p'(y_j|\mathbf{x})}\right) - \log\left(\frac{p(\tilde{y}|\mathbf{x})}{p'(\tilde{y}|\mathbf{x})}\right) \right)_j \right\| \tag{116}$$

$$= \frac{1}{2kN} \sum_{\tilde{y}\in\mathcal{Y}} \sum_{\mathcal{J}\subseteq\mathcal{Y}\setminus\{\tilde{y}\}} \mathbb{E}_{\mathbf{x}\sim p_\mathbf{x}} \left\| \left( \log(p_1(y_j|\mathbf{x}')) - \log(p_2(y_j|\mathbf{x}')) - \log(p_1(\tilde{y}|\mathbf{x}')) + \log(p_2(\tilde{y}|\mathbf{x}')) \right)_j \right\| \tag{117}$$

$$= \frac{1}{2kN} \sum_{\tilde{y}\in\mathcal{Y}} \sum_{\mathcal{J}\subseteq\mathcal{Y}\setminus\{\tilde{y}\}} \mathbb{E}_{\mathbf{x}\sim p_\mathbf{x}} \| (\log(p_1(y_j|\mathbf{x}')) - \log(p_3(y_j|\mathbf{x}')) + \log(p_3(y_j|\mathbf{x}')) - \log(p_2(y_j|\mathbf{x}'))$$
$$- \log(p_1(\tilde{y}|\mathbf{x}')) + \log(p_3(\tilde{y}|\mathbf{x}')) - \log(p_3(\tilde{y}|\mathbf{x}')) + \log(p_2(\tilde{y}|\mathbf{x}')))_j \| \tag{118}$$

$$= \frac{1}{2kN} \sum_{\tilde{y}\in\mathcal{Y}} \sum_{\mathcal{J}\subseteq\mathcal{Y}\setminus\{\tilde{y}\}} \mathbb{E}_{\mathbf{x}\sim p_\mathbf{x}} \left\| \left( \log(\frac{p_1(y_j|\mathbf{x}')}{p_3(y_j|\mathbf{x}')}) - \log(\frac{p_1(\tilde{y}|\mathbf{x}')}{p_3(\tilde{y}|\mathbf{x}')}) + \log(\frac{p_3(y_j|\mathbf{x}')}{p_2(y_j|\mathbf{x}')} - \log(\frac{p_3(\tilde{y}|\mathbf{x}')}{p_2(\tilde{y}|\mathbf{x}')}) )_j \right\| \tag{119}$$

$$\leq \frac{1}{2kN} \sum_{\tilde{y}\in\mathcal{Y}} \sum_{\mathcal{J}\subseteq\mathcal{Y}\setminus\{\tilde{y}\}} \mathbb{E}_{\mathbf{x}\sim p_\mathbf{x}} \left( \left\| (\log(\frac{p_1(y_j|\mathbf{x}')}{p_3(y_j|\mathbf{x}')}) - \log(\frac{p_1(\tilde{y}|\mathbf{x}')}{p_3(\tilde{y}|\mathbf{x}')}))_j \right\| + \left\| (\log(\frac{p_3(y_j|\mathbf{x}')}{p_2(y_j|\mathbf{x}')} - \log(\frac{p_3(\tilde{y}|\mathbf{x}')}{p_2(\tilde{y}|\mathbf{x}')}))_j \right\| \right) \tag{120}$$

$$= \frac{1}{2kN} \sum_{\tilde{y}\in\mathcal{Y}} \sum_{\mathcal{J}\subseteq\mathcal{Y}\setminus\{\tilde{y}\}} \mathbb{E}_{\mathbf{x}\sim p_\mathbf{x}} \left\| (\log(\frac{p_1(y_j|\mathbf{x}')}{p_3(y_j|\mathbf{x}')}) - \log(\frac{p_1(\tilde{y}|\mathbf{x}')}{p_3(\tilde{y}|\mathbf{x}')}))_j \right\| \tag{121}$$

$$+ \frac{1}{2kN} \sum_{\tilde{y}\in\mathcal{Y}} \sum_{\mathcal{J}\subseteq\mathcal{Y}\setminus\{\tilde{y}\}} \mathbb{E}_{\mathbf{x}\sim p_\mathbf{x}} \left\| (\log(\frac{p_3(y_j|\mathbf{x}')}{p_2(y_j|\mathbf{x}')}) - \log(\frac{p_3(\tilde{y}|\mathbf{x}')}{p_2(\tilde{y}|\mathbf{x}')}))_j \right\| \tag{122}$$

In other words: $d_{\text{LLD}}^m(p_1, p_2) \leq d_{\text{LLD}}^m(p_1, p_3) + d_{\text{LLD}}^m(p_3, p_2)$. $\qquad \square$

We are now ready to prove the theorem:

**Theorem D.9.** *For any two models from our model class, the logit distance from Def. 3.1 is a metric between their sets of distributions, $p, p'$.*

*Proof.* We will show that Def. 3.1 is a metric. That is,

   i it is non-negative, $d_{\text{logit}}(p, p') \geq 0$, and zero only when all distributions are equal, $d_{\text{logit}}(p, p') = 0 \iff p = p'$.

   ii It is symmetric, $d_{\text{logit}}(p, p') = d_{\text{logit}}(p', p)$.

   iii It satisfies the triangle inequality, $d_{\text{logit}}(p_1, p_2) \leq d_{\text{logit}}(p_1, p_3) + d_{\text{logit}}(p_3, p_2)$.

W.r.t. i) we see that Eq. (9) is non-negative, since it is a norm. We will now show that if $p = p'$, then $d_{\text{logit}}(p, p') = 0$.
Assume $p = p'$. Then, $p(y_j|\mathbf{x}) = p'(y_j|\mathbf{x})$ for all $j \in \{1, ..., k\}$ and $\mathbf{x}$ a.e.. Since we saw in Lemma D.1 that

$$\|\mathbf{u}(\mathbf{x}) - \mathbf{u}'(\mathbf{x})\|^2 = \frac{1}{2kN} \sum_{\tilde{y}\in\mathcal{Y}} \sum_{\mathcal{J}\subseteq\mathcal{Y}\setminus\{\tilde{y}\}} \left\| \left( \log\left(\frac{p(y_j|\mathbf{x})}{p'(y_j|\mathbf{x})}\right) - \log\left(\frac{p(\tilde{y}|\mathbf{x})}{p'(\tilde{y}|\mathbf{x})}\right) \right)_j \right\|_2^2 \tag{123}$$

We have that for each $\mathbf{x}$, $p(y_j|\mathbf{x}) = p'(y_j|\mathbf{x}), \forall j \in \{1, ..., k\} \implies ||\mathbf{u}(\mathbf{x}) - \mathbf{u}'(\mathbf{x})||^2 = 0$, which in turn gives us that $||\mathbf{u}(\mathbf{x}) - \mathbf{u}'(\mathbf{x})|| = 0$. So when $p = p'$, $d_{\text{logit}}(p, p') = 0$ for $\mathbf{x}$ a.e..

We will now show that if $d_{\text{logit}}(p, p') = 0$ then $p = p'$ a.e.. Assume $d_{\text{logit}}(p, p') = 0$. Then $||\mathbf{u}(\mathbf{x}) - \mathbf{u}'(\mathbf{x})|| = 0$ for $\mathbf{x}$ a.e. and Lemma D.1 gives us that

$$\left\| \left( \log\left( \frac{p(y_j|\mathbf{x})}{p'(y_j|\mathbf{x})} \right) - \log\left( \frac{p(\tilde{y}|\mathbf{x})}{p'(\tilde{y}|\mathbf{x})} \right) \right)_j \right\|_2 = 0, \ \forall \{y_j\}_{j=1}^m \subseteq \mathcal{Y}, \tilde{y} \in \mathcal{Y} \tag{124}$$

for $\mathbf{x}$ a.e. which is the same as in Eq. (109). By the same steps as in the other proof we get that $p = p'$ a.e..

W.r.t. ii) Eq. (9) is symmetric, since the norm is symmetric.

W.r.t. iii): Assume we have three models with sets of distributions, $p, p', p''$ and logits $\mathbf{u}(\mathbf{x}), \mathbf{u}'(\mathbf{x}), \mathbf{u}''(\mathbf{x})$ and let us consider $d_{\text{logit}}(p, p'')$. We see that

$$\begin{aligned} d_{\text{logit}}^2(p, p'') &= \mathbb{E}_{\mathbf{x} \sim p_{\mathbf{x}}} ||\mathbf{u}(\mathbf{x}) - \mathbf{u}''(\mathbf{x})||_2^2 = \mathbb{E}_{\mathbf{x} \sim p_{\mathbf{x}}} \left( (\mathbf{u}(\mathbf{x}) - \mathbf{u}'(\mathbf{x})) + (\mathbf{u}'(\mathbf{x}) - \mathbf{u}''(\mathbf{x})) \right)^2 \\ &= \mathbb{E}_{\mathbf{x} \sim p_{\mathbf{x}}} \left( \mathbf{u}(\mathbf{x}) - \mathbf{u}'(\mathbf{x}) \right)^2 + 2\mathbb{E}_{\mathbf{x} \sim p_{\mathbf{x}}} \left( \mathbf{u}(\mathbf{x}) - \mathbf{u}'(\mathbf{x}) \right) \cdot \left( \mathbf{u}'(\mathbf{x}) - \mathbf{u}''(\mathbf{x}) \right) + \mathbb{E}_{\mathbf{x} \sim p_{\mathbf{x}}} \left( (\mathbf{u}'(\mathbf{x}) - \mathbf{u}''(\mathbf{x}))^2 \right). \end{aligned} \tag{125}$$

Considering the mixed term, we see that by applying the Cauchy Schwarz inequality first to the inner product on $\mathbb{R}^k$ and then to $\mathbb{E}_{\mathbf{x} \sim p_{\mathbf{x}}}(\cdot)$ we find

$$\begin{aligned} \mathbb{E}_{\mathbf{x} \sim p_{\mathbf{x}}} \left( \mathbf{u}(\mathbf{x}) - \mathbf{u}'(\mathbf{x}) \right) \cdot \left( \mathbf{u}'(\mathbf{x}) - \mathbf{u}''(\mathbf{x}) \right) &\leq \mathbb{E}_{\mathbf{x} \sim p_{\mathbf{x}}} \left( ||\mathbf{u}(\mathbf{x}) - \mathbf{u}'(\mathbf{x})||_2 \cdot ||\mathbf{u}'(\mathbf{x}) - \mathbf{u}''(\mathbf{x})||_2 \right) \\ &\leq \left( \mathbb{E}_{\mathbf{x} \sim p_{\mathbf{x}}} ||\mathbf{u}(\mathbf{x}) - \mathbf{u}'(\mathbf{x})||_2^2 \right)^{1/2} \cdot \left( \mathbb{E}_{\mathbf{x} \sim p_{\mathbf{x}}} ||\mathbf{u}'(\mathbf{x}) - \mathbf{u}''(\mathbf{x})||_2^2 \right)^{1/2}. \end{aligned} \tag{126}$$

Combining Eq. (125) and Eq. (126), we get

$$d_{\text{logit}}^2(p, p'') \leq \mathbb{E}_{\mathbf{x} \sim p_{\mathbf{x}}} \left( \mathbf{u}(\mathbf{x}) - \mathbf{u}'(\mathbf{x}) \right)^2 + \mathbb{E}_{\mathbf{x} \sim p_{\mathbf{x}}} \left( (\mathbf{u}'(\mathbf{x}) - \mathbf{u}''(\mathbf{x}))^2 \right) \tag{127}$$

$$+ \left( \mathbb{E}_{\mathbf{x} \sim p_{\mathbf{x}}} ||\mathbf{u}(\mathbf{x}) - \mathbf{u}'(\mathbf{x})||_2^2 \right)^{1/2} \cdot \left( \mathbb{E}_{\mathbf{x} \sim p_{\mathbf{x}}} ||\mathbf{u}'(\mathbf{x}) - \mathbf{u}''(\mathbf{x})||_2^2 \right)^{1/2} \tag{128}$$

$$= d_{\text{logit}}^2(p, p') + d_{\text{logit}}^2(p', p'') + 2d_{\text{logit}}(p, p')d_{\text{logit}}(p', p'') \tag{129}$$

$$= (d_{\text{logit}}(p, p') + d_{\text{logit}}(p', p''))^2 \tag{130}$$

Thus, the triangle inequality is satisfied. $\qquad\square$

## E. Theoretical Material for Section 3.1: Bounding logit difference by the KL divergence

### E.1. Assumption 3.2 guarantees that embeddings and unembeddings are bounded

We observe that requiring that a model $(\mathbf{f}, \mathbf{g}) \in \Theta$ is $\tau$-lower bounded (Asm. 3.2) is equivalent to ask that embeddings and unembeddings are bounded. Note that the unembeddings are always bounded when the number of labels is finite. We first show that bounded $\mathbf{f}$ and $\mathbf{g}$ implies lower bounded probability. For this, notice that:

$$p_{\mathbf{f},\mathbf{g}}(y \mid \mathbf{x}) = \frac{\exp(\mathbf{f}(\mathbf{x})^\top \mathbf{g}(y))}{\sum_{y' \in \mathcal{Y}} \exp(\mathbf{f}(\mathbf{x})^\top \mathbf{g}(y'))} \tag{131}$$

$$\geq \frac{\exp\left( -||\mathbf{f}(\mathbf{x})|| \cdot \max_{y \in \mathcal{Y}} ||\mathbf{g}(y)|| \right)}{k \exp\left( ||\mathbf{f}(\mathbf{x})|| \cdot \max_{y \in \mathcal{Y}} ||\mathbf{g}(y)|| \right)} \tag{132}$$

$$= \min_{y \in \mathcal{Y}} \frac{\exp\left( -2||\mathbf{f}(\mathbf{x})|| \cdot ||\mathbf{g}(y)|| \right)}{k} \tag{133}$$

$$\geq \operatorname*{ess\,inf}_{\mathbf{x} \in \text{supp}(p_{\mathbf{x}})} \min_{y \in \mathcal{Y}} \frac{\exp\left( -2||\mathbf{f}(\mathbf{x})|| \cdot ||\mathbf{g}(y)|| \right)}{k}. \tag{134}$$

The last quantity is finite for bounded $\mathbf{f}$ and $\mathbf{g}$ and Asm. 3.2 holds for

$$\tau = \operatorname*{ess\,inf}_{\mathbf{x} \in \text{supp}(p_{\mathbf{x}})} \min_{y \in \mathcal{Y}} p(y \mid \mathbf{x}). \tag{135}$$

On the contrary, if embeddings or unembeddings are not bounded, Eq. (13) cannot be satisfied for $\tau > 0$. Since we only consider a finite number of labels we focus on unbounded embeddings. Assume the embeddings are unbounded. Then there is a sequence $\mathbf{x}_i$ so that $\|\mathbf{f}(\mathbf{x}_i)\| \to \infty$. By the diversity assumption we find that

$$\inf_{\mathbf{v} \in S^{m-1}} \max_{y, \bar{y} \in \mathcal{Y}} (\mathbf{g}(y) - \mathbf{g}(\bar{y})) \, \mathbf{v} \geq \alpha > 0 \tag{136}$$

for some $\alpha$. Indeed, the left hand side is a continuous function on a compact domain and pointwise positive. Thus we find a sequence $y_i$, $\bar{y}_i$ such that

$$\mathbf{f}(\mathbf{x}_i) \cdot (\mathbf{g}(y_i) - \mathbf{g}(\bar{y}_i)) = \mathbf{f}(\mathbf{x}_i) \cdot \max_{y, \bar{y} \in \mathcal{Y}} (\mathbf{g}(y) - \mathbf{g}(\bar{y})) \geq \alpha \|\mathbf{f}(\mathbf{x}_i)\|. \tag{137}$$

This implies that

$$p_{\mathbf{f},\mathbf{g}}(\bar{y}_i | \mathbf{x}_i) = p_{\mathbf{f},\mathbf{g}}(y_i | \mathbf{x}_i) e^{-\mathbf{f}(\mathbf{x}_i)^\top (\mathbf{g}(y_i) - \mathbf{g}(\bar{y}_i))} \leq 1 \cdot e^{-\alpha \|\mathbf{f}(\mathbf{x}_i)\|} \to 0 \tag{138}$$

as $i \to \infty$. Therefore, probabilities cannot be $\tau$-bounded for any $\tau > 0$.

### E.2. Bounding logit distance with KL divergence

In this section, we start by considering two generic probability distributions $\mathbf{p} = (p_1, \ldots, p_d)$ and $\mathbf{q} = (q_1, \ldots, q_d)$ with $d$ outcomes, such that $p_i, q_i \geq 0$. We show how to provide bounds for the logit differences $\sum_i (\log(p_i) - \log(q_i))^2$ in terms of the KL-divergence $\sum_i p_i \log(p_i/q_i)$ While these bounds are generally standard, we could not locate the required bounds in the literature and therefore provide complete proofs here.

Let us first clarify that this is not generally possible. We consider the two probability distributions with two outcomes ($d = 2$) whose probabilities are given by $(p_1, p_2) = (\tau, 1 - \tau)$ and $(q_1, q_2) = (\tau^2, 1 - \tau^2)$ for some $0 \leq \tau < 1$. Then, on one hand, we find

$$\sum_i (\log(p_i) - \log(q_i))^2 \geq \log(\tau)^2 \tag{139}$$

and on the other hand

$$\mathrm{KL}(\mathbf{p}\|\mathbf{q}) = \tau \log(\tau/\tau^2) + (1 - \tau) \log((1 - \tau)/(1 - \tau^2)) \leq \tau |\log(\tau)|. \tag{140}$$

Since the former diverges as $\tau \to 0$ while the latter converges to 0 it is generally not possible to bound the squared logit difference by a function of the KL-divergence only. This is also the root of the observation that small KL-divergence between models in general does not imply similar representations (Nielsen et al., 2025).

We therefore consider assuming a lower bound on $p_i$ for such a bound. We first assume in addition that $q_i$ is also lower bounded. Before stating the bound in this case we provide one auxiliary lemma that we will need in the proof.

**Lemma E.1.** *For $\tau \leq 1/3$ the following bound holds for all $x \geq \tau$*

$$\log(x)^2 \leq 4 \log(\tau)^2 (x \log(x) - x + 1). \tag{141}$$

*Proof.* We define the function $\Phi : \mathbb{R}_+ \setminus \{1\} \to \mathbb{R}$ given by

$$\Phi(x) = \frac{\log(x)^2}{x \log(x) - x + 1}. \tag{142}$$

We observe that the function at thet denominator $x \to x \log(x) - x + 1$ is convex, minimized at $x = 1$, and non-negative on $\mathbb{R}_+$ and thus $\Phi$ is well defined. We notice that $\Phi$ can be extended to a continuous function at 1 by setting $\Phi(1) = 2$. To prove the lemma we need to show that $\Phi(x) \leq 4 \log(\tau)^2$ for $x \geq \tau$. We will proceed in two steps: First we show that $\Phi$ is non-increasing and then we bound $\Phi(\tau)$. To show that $\Phi$ is non-increasing we will need the following auxiliary fact: The function $\Psi : \mathbb{R}_+ \to \mathbb{R}$ given by

$$\Psi(x) = \log(x) - 1 + \frac{1}{x} - \frac{\log(x)^2}{2} \tag{143}$$

is decreasing on $(0, \infty)$ and satisfies $\Psi(1) = 0$, so in particular $\Psi(x) \geq 0$ for $x \leq 1$ and $\Psi(x) \leq 0$ for $x \geq 1$. To show this we note that

$$\Psi'(x) = \frac{1}{x} - \frac{1}{x^2} - \frac{\log(x)}{x} = -\frac{1}{x^2}(x \log(x) - x + 1) \leq 0. \tag{144}$$

Now we calculate $\Phi'$ using $(\log(x)x - x + 1)' = \log(x)$ and find

$$
\begin{aligned}
\Phi'(x) &= \frac{2\frac{\log(x)}{x}(x \log(x) - x + 1) - \log(x)^2 \cdot \log(x)}{(x \log(x) - x + 1)^2} = \frac{2}{(x \log(x) - x + 1)^2} \log(x) \left( \log(x) - 1 + \frac{1}{x} - \frac{\log(x)^2}{2} \right) \\
&= \frac{2}{(x \log(x) - x + 1)^2} \log(x) \Psi(x) \leq 0
\end{aligned}
\tag{145}
$$

where we used that $\log(x)$ and $\Psi(x)$ both change their sign at $x = 1$ so the product is always non-positive. Therefore we conclude that

$$\sup_{x \in [\tau, \tau^{-1}] \setminus \{1\}} \Phi(x) = \Phi(\tau) = \frac{\log(\tau)^2}{\tau \log(\tau) - \tau + 1} \leq 4 \log(\tau)^2 \tag{146}$$

where we used $\tau \log(\tau) - \tau + 1 \geq 1/3 \log(1/3) - 1/3 + 1 \geq 1/4$ for $\tau \leq 1/3$ in the last step. $\qquad\square$

We can now state and prove the upper bound.

**Lemma E.2.** *Let $\mathbf{p}$ and $\mathbf{q}$ be two probability distributions satisfying $p_i \geq \tau$ and $q_i \geq \tau$ for all $i$ and some $0 < \tau \leq 1/3$. Then*

$$\sum_i (\log(p_i) - \log(q_i))^2 \leq \frac{4 \log(\tau)^2}{\tau} \mathrm{KL}(\mathbf{p}\|\mathbf{q}). \tag{147}$$

*Remark* E.3. The bound is tight up to the logarithmic terms. Indeed, consider $\mathbf{p} = (2\tau, 1 - 2\tau)$ and $\mathbf{q} = (\tau, 1 - \tau)$. Then the left hand side is bigger than $\log(2)^2$ while the KL-divergence of these two distributions is bounded by $2 \log(2)\tau$, indeed

$$\mathrm{KL}(\mathbf{p}, \mathbf{q}) = 2\tau \ln\left(\frac{2\tau}{\tau}\right) + (1 - 2\tau) \ln\left(\frac{1 - 2\tau}{1 - \tau}\right) \leq 2 \ln(2)\tau. \tag{148}$$

Moreover, the bound remains tight (up to logarithmic terms) even when the symmetric Jensen-Shannon divergence is considered because

$$\mathrm{KL}(\mathbf{q}, \mathbf{p}) = \tau \ln\left(\frac{\tau}{2\tau}\right) + (1 - \tau) \ln\left(\frac{1 - \tau}{1 - 2\tau}\right) \leq -\tau \ln(2) - \ln(1 - 2\tau) \leq (2 - \ln(2))\tau + 4\tau^2 \tag{149}$$

for $\tau \leq 1/4$ so that the Jensen-Shannon divergence is of order $\tau$ (for small $\tau$).

*Proof.* We write $p_i = x_i q_i$. By assumption, $p_i = x_i q_i \geq \tau$, which means that $x_i \geq \frac{\tau}{q_i} \geq \tau$. Also, $x_i = \frac{p_i}{q_i} \leq \frac{1}{q_i} \leq \frac{1}{\tau}$. So we have $x_i \in [\tau, \tau^{-1}]$. The starting point is to rewrite the KL-divergence as follows

$$\mathrm{KL}(\mathbf{p}\|\mathbf{q}) = \sum_i p_i \log(p_i/q_i) = \sum_i q_i(x_i \log(x_i) - x_i + 1). \tag{150}$$

Here we used in the last step that $\sum_i q_i x_i = \sum_i p_i = 1 = \sum_i q_i$. Recall that the function $x \to x \log(x) - x + 1$ is convex, minimized at $x = 1$, and non-negative. We apply (141) from Lemma E.1 to control

$$
\begin{aligned}
\sum_i (\log(p_i) - \log(q_i))^2 &= \sum_i \log(x_i)^2 \leq 4 \log(\tau)^2 \sum_i (x_i \log(x_i) - x_i + 1) \\
&\leq 4 \log(\tau)^2 \sum_i \frac{q_i}{\tau}(x_i \log(x_i) - x_i + 1) = \frac{4 \log(\tau)^2}{\tau} \mathrm{KL}(\mathbf{p}\|\mathbf{q}).
\end{aligned}
\tag{151}
$$

$\qquad\square$

We can similarly show a slightly weaker bound without assuming that $\mathbf{q}$ is lower bounded.

**Lemma E.4.** *Let $\mathbf{p}$ and $\mathbf{q}$ be two probability distributions satisfying $p_i \geq \tau$ for all $i$ and some $0 < \tau \leq 1/3$. Then*

$$\sum_i (\log(p_i) - \log(q_i))^2 \leq \frac{12 \log(\tau)^2}{\tau} \mathrm{KL}(\mathbf{p}||\mathbf{q}) + \frac{9}{\tau^2} \mathrm{KL}(\mathbf{p}||\mathbf{q})^2. \tag{152}$$

*Remark* E.5. The second term can not be avoided. Note that for $\mathbf{p} = (\tau, 1 - \tau)$ and $\mathbf{q} = (\varepsilon, 1 - \varepsilon)$ we find

$$\lim_{\varepsilon \to 0} \frac{\sum_i (\log(p_i) - \log(q_i))^2}{\log(\varepsilon)^2} = 1 \tag{153}$$

while

$$\lim_{\varepsilon \to 0} \frac{\mathrm{KL}(\mathbf{p}, \mathbf{q})}{|\log(\varepsilon)|} = \tau. \tag{154}$$

*Proof.* As before we write $p_i = x_i q_i$ but in contrast to before $x_i$ is not upper bounded but only satisfies $x_i \geq p_i \geq \tau$. The main idea is to control $\log(x_i)^2$ separately, when $\mathbf{x}$ is large we use a new bound based on a similar approach while we can apply the approach from before when $\mathbf{x}$ is small because this lower bounds $q_i$. We start with the former. Note that for $x \geq 3$ we find

$$\frac{\log(x)x}{3} \leq x \log(x) - x + 1. \tag{155}$$

Indeed, note that for $x = 3$

$$\frac{2}{3} x \log(x) - x + 1 = \frac{2}{3} \cdot 3 \cdot \log(3) - 3 + 1 > 0 \tag{156}$$

and

$$\left( \frac{2}{3} x \log(x) - x + 1 \right)' = \frac{2}{3} \log(x) + \frac{2}{3} - 1 \geq \frac{1}{3} \tag{157}$$

for $x \geq 3$. This implies that

$$\begin{aligned}
\sum_{i:\,x_i > 3} \log(x_i)_i^2 &\leq \sum_{i:x_i > 3} \left( \frac{3}{x_i} \frac{x_i \log(x_i)}{3} \right)^2 \\
&\leq \sum_{i:x_i > 3} \left( \frac{3p_i}{\tau x_i} (x_i \log(x_i) - x_i + 1) \right)^2 \\
&\leq \frac{9}{\tau^2} \sum_i (q_i(x_i \log(x_i) - x_i + 1))^2 \\
&\leq \frac{9}{\tau^2} \left( \sum_i q_i(x_i \log(x_i) - x_i + 1) \right)^2 = \frac{9}{\tau^2} \mathrm{KL}(\mathbf{p}||\mathbf{q})^2.
\end{aligned} \tag{158}$$

Here we used in the last step that all terms in the sum are non-negative which implies that all cross terms in the expansion of the square of the sum are non-negative.

We now consider the indices $i$ so that $x_i < 3$. Note that $q_i = p_i/x_i \geq \tau/3$. Therefore we obtain using $x_i \geq \tau$ and (141)

$$\begin{aligned}
\sum_{i:x_i \leq 3} \log(\mathbf{x})_i^2 &\leq 4 \log(\tau)^2 \sum_{i:x_i \leq 3} (x_i \log(x_i) - x_i + 1) \\
&\leq 4 \log(\tau)^2 \sum_{i:x_i \leq 3} \frac{q_i}{\tau/3} (x_i \log(x_i) - x_i + 1) \\
&\leq \frac{12 \log(\tau)^2}{\tau} \sum_i q_i (x_i \log(x_i) - x_i + 1) = \frac{12 \log(\tau)^2}{\tau} \mathrm{KL}(\mathbf{p}||\mathbf{q}).
\end{aligned} \tag{159}$$

Combining (158) and (159) ends the proof. $\qquad\square$

From these results it is straightforward to relate $\mathbf{f}(\mathbf{x})^\top \mathbf{g}(y_i)$ and $\mathbf{f}'(\mathbf{x})^\top \mathbf{g}'(y_i)$.

**Theorem E.6.** *For two models* $(\mathbf{f}, \mathbf{g}), (\mathbf{f}', \mathbf{g}') \in \Theta$, *denote with* $\mathbf{u}(\mathbf{x})$ *and* $\mathbf{u}'(\mathbf{x})$ *the model logits* $\mathbf{u}_i(\mathbf{x}) = \mathbf{f}(\mathbf{x})^\top \mathbf{g}(y_i)$ *and* $(\mathbf{u}'(\mathbf{x}))_i = \mathbf{f}'(\mathbf{x})^\top \mathbf{g}'(y_i)$. *Fix a point* $\mathbf{x}$. *Assume that* $\min_{y \in \mathcal{Y}} p_{\mathbf{f}, \mathbf{g}}(y|\mathbf{x}) \geq \tau$. *Then*

$$\|\mathbf{u}(\mathbf{x}) - \mathbf{u}'(\mathbf{x})\|^2 \leq \frac{12 \log(\tau)^2}{\tau} \mathrm{KL}(p_{\mathbf{f}, \mathbf{g}}(\cdot|\mathbf{x}) \| p_{\mathbf{f}', \mathbf{g}'}(\cdot|\mathbf{x})) + \frac{9}{\tau^2} \mathrm{KL}(p_{\mathbf{f}, \mathbf{g}}(\cdot|\mathbf{x}) \| p_{\mathbf{f}', \mathbf{g}'}(\cdot|\mathbf{x}))^2. \tag{160}$$

*If in addition* $\min_{y \in \mathcal{Y}} p_{\mathbf{f}', \mathbf{g}'}(y|\mathbf{x}) \geq \tau$ *then*

$$\|\mathbf{u}(\mathbf{x}) - \mathbf{u}'(\mathbf{x})\|^2 \leq \frac{4 \log(\tau)^2}{\tau} \mathrm{KL}(p_{\mathbf{f}, \mathbf{g}}(\cdot|\mathbf{x}) \| p_{\mathbf{f}', \mathbf{g}'}(\cdot|\mathbf{x})). \tag{161}$$

*Proof.* We note that by (1)

$$\mathbf{f}(\mathbf{x})^\top \mathbf{g}(y) = \log\left(p_{\mathbf{f}, \mathbf{g}}(y \mid \mathbf{x})\right) + \log\left(\sum_{y' \in \mathcal{Y}} e^{\mathbf{f}(\mathbf{x})^\top \mathbf{g}(y')}\right) \tag{162}$$

Using that $\mathbf{g}$ is centred we find

$$0 = \frac{1}{k} \sum_{y' \in \mathcal{Y}} \mathbf{f}(\mathbf{x})^\top \mathbf{g}(y) = \frac{1}{k} \sum_{y \in \mathcal{Y}} \left(\log\left(p_{\mathbf{f}, \mathbf{g}}(y \mid \mathbf{x})\right) + \log\left(\sum_{y' \in \mathcal{Y}} e^{\mathbf{f}(\mathbf{x})^\top \mathbf{g}(y')}\right)\right) = \overline{\log\left(p_{\mathbf{f}, \mathbf{g}}(\cdot \mid \mathbf{x})\right)} + \log\left(\sum_{y' \in \mathcal{Y}} e^{\mathbf{f}(\mathbf{x})^\top \mathbf{g}(y')}\right). \tag{163}$$

Where $\overline{\log\left(p_{\mathbf{f}, \mathbf{g}}(\cdot \mid \mathbf{x})\right)}$ denote the mean over $y \in \mathcal{Y}$. We infer that

$$\mathbf{f}(\mathbf{x})^\top \mathbf{g}(y) = \log\left(p_{\mathbf{f}, \mathbf{g}}(y \mid \mathbf{x})\right) - \overline{\log\left(p_{\mathbf{f}, \mathbf{g}}(\cdot \mid \mathbf{x})\right)}. \tag{164}$$

Now we can conclude that

$$\begin{aligned}
\|\mathbf{u}(\mathbf{x}) - \mathbf{u}'(\mathbf{x})\|^2 &= \sum_{y \in \mathcal{Y}} (\mathbf{f}(\mathbf{x})^\top \mathbf{g}(y) - \mathbf{f}'(\mathbf{x})^\top \mathbf{g}'(y))^2 \\
&= \sum_{y \in \mathcal{Y}} \left(\log\left(p_{\mathbf{f}, \mathbf{g}}(y \mid \mathbf{x})\right) - \overline{\log\left(p_{\mathbf{f}, \mathbf{g}}(\cdot \mid \mathbf{x})\right)} - \log\left(p_{\mathbf{f}', \mathbf{g}'}(y \mid \mathbf{x})\right) + \overline{\log\left(p_{\mathbf{f}', \mathbf{g}'}(\cdot \mid \mathbf{x})\right)}\right)^2 \\
&= \sum_{y \in \mathcal{Y}} \left(\log\left(p_{\mathbf{f}, \mathbf{g}}(y \mid \mathbf{x})\right) - \log\left(p_{\mathbf{f}', \mathbf{g}'}(y \mid \mathbf{x})\right)\right)^2 + |\mathcal{Y}| \left(\overline{\log\left(p_{\mathbf{f}', \mathbf{g}'}(\cdot \mid \mathbf{x})\right)} - \overline{\log\left(p_{\mathbf{f}, \mathbf{g}}(\cdot \mid \mathbf{x})\right)}\right)^2 \\
&\quad + 2 \left(\overline{\log\left(p_{\mathbf{f}', \mathbf{g}'}(\cdot \mid \mathbf{x})\right)} - \overline{\log\left(p_{\mathbf{f}, \mathbf{g}}(\cdot \mid \mathbf{x})\right)}\right) \sum_{y \in \mathcal{Y}} \left(\log\left(p_{\mathbf{f}, \mathbf{g}}(y \mid \mathbf{x})\right) - \log\left(p_{\mathbf{f}', \mathbf{g}'}(y \mid \mathbf{x})\right)\right) \\
&= \sum_{y \in \mathcal{Y}} \left(\log\left(p_{\mathbf{f}, \mathbf{g}}(y \mid \mathbf{x})\right) - \log\left(p_{\mathbf{f}', \mathbf{g}'}(y \mid \mathbf{x})\right)\right)^2 + |\mathcal{Y}| \left(\overline{\log\left(p_{\mathbf{f}', \mathbf{g}'}(\cdot \mid \mathbf{x})\right)} - \overline{\log\left(p_{\mathbf{f}, \mathbf{g}}(\cdot \mid \mathbf{x})\right)}\right)^2 \\
&\quad - 2|\mathcal{Y}| \left(\overline{\log\left(p_{\mathbf{f}', \mathbf{g}'}(\cdot \mid \mathbf{x})\right)} - \overline{\log\left(p_{\mathbf{f}, \mathbf{g}}(\cdot \mid \mathbf{x})\right)}\right)^2 \\
&\leq \sum_{y \in \mathcal{Y}} \left(\log\left(p_{\mathbf{f}, \mathbf{g}}(y \mid \mathbf{x})\right) - \log\left(p_{\mathbf{f}', \mathbf{g}'}(y \mid \mathbf{x})\right)\right)^2.
\end{aligned} \tag{165}$$

The last steps mirror the computation for showing that the squared norm is larger than the variance of a random variable. Now we can apply Lemma E.4 to conclude (160) and Lemma E.2 to get (161). $\quad\square$

We can integrate the previous result over $\mathbf{x}$ to relate the mean logit difference to the cross entropy loss.

**Corollary E.7.** *For two models* $(\mathbf{f}, \mathbf{g}), (\mathbf{f}', \mathbf{g}') \in \Theta$, *denote with* $\mathbf{u}(\mathbf{x})$ *and* $\mathbf{u}'(\mathbf{x})$ *the model logits* $u(\mathbf{x})_i = \mathbf{f}(\mathbf{x})^\top \mathbf{g}(y_i)$ *and* $u'(\mathbf{x})_i = \mathbf{f}'(\mathbf{x})^\top \mathbf{g}'(y_i)$. *Assume that for* $p_{\mathbf{x}}$ *almost every* $\mathbf{x} \in \mathcal{X}$

$$\min_{y \in \mathcal{Y}} p_{\mathbf{f}, \mathbf{g}}(y|\mathbf{x}) \geq \tau, \quad \min_{y \in \mathcal{Y}} p_{\mathbf{f}', \mathbf{g}'}(y|\mathbf{x}) \geq \tau. \tag{166}$$

*Then*

$$\mathbb{E}_{\mathbf{x} \sim p_{\mathbf{x}}} \|\mathbf{u}(\mathbf{x}) - \mathbf{u}'(\mathbf{x})\|^2 \leq \frac{4 \log(\tau)^2}{\tau} d_{\mathrm{KL}}(p_{\mathbf{f},\mathbf{g}}, p_{\mathbf{f}',\mathbf{g}'}). \tag{167}$$

*Proof.* The result follows by integrating (161) using relation (10). □

*Remark* E.8. Without assuming a lower bound on $p_{\mathbf{f}',\mathbf{g}'}(\cdot|\mathbf{x})$ we cannot derive a similar bound as $\mathbb{E}(\mathrm{KL}(p_{\mathbf{f},\mathbf{g}}(\cdot|\mathbf{x})\|p_{\mathbf{f}',\mathbf{g}'}(\cdot|\mathbf{x}))^2)$ cannot be bounded even when $\mathrm{KL}(p_{\mathbf{f},\mathbf{g}}(\cdot|\mathbf{x})\|p_{\mathbf{f}',\mathbf{g}'}(\cdot|\mathbf{x}))$ is small.

**Lemma E.9.** *For two distributions $p, p'$, with (finite) logits $\mathbf{u}(\mathbf{x}), \mathbf{u}'(\mathbf{x}) \in \mathbb{R}^k$, respectively, we have*

$$\|\mathbf{u}(\mathbf{x}) - \mathbf{u}'(\mathbf{x})\|^2 \geq 2\mathrm{KL}(p(\cdot \mid \mathbf{x}), p'(\cdot \mid \mathbf{x})). \tag{168}$$

*Proof.* For any distribution $p(y \mid \mathbf{x})$, the logits are given by considering

$$p(y_i \mid \mathbf{x}) = \frac{\exp u(\mathbf{x})_i}{Z(\mathbf{x})} \tag{169}$$

where $Z(\mathbf{u}(\mathbf{x})) := \sum_{i=1}^{k} \exp(u(\mathbf{x})_i)$. The KL divergence for each element $\mathbf{x} \in \mathcal{X}$ can be written as:

$$KL(p(\cdot \mid \mathbf{x})\|p'(\cdot \mid \mathbf{x})) = \sum_{i=1}^{k} p(y_i \mid \mathbf{x}) \log \frac{p(y_i \mid \mathbf{x})}{p'(y_i \mid \mathbf{x})} \tag{170}$$

$$= \sum_{i=1}^{k} \frac{e^{u_i(\mathbf{x})}}{Z(\mathbf{u}(\mathbf{x}))}(u_i(\mathbf{x}) - u_i'(\mathbf{x})) + \log \frac{Z(\mathbf{u}'(\mathbf{x}))}{Z(\mathbf{u}(\mathbf{x}))} \tag{171}$$

$$= (\nabla_{\mathbf{u}} \log Z(\mathbf{u}(\mathbf{x})))^{\top}(\mathbf{u}(\mathbf{x}) - \mathbf{u}'(\mathbf{x})) + \log Z(\mathbf{u}'(\mathbf{x})) - \log Z(\mathbf{u}(\mathbf{x})). \tag{172}$$

where $\nabla_{\mathbf{u}}$ is the gradient over the logits. Notice that, by setting $F(\cdot) = \log Z(\cdot)$, the KL divergence coincides with the Bregman divergence $D_F(\mathbf{u}'(\mathbf{x}), \mathbf{u}(\mathbf{x}))$. Denote with $\mathbf{p} = \mathrm{softmax}(\mathbf{u})$. Now, considering the Hessian of $\log Z$, which is the Fisher information matrix. We get that:

$$\nabla_{\mathbf{u}}^2 \log Z(\mathbf{u}) = \mathrm{diag}(\mathbf{p}) - \mathbf{p}\mathbf{p}^{\top}. \tag{173}$$

From this, we observe that $\nabla^2 \log Z(\mathbf{u}) \preceq \mathbf{I}$.[6] Equivalently, this implies that the $F$ is 1-smooth (Nesterov, 2013, Theorem 2.1.6) . This means that, for any $\mathbf{u}, \mathbf{u}' \in \mathbb{R}^k$, we can write

$$\|\nabla_{\mathbf{u}} \log Z(\mathbf{u}) - \nabla_{\mathbf{u}} \log Z(\mathbf{u}')\|^2 \leq 1\|\mathbf{u} - \mathbf{u}'\|^2, \tag{174}$$

or equivalently

$$\log Z(\mathbf{u}') \leq \log Z(\mathbf{u}) - \nabla(\log Z(\mathbf{u}))^{\top}(\mathbf{u} - \mathbf{u}') + \frac{1}{2}\|\mathbf{u} - \mathbf{u}'\|^2. \tag{175}$$

Substituting this to Eq. (172), we get:

$$\|\mathbf{u}(\mathbf{x}) - \mathbf{u}'(\mathbf{x})\|^2 \geq 2\mathrm{KL}(p(\cdot \mid \mathbf{x})\|p'(\cdot \mid \mathbf{x})), \tag{176}$$

which gives the result. □

## F. Bounding Representation Similarity in Terms of Canonical Correlations

We now investigate the implications for representation similarity. Recall that we consider two models specified by $(\mathbf{f}, \mathbf{g}), (\mathbf{f}', \mathbf{g}') \in \Theta$ with densities $p_{\mathbf{f},\mathbf{g}}$ and $p_{\mathbf{f}',\mathbf{g}'}$. Their representational dimensions are $m$ and $m'$ and in particular we do not assume that they agree. We denote by $\mathbf{L}$ and $\mathbf{L}'$ the matrices of unembeddings, that is

$$\mathbf{L} = \begin{pmatrix} \mathbf{g}(y_1) & \dots & \mathbf{g}(y_k) \end{pmatrix} \in \mathbb{R}^{m \times k} \quad \text{and} \quad \mathbf{L}' = \begin{pmatrix} \mathbf{g}'(y_1) & \cdots & \mathbf{g}'(y_k) \end{pmatrix} \in \mathbb{R}^{m' \times k}. \tag{177}$$

---

[6]Recall that a matrix $\mathbf{A} \preceq \mathbf{B}$ if and only if $\mathbf{B} - \mathbf{A}$ is positive semi-definite, i.e., $\mathbf{x}^{\top}(\mathbf{B} - \mathbf{A})\mathbf{x} \geq 0$ for all $\mathbf{x}$.

Note that the relations

$$\mathbf{u}(\mathbf{x}) = \mathbf{L}^\top \mathbf{f}(\mathbf{x}), \quad \mathbf{u}'(\mathbf{x}) = \mathbf{L}'^\top \mathbf{f}'(\mathbf{x}) \tag{178}$$

hold. We have seen in Theorem E.6 how to bound

$$\|\mathbf{u}(\mathbf{x}) - \mathbf{u}'(\mathbf{x})\|^2 = \|\mathbf{L}^\top \mathbf{f}(\mathbf{x}) - \mathbf{L}'^\top \mathbf{f}'(\mathbf{x})\|^2 \tag{179}$$

in terms of the KL-divergence $\mathrm{KL}(p_{\mathbf{f},\mathbf{g}}(\cdot|\mathbf{x})\|p_{\mathbf{f}',\mathbf{g}'}(\cdot|\mathbf{x}))$. The goal of this section is to bound the canonical correlation coefficients between the random variables with samples $(\mathbf{g}(y_i))_{1 \le i \le k}$ and $(\mathbf{g}'(y_i))_{1 \le i \le k}$ and the random variables $\mathbf{f}$ and $\mathbf{f}'$ by the expectation of the square logit difference. We denote by $\rho_1 \ge \rho_2 \ge \ldots \ge \rho_{\min(m,m')}$ the canonical correlations of $\mathbf{g}$ and $\mathbf{g}'$ which are given by the singular values of

$$(\mathbf{L}\mathbf{L}^\top)^{-1/2}(\mathbf{L}\mathbf{L}'^\top)(\mathbf{L}'\mathbf{L}'^\top)^{-1/2}. \tag{180}$$

Indeed, this holds because $\mathbf{L}\mathbf{L}^\top = k\Sigma_{\mathbf{g},\mathbf{g}} \in \mathbb{R}^{m \times m}$, $\mathbf{L}'\mathbf{L}'^\top = k\Sigma_{\mathbf{g}',\mathbf{g}'} \in \mathbb{R}^{m' \times m'}$, and $\mathbf{L}\mathbf{L}'^\top = k\Sigma_{\mathbf{g},\mathbf{g}'} \in \mathbb{R}^{m \times m'}$. Since the distribution $p_{\mathbf{f},\mathbf{g}}$ is not invariant to embeddings shifts $\mathbf{f} + \mathbf{c}$ it is natural to consider the moment-based canonical correlations of $\mathbf{f}$ and $\mathbf{f}'$ where instead of considering covariances one considers the second moments directly and we use the notation

$$\mathbf{M}_{\mathbf{f},\mathbf{f}} = \mathbb{E}_{\mathbf{x} \sim p_\mathbf{x}}(\mathbf{f}(\mathbf{x})\mathbf{f}^\top(\mathbf{x})). \tag{181}$$

We denote the moment-based canonical correlations of $\mathbf{f}$ and $\mathbf{f}'$ by $\tilde{\rho}_1 \ge \tilde{\rho}_2 \ge \ldots \ge \tilde{\rho}_{\min(m,m')}$ which are given by the singular values of

$$\mathbf{M}_{\mathbf{f},\mathbf{f}}^{-1/2}\mathbf{M}_{\mathbf{f},\mathbf{f}'}\mathbf{M}_{\mathbf{f}',\mathbf{f}'}^{-1/2}. \tag{182}$$

The result below extends to the standard canonical correlations with minor changes which we discuss in Appendix F.4.

To simplify the analysis slightly, we assume that $\mathbf{L}$ and $\mathbf{L}'$ have full rank $m$ and $m'$ and therefore, in particular, $k \ge \max(m,m') + 1$ (the $+1$ arises because the unembeddings are centred and therefore linearly dependent ($\sum_{y \in \mathcal{Y}} \mathbf{g}(y) = 0$) so the rank can be at most $k - 1$). This implies that $\mathbf{L}\mathbf{L}^\top \in \mathbb{R}^{m \times m}$ and $\mathbf{L}'\mathbf{L}'^\top \in \mathbb{R}^{m' \times m'}$ are invertible matrices. Similarly we assume that the moment matrices $\mathbf{M}_{\mathbf{f},\mathbf{f}}$ and $\mathbf{M}_{\mathbf{f}',\mathbf{f}'}$ defined in (181) have maximal rank $m$ and $m'$ respectively. Note that this follows from the diversity condition in Asm. 2.1 under minor regularity assumptions. We expect that extensions to the rank-deficient case are possible by considering the pseudoinverses (see (Marconato et al., 2025) for a full discussion of the identifiability perspective in this case). Moreover, note that this assumption is not a real restriction because we could equivalently focus on the space generated by the vectors $\mathbf{g}(y_i)$ and the projection of $\mathbf{f}$ on the orthogonal complement of this space is not identifiable anyway and similarly a degenerate $\mathbf{M}_{\mathbf{f},\mathbf{f}}$ results in arbitrary behaviour of $\mathbf{g}$ on the orthogonal complement of the column space of $\mathbf{M}_{\mathbf{f},\mathbf{g}}$. Our upper bounds depend on diversity properties of one of the models $(\mathbf{f}, \mathbf{g})$ (typically the teacher model in a distillation setup) and it turns out that the right object to consider is the logit covariance

$$\mathbf{M}_{\mathbf{u},\mathbf{u}} = \mathbb{E}_{\mathbf{x} \sim p_\mathbf{x}}(\mathbf{u}(\mathbf{x})\mathbf{u}^\top(\mathbf{x})) = \mathbf{L}^\top \mathbf{M}_{\mathbf{f},\mathbf{f}}\mathbf{L} \tag{183}$$

which measures how spread out the logit distribution is. We remark that the logits are invariant to the model symmetries $(\mathbf{f}, \mathbf{g}) \to (\mathbf{A}\mathbf{f}, \mathbf{A}^{-\top}\mathbf{g})$. Note that $\mathbf{L}$ and $\mathbf{M}_{\mathbf{f},\mathbf{f}}$ have maximal rank $m$ and so we denote the $m$-nonzero eigenvalues of $\mathbf{M}_{\mathbf{u},\mathbf{u}}$ by $\mu_1 \ge \mu_2 \ge \ldots \ge \mu_m$. Note that small eigenvalues indicate that the representation does not exploit all dimensions of the embedding space and therefore this spectrum allows to quantify the diversity condition. After these preliminaries we can state the main result of this section.

**Theorem F.1.** *Consider two models $(\mathbf{f}, \mathbf{g})$ and $(\mathbf{f}', \mathbf{g}')$ with representation dimensions $m$ and $m'$ and assume that $\mathbf{M}_{\mathbf{f},\mathbf{f}}$, $\mathbf{L}$ and $\mathbf{M}_{\mathbf{f}',\mathbf{f}'}$, $\mathbf{L}'$ have maximal rank $m$ and $m'$ respectively. The canonical correlations $\rho_i$ (in decreasing order) of $\mathbf{g}$ and $\mathbf{g}'$ satisfy the bound*

$$\sum_{i=1}^m (1 - \rho_i^2)\mu_i \le \mathbb{E}_{\mathbf{x} \sim p_\mathbf{x}}\|\mathbf{u}(\mathbf{x}) - \mathbf{u}'(\mathbf{x})\|^2 \tag{184}$$

*where $\mu_i$ are the eigenvalues of $\mathbf{M}_{\mathbf{u},\mathbf{u}}$ in decreasing order. Similarly the moment-based canonical correlations $\tilde{\rho}_i$ (in decreasing order) of $\mathbf{f}$ and $\mathbf{f}'$ satisfy*

$$\sum_{i=1}^m (1 - \tilde{\rho}_i^2)\mu_i \le \mathbb{E}_{\mathbf{x} \sim p_\mathbf{x}}\|\mathbf{u}(\mathbf{x}) - \mathbf{u}'(\mathbf{x})\|^2 \tag{185}$$

*where we in both cases set $\rho_i = \tilde{\rho}_i = 0$ for $i > m'$ if $m > m'$.*

The goal of the next lemmas is to provide a proof of Theorem F.1, i.e., to show (184) and (185). The final summary of the proof can be found at the end of Appendix F.2. We note that the two bounds are essentially equivalent due to the symmetry of the problem in $\mathbf{f}$ and $\mathbf{g}$, indeed note that the operation $\mathbf{L}\mathbf{L}^\top$ just corresponds (up to a scalar) to building the covariance matrix from a random variables with finitely many values. Nevertheless, we show the two proofs separately below for the convenience of the reader. The general strategy of the proof is to regress $\mathbf{L}$ on $\mathbf{L}'$ and use this to decompose $\mathbf{u}(\mathbf{x}) - \mathbf{u}'(\mathbf{x})$ (see Lemma F.2). Then one of the terms in the decomposition can be related to the canonical correlations $\rho_i$ (see Lemma F.4). Finally this relation can be used to bound the canonical correlation in Lemma F.5. We then extend the results to $\tilde{\rho}$ using the same steps.

## F.1. Bounding similarity of unembeddings

We consider the regression problem

$$\mathbf{B} = \min_{\overline{\mathbf{B}} \in \mathbb{R}^{m \times m'}} \|\mathbf{L} - \overline{\mathbf{B}}\mathbf{L}'\|^2. \tag{186}$$

The minimizer $\mathbf{B}$ of this ordinary least squares problem is given by

$$\mathbf{B} = \mathbf{L}\mathbf{L}'^\top (\mathbf{L}'\mathbf{L}'^\top)^{-1}. \tag{187}$$

We introduce the residual

$$\mathbf{\Delta} = \mathbf{L} - \mathbf{B}\mathbf{L}' \in \mathbb{R}^{m \times k}. \tag{188}$$

We then find the following expansion.

**Lemma F.2.** *With the notation introduced above, in particular* $\mathbf{B} = \mathbf{L}\mathbf{L}'^\top (\mathbf{L}'\mathbf{L}'^\top)^{-1}$ *and* $\mathbf{\Delta} = \mathbf{L} - \mathbf{B}\mathbf{L}'$ *we can decompose*

$$\|\mathbf{u}(\mathbf{x}) - \mathbf{u}'(\mathbf{x})\|^2 = \|\mathbf{L}'^\top (\mathbf{B}^\top \mathbf{f}(\mathbf{x}) - \mathbf{f}'(\mathbf{x}))\|^2 + \|\mathbf{\Delta}^\top \mathbf{f}(\mathbf{x})\|^2. \tag{189}$$

*Remark* F.3. This lemma can be used to bound the norm of the residual $\|\mathbf{\Delta}\|$ of the regression of $\mathbf{g}$ on $\mathbf{g}'$. Indeed, we find that

$$\mathbb{E}_{\mathbf{x} \sim p_{\mathbf{x}}} \|\mathbf{u}(\mathbf{x}) - \mathbf{u}'(\mathbf{x})\|^2 \geq \mathbb{E}_{\mathbf{x} \sim p_{\mathbf{x}}} \|\mathbf{\Delta}^\top \mathbf{f}(\mathbf{x})\|^2 = \mathbb{E}_{\mathbf{x} \sim p_{\mathbf{x}}} \operatorname{Tr}\left(\mathbf{\Delta}^\top \mathbf{f}(\mathbf{x})\mathbf{f}^\top(\mathbf{x})\mathbf{\Delta}\right) = \operatorname{Tr}\left(\mathbf{\Delta}^\top \mathbf{M}_{\mathbf{f},\mathbf{f}}\mathbf{\Delta}\right) \geq \lambda_{\min}(\mathbf{M}_{\mathbf{f},\mathbf{f}})\|\mathbf{\Delta}\|^2. \tag{190}$$

However, this bound is not invariant to model reparametrizations while the canonical correlations are. However, it is sufficient to conclude linear identifiability when the logits agree almost everywhere.

*Proof.* Note that

$$\mathbf{\Delta}\mathbf{L}'^\top = \mathbf{L}\mathbf{L}'^\top - \mathbf{L}\mathbf{L}'^\top (\mathbf{L}'\mathbf{L}'^\top)^{-1}\mathbf{L}'\mathbf{L}'^\top = \mathbf{0}, \tag{191}$$

where $\mathbf{0}$ is an $m \times m'$ matrix of zeros. Then we get

$$\begin{aligned}
\|\mathbf{u}(\mathbf{x}) - \mathbf{u}'(\mathbf{x})\|^2 &= \|\mathbf{L}^\top \mathbf{f}(\mathbf{x}) - \mathbf{L}'^\top \mathbf{f}'(\mathbf{x})\|^2 \\
&= \|\mathbf{\Delta}^\top \mathbf{f}(\mathbf{x}) + \mathbf{L}'^\top \mathbf{B}^\top \mathbf{f}(\mathbf{x}) - \mathbf{L}'^\top \mathbf{f}'(\mathbf{x})\|^2 \\
&= \mathbf{f}(\mathbf{x})^\top \mathbf{\Delta}\mathbf{\Delta}^\top \mathbf{f}(\mathbf{x}) + (\mathbf{B}^\top \mathbf{f}(\mathbf{x}) - \mathbf{f}'(\mathbf{x}))^\top \mathbf{L}'\mathbf{L}'^\top (\mathbf{B}^\top \mathbf{f}(\mathbf{x}) - \mathbf{f}'(\mathbf{x})) \\
&\quad + \mathbf{f}(\mathbf{x})^\top \underbrace{\mathbf{\Delta}\mathbf{L}'^\top}_{=0}(\mathbf{B}^\top \mathbf{f}(\mathbf{x}) - \mathbf{f}'(\mathbf{x})) + (\mathbf{B}^\top \mathbf{f}(\mathbf{x}) - \mathbf{f}'(\mathbf{x}))^\top \underbrace{\mathbf{L}'\mathbf{\Delta}^\top}_{=0}\mathbf{f}(\mathbf{x}) \\
&= \|\mathbf{\Delta}^\top \mathbf{f}(\mathbf{x})\|^2 + \|\mathbf{L}'^\top (\mathbf{B}^\top \mathbf{f}(\mathbf{x}) - \mathbf{f}'(\mathbf{x}))\|^2.
\end{aligned} \tag{192}$$

This completes the proof. $\qquad\square$

As a next step we have to relate the term $\|\mathbf{\Delta}^\top \mathbf{f}(x)\|^2$ to the canonical correlation of $\mathbf{g}(y)$ and $\mathbf{g}'(y)$. We first show the following intermediate result.

**Lemma F.4.** *Denote the eigenvalues of* $(\mathbf{LL}^\top)^{-1/2}\boldsymbol{\Delta}\boldsymbol{\Delta}^\top(\mathbf{LL}^\top)^{-1/2}$ *by* $\lambda_1 \leq \ldots \leq \lambda_m$ *and the squared canonical correlations of* $\mathbf{g}(y_1), \ldots, \mathbf{g}(y_k)$ *and* $\mathbf{g}'(y_1), \ldots, \mathbf{g}'(y_k)$ *by* $\rho_1^2 \geq \ldots \geq \rho_{\min(m,m')}^2$. *Then the relation*

$$\lambda_i = 1 - \rho_i^2 \tag{193}$$

*holds, where we set* $\rho_i = 0$ *for* $m' < i \leq m$ *if* $m' < m$.

*Proof.* We expand

$$
\begin{aligned}
\boldsymbol{\Delta}\boldsymbol{\Delta}^\top &= \mathbf{LL}^\top - \mathbf{LL}'^\top\mathbf{B}^\top - \mathbf{BL}'\mathbf{L}^\top + \mathbf{BL}'\mathbf{L}'^\top\mathbf{B}^\top \\
&= \mathbf{LL}^\top - \mathbf{LL}'^\top(\mathbf{L}'\mathbf{L}'^\top)^{-1}\mathbf{L}'\mathbf{L}^\top - \mathbf{LL}'^\top(\mathbf{L}'\mathbf{L}'^\top)^{-1}\mathbf{L}'\mathbf{L}^\top + \mathbf{LL}'^\top(\mathbf{L}'\mathbf{L}'^\top)^{-1}\mathbf{L}'\mathbf{L}'^\top(\mathbf{L}'\mathbf{L}'^\top)^{-1}\mathbf{L}'\mathbf{L}^\top \\
&= \mathbf{LL}^\top - \mathbf{LL}'^\top(\mathbf{L}'\mathbf{L}'^\top)^{-1}\mathbf{L}'\mathbf{L}^\top.
\end{aligned}
\tag{194}
$$

Therefore

$$
\begin{aligned}
(\mathbf{LL}^\top)^{-1/2}\boldsymbol{\Delta}\boldsymbol{\Delta}^\top(\mathbf{LL}^\top)^{-1/2} &= (\mathbf{LL}^\top)^{-1/2}\mathbf{LL}^\top(\mathbf{LL}^\top)^{-1/2} - (\mathbf{LL}^\top)^{-1/2}\mathbf{LL}'^\top(\mathbf{L}'\mathbf{L}'^\top)^{-1}\mathbf{L}'\mathbf{L}^\top(\mathbf{LL}^\top)^{-1/2} \\
&= \mathbf{1}_{m \times m} - (\mathbf{LL}^\top)^{-1/2}\mathbf{LL}'^\top(\mathbf{L}'\mathbf{L}'^\top)^{-1/2}\left((\mathbf{LL}^\top)^{-1/2}\mathbf{LL}'^\top(\mathbf{L}'\mathbf{L}'^\top)^{-1/2}\right)^\top.
\end{aligned}
\tag{195}
$$

Recall that by (180) the singular values $\rho_1, \ldots, \rho_{\min(m,m')}$ of $(\mathbf{LL}^\top)^{-1/2}\mathbf{LL}'^\top(\mathbf{L}'\mathbf{L}'^\top)^{-1/2}$ are the canonical correlation of $\mathbf{g}$ and $\mathbf{g}'$. Moreover, from (195) (and using that $\mathbf{AA}^\top$ has squared singular values as eigenvalues) we find

$$\lambda_i = 1 - \rho_i^2 \tag{196}$$

where we extend $\rho_i = 0$ for $m' < i \leq m$ if $m \geq m'$. $\square$

**Lemma F.5.** *Recall that the non-zero eigenvalues of the second moment of the logits* $\mathbb{E}_{\mathbf{x} \sim p_\mathbf{x}}(\mathbf{u}(\mathbf{x})\mathbf{u}^\top(\mathbf{x})) \in \mathbb{R}^{k \times k}$ *are denoted by* $\mu_1 \geq \mu_2 \geq \ldots \geq \mu_m$ *and the canonical correlations of* $\mathbf{g}$ *and* $\mathbf{g}'$ *by* $\rho_1 \geq \rho_2 \geq \ldots \geq \rho_{\min(m,m')}$. *Then the following bound holds*

$$\sum_{i=1}^m (1 - \rho_i^2)\mu_i \leq \mathbb{E}_{\mathbf{x} \sim p_\mathbf{x}}\|\boldsymbol{\Delta}^\top\mathbf{f}(\mathbf{x})\|^2. \tag{197}$$

*Proof.* We rewrite using cyclicity of the trace

$$
\begin{aligned}
\|\boldsymbol{\Delta}^\top\mathbf{f}(\mathbf{x})\|^2 &= \mathrm{Tr}\left(\boldsymbol{\Delta}^\top(\mathbf{LL}^\top)^{-1/2}(\mathbf{LL}^\top)^{1/2}\mathbf{f}(\mathbf{x})\mathbf{f}^\top(\mathbf{x})(\mathbf{LL}^\top)^{1/2}(\mathbf{LL}^\top)^{-1/2}\boldsymbol{\Delta}\right) \\
&= \mathrm{Tr}\left(\left((\mathbf{LL}^\top)^{-1/2}\boldsymbol{\Delta}\boldsymbol{\Delta}^\top(\mathbf{LL}^\top)^{-1/2}\right)\left((\mathbf{LL}^\top)^{1/2}\mathbf{f}(\mathbf{x})\mathbf{f}^\top(\mathbf{x})(\mathbf{LL}^\top)^{1/2}\right)\right).
\end{aligned}
\tag{198}
$$

We have seen in Lemma F.4 that the eigenvalues of $(\mathbf{LL}^\top)^{-1/2}\boldsymbol{\Delta}\boldsymbol{\Delta}^\top(\mathbf{LL}^\top)^{-1/2}$ are given (in decreasing order) by $1 - \rho_i^2$ where $\rho_i$ are the canonical correlations of $\mathbf{g}$ and $\mathbf{g}'$. We now consider the other term and claim that the eigenvalues of

$$(\mathbf{LL}^\top)^{1/2}\mathbb{E}_{\mathbf{x} \sim p_\mathbf{x}}\left(\mathbf{f}(\mathbf{x})\mathbf{f}^\top(\mathbf{x})\right)(\mathbf{LL}^\top)^{1/2} \tag{199}$$

are given by $\mu_1 \geq \mu_2 \geq \ldots \geq \mu_m$ defined as the non-zero eigenvalues of

$$\mathbb{E}_{\mathbf{x} \sim p_\mathbf{x}}(\mathbf{u}(\mathbf{x})\mathbf{u}^\top(\mathbf{x})) = \mathbb{E}_{\mathbf{x} \sim p_\mathbf{x}}\left(\mathbf{L}^\top\mathbf{f}(\mathbf{x})\mathbf{f}^\top(\mathbf{x})\mathbf{L}\right) \in \mathbb{R}^{k \times k}. \tag{200}$$

Note that there are indeed at most $m$ non-zero eigenvalues since $\mathbf{L}$ has rank $m$, potentially there are less if the image of $f$ is restricted to a hyperplane. Since the eigenvalues of $\mathbf{AB}$ and $\mathbf{BA}$ agree for all matrices we find that the spectrum $\sigma$ satisfies

$$\sigma\left((\mathbf{LL}^\top)^{1/2}\mathbb{E}_{\mathbf{x} \sim p_\mathbf{x}}\left(\mathbf{f}(\mathbf{x})\mathbf{f}^\top(\mathbf{x})\right)(\mathbf{LL}^\top)^{1/2}\right) = \sigma\left(\mathbf{LL}^\top\mathbb{E}_{\mathbf{x} \sim p_\mathbf{x}}\left(\mathbf{f}(\mathbf{x})\mathbf{f}^\top(\mathbf{x})\right)\right) = \sigma\left(\mathbb{E}_{\mathbf{x} \sim p_\mathbf{x}}\left(\mathbf{L}^\top\mathbf{f}(\mathbf{x})\mathbf{f}^\top(\mathbf{x})\mathbf{L}\right)\right) \tag{201}$$

We can now apply von Neumann's trace inequality (see (32)) and conclude that (note that $\rho_i$ and $\mu_i$ were both defined as decreasing so $(1 - \rho_i^2)$ is increasing

$$\sum_{i=1}^{m}(1 - \rho_i^2)\mu_i \leq \mathbb{E}_{\mathbf{x}\sim p_{\mathbf{x}}} \operatorname{Tr}\left(\left((\mathbf{L}\mathbf{L}^\top)^{-1/2}\boldsymbol{\Delta}\boldsymbol{\Delta}^\top(\mathbf{L}\mathbf{L}^\top)^{-1/2}\right)\left((\mathbf{L}\mathbf{L}^\top)^{1/2}\mathbf{f}(\mathbf{x})\mathbf{f}^\top(\mathbf{x})(\mathbf{L}\mathbf{L}^\top)^{1/2}\right)\right) = \mathbb{E}_{\mathbf{x}\sim p_{\mathbf{x}}}\|\boldsymbol{\Delta}^\top\mathbf{f}(\mathbf{x})\|^2. \tag{202}$$

$\square$

### F.2. Bounding similarity of the embeddings

We now perform essentially the same analysis to obtain bounds for the canonical correlations of $\mathbf{f}$ and $\mathbf{f}'$. We present the results in slightly shorter form.

We consider the regression problem

$$\tilde{\mathbf{B}} = \min_{\overline{\mathbf{B}}\in\mathbb{R}^{m\times m'}} \mathbb{E}_{\mathbf{x}\sim p_{\mathbf{x}}}\|\mathbf{f}(\mathbf{x}) - \overline{\mathbf{B}}\mathbf{f}'(\mathbf{x})\|^2 \tag{203}$$

which has the solution

$$\tilde{\mathbf{B}} = \mathbb{E}_{\mathbf{x}\sim p_{\mathbf{x}}}(\mathbf{f}(\mathbf{x})\mathbf{f}'(\mathbf{x})^\top)\left(\mathbb{E}_{\mathbf{x}\sim p_{\mathbf{x}}}(\mathbf{f}'(\mathbf{x})\mathbf{f}'(\mathbf{x})^\top)\right)^{-1}. \tag{204}$$

It is convenient to introduce the notation

$$\mathbf{M}_{\mathbf{f},\mathbf{f}} = \mathbb{E}_{\mathbf{x}\sim p_{\mathbf{x}}}(\mathbf{f}(\mathbf{x})\mathbf{f}(\mathbf{x})^\top) \tag{205}$$

for the moment matrices and $\mathbf{M}_{\mathbf{f},\mathbf{f}'}$ are defined similarly. Then we can write

$$\tilde{\mathbf{B}} = \mathbf{M}_{\mathbf{f},\mathbf{f}'}\mathbf{M}_{\mathbf{f}',\mathbf{f}'}^{-1}. \tag{206}$$

We again consider the residual

$$\tilde{\boldsymbol{\Delta}}(\mathbf{x}) = \mathbf{f}(\mathbf{x}) - \tilde{\mathbf{B}}\mathbf{f}'(\mathbf{x}) \tag{207}$$

which satisfies

$$\mathbf{M}_{\tilde{\boldsymbol{\Delta}},\mathbf{f}'} = 0 \tag{208}$$

where $\mathbf{M}_{\tilde{\boldsymbol{\Delta}},\mathbf{f}'} = \mathbb{E}(\tilde{\boldsymbol{\Delta}}(\mathbf{x})\mathbf{f}(\mathbf{x})^\top)$ is the moment matrix defined similar to (181). Similar to Lemma F.2 we get the following result.

**Lemma F.6.** *With the notation introduced above, we can decompose*

$$\mathbb{E}_{\mathbf{x}\sim p_{\mathbf{x}}}\|\mathbf{u}(\mathbf{x}) - \mathbf{u}'(\mathbf{x})\|^2 = \mathbb{E}_{\mathbf{x}\sim p_{\mathbf{x}}}\left(\|\mathbf{L}^\top\tilde{\boldsymbol{\Delta}}(\mathbf{x})\|^2 + \|\mathbf{L}^\top\tilde{\mathbf{B}}\mathbf{f}'(\mathbf{x}) - \mathbf{L}'^\top\mathbf{f}'(\mathbf{x})\|^2\right). \tag{209}$$

*Remark* F.7. Similar to Remark F.3 we can bound the norm of the residual

$$\mathbb{E}_{\mathbf{x}\sim p_{\mathbf{x}}}\|\mathbf{u}(\mathbf{x}) - \mathbf{u}'(\mathbf{x})\|^2 \geq \lambda_{\min}(\mathbf{L}\mathbf{L}^\top)\mathbb{E}_{\mathbf{x}\sim p_{\mathbf{x}}}\|\tilde{\boldsymbol{\Delta}}(\mathbf{x})\|^2. \tag{210}$$

*Proof.* We calculate

$$\begin{aligned}
\|\mathbf{u}(\mathbf{x}) - \mathbf{u}'(\mathbf{x})\|^2 &= \|\mathbf{L}^\top\mathbf{f}(\mathbf{x}) - \mathbf{L}'^\top\mathbf{f}'(\mathbf{x})\|^2 \\
&= \|\mathbf{L}^\top\tilde{\boldsymbol{\Delta}}(\mathbf{x}) + \mathbf{L}^\top\tilde{\mathbf{B}}\mathbf{f}'(\mathbf{x}) - \mathbf{L}'^\top\mathbf{f}'\|^2 \\
&= \tilde{\boldsymbol{\Delta}}(\mathbf{x})^\top\mathbf{L}\mathbf{L}^\top\tilde{\boldsymbol{\Delta}}(\mathbf{x}) + \left(\mathbf{L}^\top\tilde{\mathbf{B}}\mathbf{f}'(\mathbf{x}) - \mathbf{L}'^\top\mathbf{f}'(\mathbf{x})\right)^\top\left(\mathbf{L}^\top\tilde{\mathbf{B}}\mathbf{f}'(\mathbf{x}) - \mathbf{L}'^\top\mathbf{f}'(\mathbf{x})\right) \\
&\quad + \tilde{\boldsymbol{\Delta}}(\mathbf{x})^\top\mathbf{L}\left(\mathbf{L}^\top\tilde{\mathbf{B}}\mathbf{f}'(\mathbf{x}) - \mathbf{L}'^\top\mathbf{f}'(\mathbf{x})\right) + \left(\mathbf{L}^\top\tilde{\mathbf{B}}\mathbf{f}'(\mathbf{x}) - \mathbf{L}'^\top\mathbf{f}'(\mathbf{x})\right)^\top\mathbf{L}^\top\tilde{\boldsymbol{\Delta}}(\mathbf{x}) \\
&= \|\mathbf{L}^\top\tilde{\boldsymbol{\Delta}}(\mathbf{x})\|^2 + \|\mathbf{L}^\top\tilde{\mathbf{B}}\mathbf{f}'(\mathbf{x}) - \mathbf{L}'^\top\mathbf{f}'(\mathbf{x})\|^2 \\
&\quad + \operatorname{Tr}\left(\mathbf{f}'(\mathbf{x})\tilde{\boldsymbol{\Delta}}(\mathbf{x})^\top\mathbf{L}\left(\mathbf{L}^\top\tilde{\mathbf{B}} - \mathbf{L}'^\top\right)\right) + \operatorname{Tr}\left(\left(\tilde{\mathbf{B}}^\top\mathbf{L} - \mathbf{L}'\right)^\top\mathbf{L}^\top\tilde{\boldsymbol{\Delta}}(\mathbf{x})\mathbf{f}'(\mathbf{x})^\top\right).
\end{aligned} \tag{211}$$

Integrating over $\mathbf{x}$ we find using (208) that the cross terms vanish

$$\text{Tr}\left(\underbrace{\mathbb{E}_{\mathbf{x}\sim p_{\mathbf{x}}}\left(\mathbf{f}'(\mathbf{x})\tilde{\boldsymbol{\Delta}}(\mathbf{x})^{\top}\right)}_{=0}\mathbf{L}\left(\mathbf{L}^{\top}\tilde{\mathbf{B}}-\mathbf{L}'^{\top}\right)\right)+\text{Tr}\left(\left(\tilde{\mathbf{B}}^{\top}\mathbf{L}-\mathbf{L}'\right)^{\top}\mathbf{L}^{\top}\underbrace{\mathbb{E}_{\mathbf{x}\sim p_{\mathbf{x}}}\left(\tilde{\boldsymbol{\Delta}}(\mathbf{x})\mathbf{f}'(\mathbf{x})^{\top}\right)}_{=0}\right)=0. \quad (212)$$

And the claim follows by taking expectation in (211). $\qquad\square$

We now want to relate the term $\mathbb{E}_{\mathbf{x}\sim p_{\mathbf{x}}}\left(\|\mathbf{L}^{\top}\tilde{\boldsymbol{\Delta}}(\mathbf{x})\|^{2}\right)$ to the canonical correlation of $\mathbf{f}(\mathbf{x})$ and $\mathbf{f}'(\mathbf{x})$. For this we prove a result similar to Lemma F.4. Recall that the canonical correlation of $\mathbf{f}$ and $\mathbf{f}'$, given by the singular values of $\mathbf{M}_{\mathbf{f},\mathbf{f}}^{-1/2}\mathbf{M}_{\mathbf{f},\mathbf{f}'}\mathbf{M}_{\mathbf{f}',\mathbf{f}'}^{-1/2}$, by $\tilde{\rho}_{1}\geq\tilde{\rho}_{2}\geq\ldots\geq\tilde{\rho}_{\min(m,m')}$ (which we extend to 0 for $i>\min(m,m')$ if necessary).

**Lemma F.8.** *Denote the eigenvalues of* $\mathbf{M}_{\mathbf{f},\mathbf{f}}^{-1/2}\mathbb{E}_{\mathbf{x}\sim p_{\mathbf{x}}}\left(\tilde{\boldsymbol{\Delta}}(\mathbf{x})\tilde{\boldsymbol{\Delta}}^{\top}(\mathbf{x})\right)\mathbf{M}_{\mathbf{f},\mathbf{f}}^{-1/2}$ *by* $\tilde{\lambda}_{1}\leq\ldots\leq\tilde{\lambda}_{m}$. *Then the relation*

$$\tilde{\lambda}_{i}=1-\tilde{\rho}_{i}^{2} \quad (213)$$

*holds, where we set* $\rho_{i}=0$ *for* $m'<i\leq m$ *if* $m'<m$.

*Proof.* We expand

$$\begin{aligned}\mathbb{E}_{\mathbf{x}\sim p_{\mathbf{x}}}\left(\tilde{\boldsymbol{\Delta}}(\mathbf{x})\tilde{\boldsymbol{\Delta}}^{\top}(\mathbf{x})\right)&=\mathbb{E}_{\mathbf{x}\sim p_{\mathbf{x}}}\left(\mathbf{f}(\mathbf{x})\mathbf{f}(\mathbf{x})^{\top}-\mathbf{f}(\mathbf{x})\mathbf{f}'(\mathbf{x})^{\top}\mathbf{M}_{\mathbf{f}',\mathbf{f}'}^{-1}\mathbf{M}_{\mathbf{f}',\mathbf{f}}-\mathbf{M}_{\mathbf{f},\mathbf{f}'}\mathbf{M}_{\mathbf{f}',\mathbf{f}'}^{-1}\mathbf{f}(\mathbf{x})^{\top}\mathbf{f}'(\mathbf{x})\right.\\&\left.+\mathbf{M}_{\mathbf{f},\mathbf{f}'}\mathbf{M}_{\mathbf{f}',\mathbf{f}'}^{-1}\mathbf{f}(\mathbf{x})^{\top}\mathbf{f}'(\mathbf{x})^{\top}\mathbf{M}_{\mathbf{f}',\mathbf{f}'}^{-1}\mathbf{M}_{\mathbf{f}',\mathbf{f}}\right)\\&=\mathbf{M}_{\mathbf{f},\mathbf{f}}-2\mathbf{M}_{\mathbf{f},\mathbf{f}'}\mathbf{M}_{\mathbf{f}',\mathbf{f}'}^{-1}\mathbf{M}_{\mathbf{f}',\mathbf{f}}+\mathbf{M}_{\mathbf{f},\mathbf{f}'}\mathbf{M}_{\mathbf{f}',\mathbf{f}'}^{-1}\mathbf{M}_{\mathbf{f}',\mathbf{f}'}\mathbf{M}_{\mathbf{f}',\mathbf{f}'}^{-1}\mathbf{M}_{\mathbf{f}',\mathbf{f}}\\&=\mathbf{M}_{\mathbf{f},\mathbf{f}}-\mathbf{M}_{\mathbf{f},\mathbf{f}'}\mathbf{M}_{\mathbf{f}',\mathbf{f}'}^{-1}\mathbf{M}_{\mathbf{f}',\mathbf{f}}.\end{aligned} \quad (214)$$

Thus we find

$$\mathbf{M}_{\mathbf{f},\mathbf{f}}^{-1/2}\mathbb{E}_{\mathbf{x}\sim p_{\mathbf{x}}}\left(\tilde{\boldsymbol{\Delta}}(\mathbf{x})\tilde{\boldsymbol{\Delta}}^{\top}(\mathbf{x})\right)\mathbf{M}_{\mathbf{f},\mathbf{f}}^{-1/2}=\mathbf{1}_{m\times m}-\left(\mathbf{M}_{\mathbf{f},\mathbf{f}}^{-1/2}\mathbf{M}_{\mathbf{f},\mathbf{f}'}\mathbf{M}_{\mathbf{f}',\mathbf{f}'}^{-1/2}\right)^{\top}\left(\mathbf{M}_{\mathbf{f},\mathbf{f}}^{-1/2}\mathbf{M}_{\mathbf{f},\mathbf{f}'}\mathbf{M}_{\mathbf{f}',\mathbf{f}'}^{-1/2}\right). \quad (215)$$

This then implies that

$$\tilde{\lambda}_{i}=1-\tilde{\rho}_{i}^{2} \quad (216)$$

where we extend $\tilde{\rho}_{i}=0$ for $m'<i\leq m$ if $m\geq m'$. $\qquad\square$

Based on this result we can now derive a bound for canonical correlation coefficients deriving the analogue of Lemma F.5.

**Lemma F.9.** *Recall that we denoted the eigenvalues of the second moment of the logits* $\mathbb{E}_{\mathbf{x}\sim p_{\mathbf{x}}}(\mathbf{u}(\mathbf{x})\mathbf{u}^{\top}(\mathbf{x}))\in\mathbb{R}^{k\times k}$ *by* $\mu_{1}\geq\mu_{2}\geq\ldots\geq\mu_{m}$ *and the canonical correlations of* $\mathbf{f}$ *and* $\mathbf{f}'$ *by* $\tilde{\rho}_{1}\geq\tilde{\rho}_{2}\geq\ldots\geq\tilde{\rho}_{\min(m,m')}$ *Then the following bound holds*

$$\sum_{i=1}^{m}(1-\tilde{\rho}_{i}^{2})\mu_{i}\leq\mathbb{E}_{\mathbf{x}\sim p_{\mathbf{x}}}\|\mathbf{L}^{\top}\tilde{\boldsymbol{\Delta}}(\mathbf{x})\|^{2}. \quad (217)$$

*Proof.* We rewrite using cyclicity of the trace as in (198)

$$\begin{aligned}\|\mathbf{L}^{\top}\tilde{\boldsymbol{\Delta}}(\mathbf{x})\|^{2}&=\text{Tr}\left(\tilde{\boldsymbol{\Delta}}(\mathbf{x})^{\top}\mathbf{M}_{\mathbf{f},\mathbf{f}}^{-1/2}\mathbf{M}_{\mathbf{f},\mathbf{f}}^{1/2}\mathbf{L}\mathbf{L}^{\top}\mathbf{M}_{\mathbf{f},\mathbf{f}}^{1/2}\mathbf{M}_{\mathbf{f},\mathbf{f}}^{-1/2}\tilde{\boldsymbol{\Delta}}(\mathbf{x})\right)\\&=\text{Tr}\left(\left(\mathbf{M}_{\mathbf{f},\mathbf{f}}^{-1/2}\tilde{\boldsymbol{\Delta}}(\mathbf{x})\tilde{\boldsymbol{\Delta}}(\mathbf{x})^{\top}\mathbf{M}_{\mathbf{f},\mathbf{f}}^{-1/2}\right)\left(\mathbf{M}_{\mathbf{f},\mathbf{f}}^{1/2}\mathbf{L}\mathbf{L}^{\top}\mathbf{M}_{\mathbf{f},\mathbf{f}}^{1/2}\right)\right).\end{aligned} \quad (218)$$

Applying Lemma F.8 we find that the eigenvalues of $\mathbf{M}_{\mathbf{f},\mathbf{f}}^{-1/2}\tilde{\boldsymbol{\Delta}}(\mathbf{x})\tilde{\boldsymbol{\Delta}}(\mathbf{x})^{\top}\mathbf{M}_{\mathbf{f},\mathbf{f}}^{-1/2}$ are given (in decreasing order) by $1-\tilde{\rho}_{i}^{2}$ where $\tilde{\rho}_{i}$ are the canonical correlations of $\mathbf{f}$ and $\mathbf{f}'$. As in Lemma F.5 the other term

$$\mathbf{M}_{\mathbf{f},\mathbf{f}}^{1/2}\mathbf{L}\mathbf{L}^{\top}\mathbf{M}_{\mathbf{f},\mathbf{f}}^{1/2} \quad (219)$$

has eigenvalues $\mu_1 \geq \mu_2 \geq \ldots \geq \mu_m$. Indeed, we find

$$\sigma\left(\mathbf{M}_{\mathbf{f},\mathbf{f}}^{1/2}\mathbf{L}\mathbf{L}^\top\mathbf{M}_{\mathbf{f},\mathbf{f}}^{1/2}\right) = \sigma\left(\mathbf{M}_{\mathbf{f},\mathbf{f}}\mathbf{L}\mathbf{L}^\top\right) = \sigma\left(\mathbf{L}^\top\mathbf{M}_{\mathbf{f},\mathbf{f}}\mathbf{L}\right). \tag{220}$$

Then von Neumann's trace inequality (32) implies

$$\sum_{i=1}^m (1 - \tilde{\rho}_i^2)\mu_i \leq \mathbb{E}_{\mathbf{x}\sim p_{\mathbf{x}}}\|\mathbf{L}^\top\tilde{\boldsymbol{\Delta}}(\mathbf{x})\|^2. \tag{221}$$

$\square$

After these preparations the proof of Theorem F.1 is straightforward.

*Proof of Theorem F.1.* Applying Lemma F.5 followed by Lemma F.2 we conclude that

$$\sum_{i=1}^m (1 - \rho_i^2)\mu_i \leq \mathbb{E}_{\mathbf{x}\sim p_{\mathbf{x}}}\|\boldsymbol{\Delta}^\top\mathbf{f}(\mathbf{x})\|^2 \leq \mathbb{E}_{\mathbf{x}\sim p_{\mathbf{x}}}\|\mathbf{u}(\mathbf{x}) - \mathbf{u}'(\mathbf{x})\|^2 \tag{222}$$

which is (184). The bound (185) follows similarly from Lemma F.6 and F.9. $\square$

## F.3. Relating embeddings and unembeddings

The results in Theorem F.1 can be used to show that $\mathbf{f}$ and $\mathbf{f}'$ and $\mathbf{g}$ and $\mathbf{g}'$ are linearly related when the model logits are similar. Concretely we bounded the canonical correlations and we can also bound the residual (see Remark F.3). However, while these are quantitative versions of the linear identifiability of $\mathbf{f}$ and $\mathbf{g}$ this alone is not sufficient to recover the identifiability result as this also requires that $\mathbf{f} = \tilde{\mathbf{B}}\mathbf{f}'$, $\mathbf{g} = \mathbf{B}\mathbf{g}'$ and $\tilde{\mathbf{B}}_{\mathcal{J}}^{-\top} = \tilde{\mathbf{B}}$. Therefore, a natural question is whether we can show quantitative bounds for $\mathbf{B}^\top\tilde{\mathbf{B}} - \mathbf{1}_{m'\times m'}$ (this matrix can also be considered for $m' \neq m$). However, when considering the equivalent model $(\mathbf{A}^\top\mathbf{f}', \mathbf{A}^{-1}\mathbf{g}')$ for $\mathbf{A} \in \mathrm{GL}(m')$ we find that the transition matrix becomes $\mathbf{A}^{-1}\mathbf{B}^\top\tilde{\mathbf{B}}\mathbf{A}$. However, if $\mathbf{B}^\top\tilde{\mathbf{B}}$ is not a multiple of the identity we have

$$\sup_{\mathbf{A}\in\mathrm{GL}(m')}\|\mathbf{A}^{-1}\mathbf{B}^\top\tilde{\mathbf{B}}\mathbf{A}\| = \infty \tag{223}$$

and thus no meaningful bounds on $\|\mathbf{B}^\top\tilde{\mathbf{B}}\|$ in terms of $\mathbb{E}_{\mathbf{x}\sim p_{\mathbf{x}}}\|\mathbf{u}(\mathbf{x}) - \mathbf{u}'(\mathbf{x})\|^2$ are possible. Instead, we can only show a weaker statement for the relation of $\mathbf{B}$ and $\tilde{\mathbf{B}}$. To achieve this we argue as in Lemmas F.2 and F.6.

**Lemma F.10.** *When* $\mathbb{E}_{\mathbf{x}\sim p_{\mathbf{x}}}\|\mathbf{u}'(\mathbf{x}) - \mathbf{u}(\mathbf{x})\|^2 = 0$, *then* $\tilde{\mathbf{B}}^\top\tilde{\mathbf{B}} = \mathbf{1}_{m'\times m'}$. *Moreover,* $\tilde{\mathbf{B}}^\top\tilde{\mathbf{B}}$ *behaves like an approximate identity in relevant model direction in the sense that*

$$\mathbb{E}_{\mathbf{x}\sim p_{\mathbf{x}}}\|\mathbf{L}'^\top\mathbf{B}^\top\tilde{\mathbf{B}}\mathbf{f}'(\mathbf{x}) - \mathbf{L}'^\top\mathbf{f}'(\mathbf{x})\|^2 \leq \mathbb{E}_{\mathbf{x}\sim p_{\mathbf{x}}}\|\mathbf{u}(\mathbf{x}) - \mathbf{u}'(\mathbf{x})\|^2 \tag{224}$$

*and the logits* $\mathbf{L}'^\top\mathbf{B}^\top\tilde{\mathbf{B}}\mathbf{f}'(\mathbf{x})$ *of the model* $(\mathbf{B}\mathbf{g}', \tilde{\mathbf{B}}\mathbf{f}')$ *also satisfy*

$$\mathbb{E}_{\mathbf{x}\sim p_{\mathbf{x}}}\|\mathbf{L}'^\top\mathbf{B}^\top\tilde{\mathbf{B}}\mathbf{f}'(\mathbf{x}) - \mathbf{u}(\mathbf{x})\|^2 \leq 4\mathbb{E}_{\mathbf{x}\sim p_{\mathbf{x}}}\|\mathbf{u}'(\mathbf{x}) - \mathbf{u}(\mathbf{x})\|^2. \tag{225}$$

*Remark* F.11. One could derive bounds on $\|\tilde{\mathbf{B}}^\top\tilde{\mathbf{B}} - \mathbf{1}_{m'\times m'}\|$ when assuming lower bounds on the spectrum of the moments $\mathbf{M}_{\mathbf{f}',\mathbf{f}'}$ and $\mathbf{L}'\mathbf{L}'^\top$, however these assumptions are again not invariant under equivalence transformations of the model.

*Proof.* We start from (209) and decompose the second term further using

$$\begin{aligned}\|\mathbf{L}^\top\tilde{\mathbf{B}}\mathbf{f}'(\mathbf{x}) - \mathbf{L}'^\top\mathbf{f}'(\mathbf{x})\|^2 &= \|\boldsymbol{\Delta}^\top\tilde{\mathbf{B}}\mathbf{f}'(\mathbf{x}) + \mathbf{L}'^\top\mathbf{B}^\top\tilde{\mathbf{B}}\mathbf{f}'(\mathbf{x}) - \mathbf{L}'^\top\mathbf{f}'(\mathbf{x})\|^2 \\ &= \|\boldsymbol{\Delta}^\top\tilde{\mathbf{B}}\mathbf{f}'(\mathbf{x})\|^2 + \|\mathbf{L}'^\top\mathbf{B}^\top\tilde{\mathbf{B}}\mathbf{f}'(\mathbf{x}) - \mathbf{L}'^\top\mathbf{f}'(\mathbf{x})\|^2.\end{aligned} \tag{226}$$

Here the Pythagorean identity in the last step follows as in Lemma F.2 from $\boldsymbol{\Delta}\mathbf{L}'^\top = 0$ (see (191)). From Lemma F.6 we infer that

$$\mathbb{E}_{\mathbf{x}\sim p_{\mathbf{x}}}\|\mathbf{L}'^\top\mathbf{B}^\top\tilde{\mathbf{B}}\mathbf{f}'(\mathbf{x}) - \mathbf{L}'^\top\mathbf{f}'(\mathbf{x})\|^2 \leq \mathbb{E}_{\mathbf{x}\sim p_{\mathbf{x}}}\|\mathbf{u}(\mathbf{x}) - \mathbf{u}'(\mathbf{x})\|^2. \tag{227}$$

This implies that $\mathbf{B}^\top \tilde{\mathbf{B}} = \mathbf{1}_{m' \times m'}$ (when $\mathbf{M}_{\mathbf{f}',\mathbf{f}}$ and $\mathbf{L}\mathbf{L}^\top$ are non-degenerate), in particular $m = m'$. Moreover, we conclude using $(a + b)^2 \leq 2a^2 + 2b^2$ that

$$\mathbb{E}_{\mathbf{x} \sim p_\mathbf{x}} \| \mathbf{L}'^\top \mathbf{B}^\top \tilde{\mathbf{B}} \mathbf{f}'(\mathbf{x}) - \mathbf{u}(\mathbf{x}) \|^2 \leq 2 \mathbb{E}_{\mathbf{x} \sim p_\mathbf{x}} \left( \| \mathbf{L}'^\top \mathbf{B}^\top \tilde{\mathbf{B}} \mathbf{f}'(\mathbf{x}) - \mathbf{u}'(\mathbf{x}) \|^2 + \| \mathbf{u}'(\mathbf{x}) - \mathbf{u}(\mathbf{x}) \|^2 \right) \tag{228}$$
$$\leq 4 \mathbb{E}_{\mathbf{x} \sim p_\mathbf{x}} \| \mathbf{u}'(\mathbf{x}) - \mathbf{u}(\mathbf{x}) \|^2.$$

In particular the logits of $(\mathbf{Bg}', \tilde{\mathbf{B}}\mathbf{f}')$ are, on average, close to the logits of the model $(\mathbf{g}, \mathbf{f})$. $\qquad \square$

### F.4. Lower Bounds for the Mean Canonical Correlation

In this section we collect applications of Theorem F.1. The proof of Theorem 3.4 can be found at the end of this section. First we show how the theorem can be used to bound the mean canonical correlation coefficient, $m_{\mathrm{CCA}}$.

**Theorem F.12.** *Assume the hypothesis of Theorem F.1, in particular we assume that the moment matrix $\mathbf{M}_{\mathbf{u},\mathbf{u}} = \mathbb{E}_{\mathbf{x} \sim p_\mathbf{x}}(\mathbf{u}(\mathbf{x})\mathbf{u}(\mathbf{x})^\top)$ has rank $m$. Denote by $\mu_{\min} = \mu_m > 0$ the smallest non-zero eigenvalue of $\mathbf{M}_{\mathbf{u},\mathbf{u}}$. Then the bounds*

$$\sum_{i=1}^m \rho_i \geq \sum_{i=1}^m \rho_i^2 \geq m - \frac{\mathbb{E}_{\mathbf{x} \sim p_\mathbf{x}} \| \mathbf{u}(\mathbf{x}) - \mathbf{u}'(\mathbf{x}) \|^2}{\mu_{\min}} \quad \text{and} \tag{229}$$

$$\sum_{i=1}^m \tilde{\rho}_i \geq \sum_{i=1}^m \tilde{\rho}_i^2 \geq m - \frac{\mathbb{E}_{\mathbf{x} \sim p_\mathbf{x}} \| \mathbf{u}(\mathbf{x}) - \mathbf{u}'(\mathbf{x}) \|^2}{\mu_{\min}} \tag{230}$$

*hold.*

*Proof.* Theorem F.1 implies

$$\frac{\mathbb{E}_{\mathbf{x} \sim p_\mathbf{x}} \| \mathbf{u}(\mathbf{x}) - \mathbf{u}'(\mathbf{x}) \|^2}{\mu_{\min}} \geq \sum_{i=1}^m (1 - \rho_i^2) \frac{\mu_i}{\mu_{\min}} \geq \sum_{i=1}^m (1 - \rho_i^2) \tag{231}$$

where we used $\rho_i^2 \leq 1$ (so $1 - \rho_i^2 \geq 0$) and $\frac{\mu_i}{\mu_{\min}} \geq 1$ in the last step. This implies the first bound and the second estimate follows similarly. $\qquad \square$

Note that there are at most $\min(m', m)$ non-zero canonical correlations so the result directly implies lower bound on $m'$ to match the other models distribution. Or, stated equivalently, we can bound the information loss a smaller model must incur. In particular, we conclude lower bounds for the representation dimension of a student that tries to match the teacher's logits with a given accuracy and those bounds are explicit functions of the teacher's logits. This result can be slightly refined and we obtain the following theorem which closely resembles the reconstruction bound for PCA.

**Theorem F.13.** *Assume the hypothesis of Theorem F.1 and denote the eigenvalues of $\mathbf{M}_{\mathbf{u},\mathbf{u}}$ in decreasing order by $\mu_i$. Suppose the dimension $m'$ of the student model satisfies $m' < m$. Then we find*

$$\mathbb{E}_{\mathbf{x} \sim p_\mathbf{x}} \| \mathbf{u}(\mathbf{x}) - \mathbf{u}'(\mathbf{x}) \|^2 \geq \sum_{i=m'+1}^m \mu_i. \tag{232}$$

*Proof.* The result follows from Theorem F.1 by using that $\rho_i = 0$ for $i > m'$. $\qquad \square$

Finally, we discuss the modifications to consider the standard covariance matrix based canonical correlations. For this, we will need the slightly stronger diversity condition that the covariance matrix

$$\mathbf{\Sigma}_{\mathbf{u},\mathbf{u}} = \mathbb{E}_{\mathbf{x} \sim p_\mathbf{x}} \left( (\mathbf{u}(\mathbf{x}) - \mathbb{E}_{\mathbf{x} \sim p_\mathbf{x}}(\mathbf{u}(\mathbf{x}))) (\mathbf{u}(\mathbf{x}) - \mathbb{E}_{\mathbf{x} \sim p_\mathbf{x}}(\mathbf{u}(\mathbf{x})))^\top \right) \tag{233}$$

has rank $m$. Let us denote the standard (covariance-based) canonical correlations of $\mathbf{f}$ and $\mathbf{f}'$ by $\bar{\rho}_i$. Then the following result holds.

**Theorem F.14.** *Consider two models* $(\mathbf{f}, \mathbf{g})$ *and* $(\mathbf{f}', \mathbf{g}')$ *with representation dimensions* $m$ *and* $m'$ *and assume that* $\boldsymbol{\Sigma}_{\mathbf{f},\mathbf{f}}$, $\mathbf{L}$ *and* $\boldsymbol{\Sigma}_{\mathbf{f}',\mathbf{f}'}$, $\mathbf{L}'$ *have maximal rank* $m$ *and* $m'$ *respectively. Denote by* $\mu_m > 0$ *the smallest non-zero eigenvalue of* $\boldsymbol{\Sigma}_{\mathbf{u},\mathbf{u}}$. *Then the bounds*

$$\sum_{i=1}^{m} \rho_i \geq \sum_{i=1}^{m} \rho_i^2 \geq m - \frac{\mathbb{E}_{\mathbf{x}\sim p_{\mathbf{x}}}\|\mathbf{u}(\mathbf{x}) - \mathbf{u}'(\mathbf{x})\|^2}{\mu_m} \quad and \tag{234}$$

$$\sum_{i=1}^{m} \bar{\rho}_i \geq \sum_{i=1}^{m} \bar{\rho}_i^2 \geq m - \frac{\mathbb{E}_{\mathbf{x}\sim p_{\mathbf{x}}}\|\mathbf{u}(\mathbf{x}) - \mathbf{u}'(\mathbf{x})\|^2}{\mu_m} \tag{235}$$

*hold.*

*Proof.* Denote the mean of the embeddings by $\bar{\mathbf{f}} = \mathbb{E}_{\mathbf{x}\sim p_{\mathbf{x}}}(\mathbf{f}(\mathbf{x}))$ and $\bar{\mathbf{f}}' = \mathbb{E}_{\mathbf{x}\sim p_{\mathbf{x}}}(\mathbf{f}'(\mathbf{x}))$. We consider the embedding-centred models $(\mathbf{f}^c, \mathbf{g}) = (\mathbf{f} - \bar{\mathbf{f}}, \mathbf{g})$ and $((\mathbf{f}')^c, \mathbf{g}') = (\mathbf{f}' - \bar{\mathbf{f}}', \mathbf{g}')$. The logits $\mathbf{u}^c$ of the shifted model are given by

$$\mathbf{u}^c = \mathbf{L}^\top (\mathbf{f}(\mathbf{x}) - \bar{\mathbf{f}}) = \mathbf{u}(\mathbf{x}) - \mathbb{E}_{\mathbf{x}\sim p_{\mathbf{x}}}(\mathbf{u}(\mathbf{x})) \tag{236}$$

where $\mathbf{u}$ are the logits of the original model and a similar expression holds for $\mathbf{u}'$. This implies $\mathbf{M}_{\mathbf{u}^c, \mathbf{u}^c} = \boldsymbol{\Sigma}_{\mathbf{u},\mathbf{u}}$. Moreover, the moment-based and covariance-based canonical correlations agree for the shifted models $(\mathbf{f}^c, \mathbf{g})$ and $((\mathbf{f}')^c, \mathbf{g}')$ and they further agree with the covariance-based (i.e., shift invariant) canoncial correlations of $\mathbf{f}$ and $\mathbf{f}'$. We can thus now apply Theorem F.12 to the models with the shifted logits and we find

$$\sum_{i=1}^{m} \bar{\rho}_i \geq \sum_{i=1}^{m} \bar{\rho}_i^2 \geq m - \frac{\mathbb{E}_{\mathbf{x}\sim p_{\mathbf{x}}}\|\mathbf{u}^c(\mathbf{x}) - (\mathbf{u}')^c(\mathbf{x})\|^2}{\mu_m} \geq m - \frac{\mathbb{E}_{\mathbf{x}\sim p_{\mathbf{x}}}\|\mathbf{u}(\mathbf{x}) - \mathbf{u}'(\mathbf{x})\|^2}{\mu_m}. \tag{237}$$

Here we used

$$\mathbb{E}_{\mathbf{x}\sim p_{\mathbf{x}}}\|\mathbf{u}(\mathbf{x}) - \mathbf{u}'(\mathbf{x})\|^2 \geq \mathbb{E}_{\mathbf{x}\sim p_{\mathbf{x}}}\|\mathbf{u}^c(\mathbf{x}) - (\mathbf{u}')^c(\mathbf{x})\|^2 \tag{238}$$

in the last step. The proof of (234) is similar. □

*Proof of Theorem 3.4.* The statement follows directly from Theorem F.14 when plugging in the definition (30) of $m_{\text{CCA}}$. □

### F.5. Illustration of Bound from Thm. F.1

In Fig. 3 we illustrate the bound from Thm. F.1 using 5 seeds of students trained to match a teacher model on `CIFAR`. We see that the dashed lines (computed values of the left hand side of the inequality) are above the solid lines (computed values of the right hand side of the inequality) for all measured epochs as the theorem says.

## G. Example of Models with small KL Divergence and Equal Embeddings, but Dissimilar Unembeddings

In this section, we give an illustration of why we want to measure representational similarity that takes into account the equivalence relation of the identifiability class and why it is not enough to consider a measure which only considers similarity between the embeddings (or unembeddings) (see App. H for another example).

We do this by constructing a pair of models with equal embeddings and small KL divergence, but dissimilar unembeddings as measured with $m_{\text{CCA}}$. Note that these models will also be dissimilar in terms of $d_{\text{rep}}$, since $d_{\text{rep}}$ includes the unembeddings through the unembeddings matrices. These constructed models are displayed in Figs. 4 and 5

Assume we have models defining a distribution over $8$ labels, i.e., $\mathcal{Y} = \{y_1, ..., y_8\}$, and with representation dimensionality $m = 2$. For a subset of input data, a reference model $(\mathbf{f}, \mathbf{g}) \in \Theta$ assigns high probability to one among the first four labels, as displayed by embeddings aligning to unembeddings directions $\mathbf{g}(y_i)$, $i = 1, \ldots, 4$. On the remaining input points, the model assign high mass, but equal probability, to either the labels $y_5$ and $y_6$, or $y_7$ and $y_8$. This can be seen from the embeddings aligned to $\mathbf{g}(y_5) + \mathbf{g}(y_6)$ or to $\mathbf{g}(y_7) + \mathbf{g}(y_8)$.

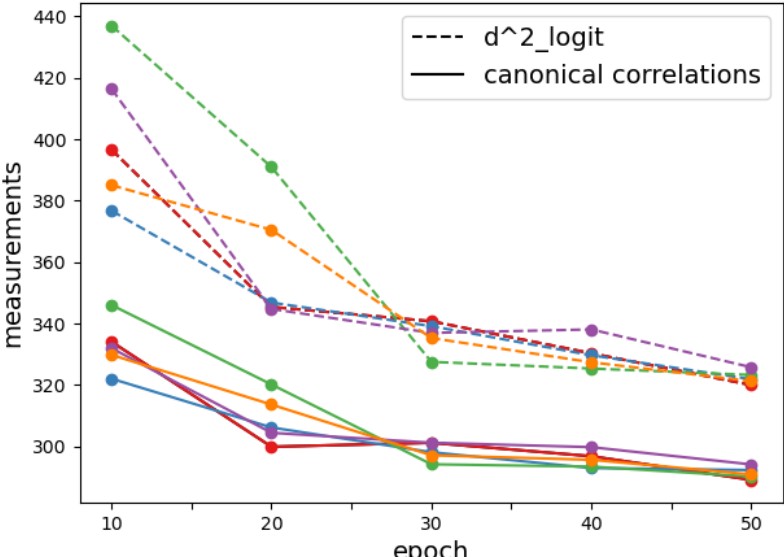

*Figure 3.* Illustration of the bound in Thm. F.1 for 5 seeds of students to a teacher on `CIFAR` at training epochs $10, 20, 30, 40$ and $50$. The solid lines are the sum including the canonical correlations between student and teacher from the left hand side of the inequality, while the dashed lines are the squared logit distances from the right hand side. We see that the squared logit distances are larger for all epochs.

A second model $(\mathbf{f}', \mathbf{g}') \in \Theta$ has equal embeddings to $(\mathbf{f}, \mathbf{g})$, that is $\mathbf{f}'(\mathbf{x}) = \mathbf{f}(\mathbf{x})$ for all $\mathbf{x} \in \mathcal{X}$, but dissimilar unembeddings. It is constructed so as to have equal unembeddings for the first 4 labels, i.e., $\mathbf{g}'(y_i) = \mathbf{g}(y_i)$ for $i = 1, \ldots, 4$, and swapped unembeddings for $y_5$ and $y_6$ swapped and for $y_7$ and $y_8$:

$$\begin{cases} \mathbf{g}'(y_5) = \mathbf{g}(y_6) \\ \mathbf{g}'(y_6) = \mathbf{g}(y_5) \end{cases} \quad \text{and} \quad \begin{cases} \mathbf{g}'(y_7) = \mathbf{g}(y_8) \\ \mathbf{g}'(y_8) = \mathbf{g}(y_7) \end{cases}. \tag{239}$$

Despite the unembeddings of the models being dissimilar in a linear sense, we can now make $d_{\text{KL}}$ arbitrarily small by increasing the norm of the embeddings or the unembeddings (For the illustrated example, we have $d_{\text{KL}}((f,g)|(f',g')) = 0.002$ and $d_{\text{KL}}((f',g')|(f,g)) = 0.0017$). Notice that the embeddings that are between the unembeddings on the left are exactly in the middle in terms of angle. Therefore, since all the unembeddings have the same norm, probability values of the two most likely labels will be equal for the two models. However, increasing the norm, will not make the unembeddings of the two models closer to being linear transformations of each other.

Note also that we can choose any of the first four labels $y_1, ..., y_4$, to make the $\mathbf{L}, \mathbf{L}$ matrices with columns $\tilde{\mathbf{g}}(y_i), \tilde{\mathbf{g}}'(y_i)$, and this will give us $\mathbf{L} = \mathbf{L}'$. In particular, we have $\mathbf{A} = \mathbf{L}^{-\top}\mathbf{L}'^{\top} = \mathbf{I}$, so

$$\mathbf{f}(\mathbf{x}) = \mathbf{A}\mathbf{f}'(\mathbf{x}) \tag{240}$$

for all $\mathbf{x}$ like in the identifiability result, but we do not have the corresponding part for the unembeddings. That is, $\tilde{\mathbf{g}}(y) \neq \mathbf{A}^{-\top}\tilde{\mathbf{g}}'(y)$.

In terms of the mean canonical correlation, $m_{\text{CCA}}$, we have for the embeddings $m_{\text{CCA}}(\mathbf{f}(\mathbf{x}), \mathbf{f}'(\mathbf{x})) = 1$ and for the unembeddings $m_{\text{CCA}}(\mathbf{g}(y), \mathbf{g}'(y)) = 0.67$ (The $m_{\text{CCA}}$ for the unembeddings can be made smaller by spreading out unembeddings for labels $y_5, y_6, y_7, y_8$ further).

Thus, we see that there can be a perfect linear relationship between embeddings and, even if the distributions are very close in terms of KL divergence, there is still no guarantee for the unembeddings to be similar up to the inverse linear transformation of the embeddings.

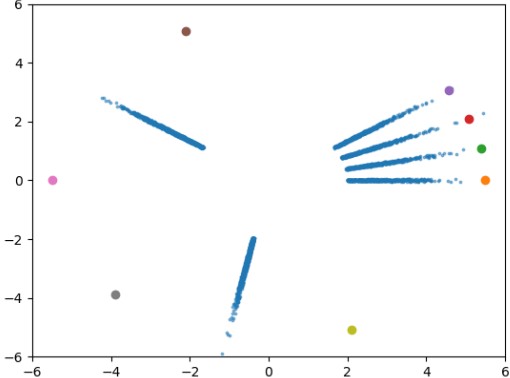

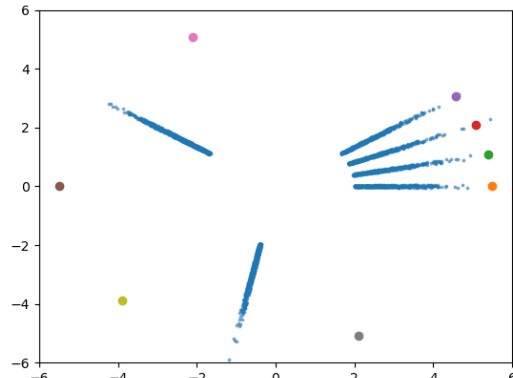

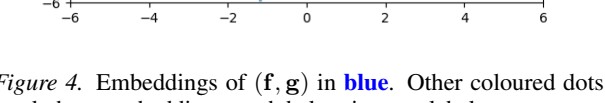

*Figure 4.* Embeddings of $(\mathbf{f}, \mathbf{g})$ in **blue**. Other coloured dots mark the unembeddings, each belonging to a label.

*Figure 5.* Embeddings of $(\mathbf{f}', \mathbf{g}')$ in **blue**. Other coloured dots mark the unembeddings. Note that brown and pink-colored unembeddings are swapped compared to Fig. 4, as are grey and yellow.

## H. Example of Models with Maximal Mean Canonical Correlation, but Different Distributions

In this section, we give an illustration of why we want to measure representational similarity that takes into account the equivalence relation of the identifiability class and why it is not enough to consider a measure which only considers similarity between the embeddings (or unembeddings) (see App. G above for another example).

The main idea of this example is that we can make two models, where the representations of one is a rotated version of the representations of the other, but because the rotations of the embeddings and unembeddings do not match, the distributions of the models are very different (see Figs. 6 and 7). We give the specific details below.

Let the models have 5 possible labels. We construct model 1, $(\mathbf{f}, \mathbf{g})$, to have unembeddings at angles $0, \pi/2, \pi, \frac{5\pi}{4}$, and $\frac{7\pi}{4}$ with lengths 8. The embeddings are gaussian distributions with centers at the same angles and lengths 5, Fig. 6.

We construct model 2, $(\mathbf{f}', \mathbf{g}')$, by giving it unembeddings equal a clockwise rotation of the unembeddings of model 1 by $\frac{\pi}{4}$, and embeddings equal to a counter-clockwise rotation of the embeddings of model 1 by $\frac{\pi}{4}$, Fig. 7.

Now by construction, $m_{\text{CCA}}(\mathbf{f}(\mathbf{x}), \mathbf{f}'(\mathbf{x})) = 1$ and $m_{\text{CCA}}(\mathbf{g}(y), \mathbf{g}'(y)) = 1$, but the distributions of the models are very different. For example, the embeddings which model 1 will assign label 1, will be assigned label 2 by model 2, and those assigned label 2 by model 1, will be assigned label 3 by model 2.

## I. Theoretical Material for Section 3.4: The Logit Distance Bounds the Linear Identifiability Dissimilarity

In this section, we will first show how the squared distance between logits can be written as a normalized distance between representations App. I.1, and then show how the logit distance bounds the linear identifiability dissimilarity from Def. 3.7 in App. I.2.

### I.1. Connection Between Distance between Logits and Representations

**Theorem I.1.** *Let $p, p'$ be as defined above, and $m$ the dimension of the representations. Let $N = \binom{k-2}{m-1}$. Then the squared distance between logits can be written as a normalized distance between the representations in the following sense: Let $\tilde{\mathbf{L}}_{\mathcal{J}} = \mathbf{U}\mathbf{D}\mathbf{V}^{\top}$ be the singular value decomposition of $\tilde{\mathbf{L}}_{\mathcal{J}}$, and let $\tilde{\mathbf{B}}_{\mathcal{J}} = \mathbf{D}^{-1}\mathbf{U}^{\top}$. Then we have*

$$\|\mathbf{u}(\mathbf{x}) - \mathbf{u}'(\mathbf{x})\|_2^2 = \frac{1}{2kN} \sum_{\tilde{y} \in \mathcal{Y}} \sum_{\mathcal{J} \subseteq \mathcal{Y} \setminus \{\tilde{y}\}} \left\| \tilde{\mathbf{B}}_{\mathcal{J}}^{-\top} \left( \mathbf{f}(\mathbf{x}) - \tilde{\mathbf{A}}_{\mathcal{J}} \mathbf{f}'(\mathbf{x}) \right) \right\|_2^2 . \tag{241}$$

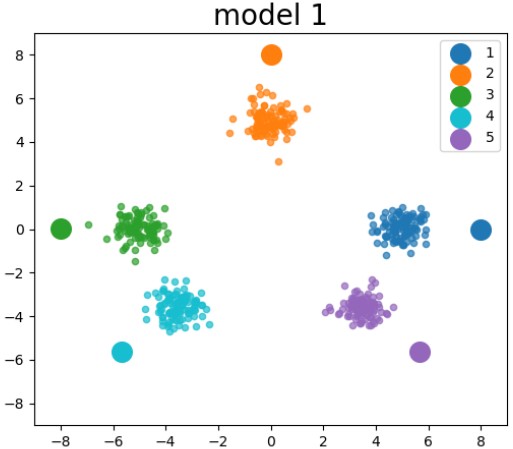
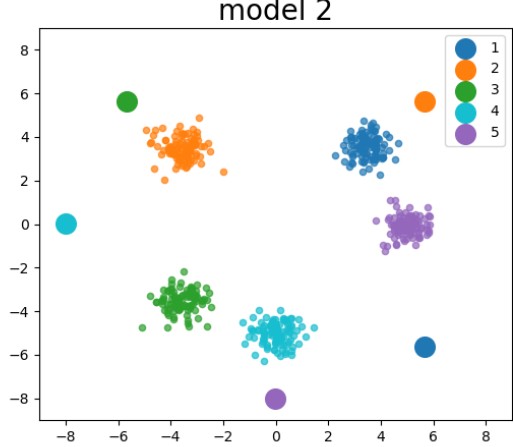

*Figure 6.* Embeddings and unembeddings of model 1, $(\mathbf{f}, \mathbf{g})$. Embeddings are coloured by the label the model will assign to them.

*Figure 7.* Embeddings and unembeddings of model 2, $(\mathbf{f}', \mathbf{g}')$. Embeddings are coloured by the label model 1 will assign to them. Both embeddings and unembeddings of model 2 are rotations of the embeddings and unembeddings of model 1 (so will have $m_{\mathrm{CCA}}$ of 1), but the distributions of the models are very different. For example, the embeddings which model 1 will assign label 1, will be assigned label 2 by model 2.

*Proof.* We consider the norm on the right hand side of Eq. (241):

$$\|\tilde{\mathbf{B}}_{\mathcal{J}}^{-\top}\left(\mathbf{f}(\mathbf{x}) - \tilde{\mathbf{A}}_{\mathcal{J}}\mathbf{f}'(\mathbf{x})\right)\|_2^2 \tag{242}$$

$$= \|\tilde{\mathbf{B}}_{\mathcal{J}}^{-\top}\tilde{\mathbf{L}}_{\mathcal{J}}^{-\top}\left(\tilde{\mathbf{L}}_{\mathcal{J}}^{\top}\mathbf{f}(\mathbf{x}) - \tilde{\mathbf{L}}_{\mathcal{J}}^{'\top}\mathbf{f}'(\mathbf{x})\right)\|_2^2 \tag{243}$$

$$= \|(\tilde{\mathbf{B}}_{\mathcal{J}}\tilde{\mathbf{L}}_{\mathcal{J}})^{-\top}\left(\tilde{\mathbf{L}}_{\mathcal{J}}^{\top}\mathbf{f}(\mathbf{x}) - \tilde{\mathbf{L}}_{\mathcal{J}}^{'\top}\mathbf{f}'(\mathbf{x})\right)\|_2^2 \tag{244}$$

Using the definition of $\tilde{\mathbf{B}}_{\mathcal{J}} = \mathbf{D}^{-1}\mathbf{U}^{\top}$ and recalling that the singular value decomposition of $\tilde{\mathbf{L}}_{\mathcal{J}}$ was $\tilde{\mathbf{L}}_{\mathcal{J}} = \mathbf{U}\mathbf{D}\mathbf{V}^{\top}$, we see that

$$\tilde{\mathbf{B}}_{\mathcal{J}}\tilde{\mathbf{L}}_{\mathcal{J}} = \mathbf{D}^{-1}\mathbf{U}^{\top}\mathbf{U}\mathbf{D}\mathbf{V}^{\top} = \mathbf{V}^{\top} \tag{245}$$

So we have that

$$\|(\tilde{\mathbf{B}}_{\mathcal{J}}\tilde{\mathbf{L}}_{\mathcal{J}})^{-\top}\left(\tilde{\mathbf{L}}_{\mathcal{J}}^{\top}\mathbf{f}(\mathbf{x}) - \tilde{\mathbf{L}}_{\mathcal{J}}^{'\top}\mathbf{f}'(\mathbf{x})\right)\|_2^2 \tag{246}$$

$$= \|\mathbf{V}^{\top}\left(\tilde{\mathbf{L}}_{\mathcal{J}}^{\top}\mathbf{f}(\mathbf{x}) - \tilde{\mathbf{L}}_{\mathcal{J}}^{'\top}\mathbf{f}'(\mathbf{x})\right)\|_2^2 \tag{247}$$

Now since $\mathbf{V}^{\top}$ is an orthonormal matrix, it does not change the norm of a vector. Recall, that for a vector $\mathbf{a}$, we have:

$$\|\mathbf{V}^{\top}\mathbf{a}\|_2^2 = (\mathbf{V}^{\top}\mathbf{a})^{\top}\mathbf{V}^{\top}\mathbf{a} = \mathbf{a}^{\top}\mathbf{V}\mathbf{V}^{\top}\mathbf{a} = \mathbf{a}^{\top}\mathbf{a} = \|\mathbf{a}\|_2^2 \tag{248}$$

So we have that

$$\|\mathbf{V}^{\top}\left(\tilde{\mathbf{L}}_{\mathcal{J}}^{\top}\mathbf{f}(\mathbf{x}) - \tilde{\mathbf{L}}_{\mathcal{J}}^{'\top}\mathbf{f}'(\mathbf{x})\right)\|_2^2 \tag{249}$$

$$= \|\tilde{\mathbf{L}}_{\mathcal{J}}^{\top}\mathbf{f}(\mathbf{x}) - \tilde{\mathbf{L}}_{\mathcal{J}}^{'\top}\mathbf{f}'(\mathbf{x})\|_2^2 \tag{250}$$

So using Lemma D.4, we see that

$$\frac{1}{2kN}\sum_{\tilde{y}\in\mathcal{Y}}\sum_{\mathcal{J}\subseteq\mathcal{Y}\setminus\{\tilde{y}\}}\mathbb{E}_{\mathbf{x}\sim p_{\mathbf{x}}}\|\tilde{\mathbf{B}}_{\mathcal{J}}^{-\top}\left(\mathbf{f}(\mathbf{x})-\tilde{\mathbf{A}}_{\mathcal{J}}\mathbf{f}'(\mathbf{x})\right)\|_2^2 = \frac{1}{2kN}\sum_{\tilde{y}\in\mathcal{Y}}\sum_{\mathcal{J}\subseteq\mathcal{Y}\setminus\{\tilde{y}\}}\mathbb{E}_{\mathbf{x}\sim p_{\mathbf{x}}}\|\tilde{\mathbf{L}}_{\mathcal{J}}^{\top}\mathbf{f}(\mathbf{x})-\tilde{\mathbf{L}}_{\mathcal{J}}^{'\top}\mathbf{f}'(\mathbf{x})\|_2^2 \tag{251}$$

$$= \|\mathbf{u}(\mathbf{x})-\mathbf{u}'(\mathbf{x})\|_2^2 \tag{252}$$

$\square$

This gives us the following corollary (Corollary 3.8 from the main paper):

**Corollary I.2.** *For* $(\mathbf{f},\mathbf{g}),(\mathbf{f}',\mathbf{g}')\in\Theta$, *we have*

$$d_{\mathrm{rep}}((\mathbf{f},\mathbf{g}),(\mathbf{f}',\mathbf{g}')) = 0 \iff d_{\mathrm{logit}}(p_{\mathbf{f},\mathbf{g}},p_{\mathbf{f}',\mathbf{g}'}) = 0\,.$$

*Proof.* " $\Longleftarrow$ ": Assume $d_{\mathrm{logit}}(p,p') = 0$. Since $d_{\mathrm{logit}}$ is a metric between probability distributions, this gives us $p = p'$. Therefore, Corollary 3.6 gives us $d_{\mathrm{rep}}((\mathbf{f},\mathbf{g}),(\mathbf{f}',\mathbf{g}')) = 0$.

" $\Longrightarrow$ ": Assume $d_{\mathrm{rep}}((\mathbf{f},\mathbf{g}),(\mathbf{f}',\mathbf{g}')) = 0$. From Thm. I.1 we get that $d_{\mathrm{logit}}(p,p') = 0$. $\square$

### I.2. Logit Distance Bounds the Linear Identifiability Dissimilarity

In this section we present the full proof of Thm. 3.9, however, we first prove a lemma to help with the main result.

**Lemma I.3.** *Let* $(\mathbf{f},\mathbf{g}),(\underline{\mathbf{f}},\underline{\mathbf{g}})\in\Theta$. *Let* $\tilde{\mathbf{L}}_{\mathcal{J}}$ *be as defined in the priliminaries Eq. (4), and let* $\tilde{\mathbf{L}}_{\mathcal{J}} = \mathbf{U}\mathbf{D}\mathbf{V}^{\top}$ *be the singular value decomposition of* $\tilde{\mathbf{L}}_{\mathcal{J}}$. *Let* $\tilde{\mathbf{B}}_{\mathcal{J}} = \mathbf{D}^{-1}\mathbf{U}^{\top}$. *Let* $\sigma_{\tilde{\mathbf{L}}_{\mathcal{J}}\,\min}$ *be the smallest singular value of* $\tilde{\mathbf{L}}_{\mathcal{J}}$, *and let* $\sigma_{\min}$ *be the smallest singular value of all* $\tilde{\mathbf{L}}_{\mathcal{J}}$. *Then for every choice of labels and input,* $\mathbf{x}$, *we have the following bounds:*

$$\frac{\|\tilde{\mathbf{B}}_{\mathcal{J}}^{-\top}\left(\mathbf{f}(\mathbf{x})-\tilde{\mathbf{A}}_{\mathcal{J}}\mathbf{f}'(\mathbf{x})\right)\|_2}{\sigma_{\min}} \geq \frac{\|\tilde{\mathbf{B}}_{\mathcal{J}}^{-\top}\left(\mathbf{f}(\mathbf{x})-\tilde{\mathbf{A}}_{\mathcal{J}}\mathbf{f}'(\mathbf{x})\right)\|_2}{\sigma_{\tilde{\mathbf{L}}_{\mathcal{J}}\,\min}} \geq \|\mathbf{f}(\mathbf{x})-\tilde{\mathbf{A}}_{\mathcal{J}}\mathbf{f}'(\mathbf{x})\|_2 \tag{253}$$

*And similarly, if we let* $\sigma_{\tilde{\mathbf{L}}_{\mathcal{J}}\,\max}$ *be the largest singular value of* $\tilde{\mathbf{L}}_{\mathcal{J}}$, *and* $\sigma_{\max}$ *be the largest singular value of all* $\tilde{\mathbf{L}}_{\mathcal{J}}$, *we also have:*

$$\frac{\|\tilde{\mathbf{B}}_{\mathcal{J}}^{-\top}\left(\mathbf{f}(\mathbf{x})-\tilde{\mathbf{A}}_{\mathcal{J}}\mathbf{f}'(\mathbf{x})\right)\|_2}{\sigma_{\max}} \leq \frac{\|\tilde{\mathbf{B}}_{\mathcal{J}}^{-\top}\left(\mathbf{f}(\mathbf{x})-\tilde{\mathbf{A}}_{\mathcal{J}}\mathbf{f}'(\mathbf{x})\right)\|_2}{\sigma_{\tilde{\mathbf{L}}_{\mathcal{J}}\,\max}} \leq \|\mathbf{f}(\mathbf{x})-\tilde{\mathbf{A}}_{\mathcal{J}}\mathbf{f}'(\mathbf{x})\|_2 \tag{254}$$

*Proof.* We consider the norm on the left hand side of Eq. (253):, where we recall that $\tilde{\mathbf{B}}_{\mathcal{J}}^{-\top} = \mathbf{D}\mathbf{U}^{\top}$:

$$\|\tilde{\mathbf{B}}_{\mathcal{J}}^{-\top}\left(\mathbf{f}(\mathbf{x})-\tilde{\mathbf{A}}_{\mathcal{J}}\mathbf{f}'(\mathbf{x})\right)\|_2 \tag{255}$$

$$= \|\mathbf{D}\mathbf{U}^{\top}\left(\mathbf{f}(\mathbf{x})-\tilde{\mathbf{A}}_{\mathcal{J}}\mathbf{f}'(\mathbf{x})\right)\|_2 \tag{256}$$

$$= \left\|\mathbf{D}\left(\mathbf{U}^{\top}\mathbf{f}(\mathbf{x})-\mathbf{U}^{\top}\tilde{\mathbf{A}}_{\mathcal{J}}\mathbf{f}'(\mathbf{x})\right)\right\|_2 \tag{257}$$

Now $\mathbf{D}$ is the diagonal matrix with the singular values of $\tilde{\mathbf{L}}_{\mathcal{J}}$ in the diagonal. We define $\mathbf{D}_{\min}$ to be the diagonal matrix with the minimum singular value of $\tilde{\mathbf{L}}_{\mathcal{J}}$, $\sigma_{\tilde{\mathbf{L}}_{\mathcal{J}}\,\min}$, in all the diagonal entries. This means that multiplying a vector with $\mathbf{D}_{\min}$ will make the norm smaller compared to multiplying with $\mathbf{D}$. In other words, we have:

$$\left\|\mathbf{D}\left(\mathbf{U}^{\top}\mathbf{f}(\mathbf{x})-\mathbf{U}^{\top}\tilde{\mathbf{A}}_{\mathcal{J}}\mathbf{f}'(\mathbf{x})\right)\right\|_2 \geq \left\|\mathbf{D}_{\min}\left(\mathbf{U}^{\top}\mathbf{f}(\mathbf{x})-\mathbf{U}^{\top}\tilde{\mathbf{A}}_{\mathcal{J}}\mathbf{f}'(\mathbf{x})\right)\right\|_2 \tag{258}$$

We can calculate the effect of multiplying a vector, $\mathbf{a}$, with $\mathbf{D}_{\min}$:

$$\|\mathbf{D}_{\min}\mathbf{a}\|_2 = \sqrt{\sum_{i=1}^{D}(\sigma_{\tilde{\mathbf{L}}_{\mathcal{J}}\,\min}a_i)^2} = \sqrt{\sigma_{\tilde{\mathbf{L}}_{\mathcal{J}}\,\min}^2\sum_{i=1}^{D}a_i^2} = \sigma_{\tilde{\mathbf{L}}_{\mathcal{J}}\,\min}\sqrt{\sum_{i=1}^{D}a_i^2} = \sigma_{\tilde{\mathbf{L}}_{\mathcal{J}}\,\min}\|\mathbf{a}\|_2 \tag{259}$$

So we have that

$$\left\|\mathbf{D}_{\min}\left(\mathbf{U}^\top \mathbf{f}(\mathbf{x}) - \mathbf{U}^\top \tilde{\mathbf{A}}_{\mathcal{J}} \mathbf{f}'(\mathbf{x})\right)\right\|_2 \tag{260}$$

$$= \sigma_{\tilde{\mathbf{L}}_{\mathcal{J}} \min} \left\|\mathbf{U}^\top \mathbf{f}(\mathbf{x}) - \mathbf{U}^\top \tilde{\mathbf{A}}_{\mathcal{J}} \mathbf{f}'(\mathbf{x})\right\|_2 \tag{261}$$

$$= \sigma_{\tilde{\mathbf{L}}_{\mathcal{J}} \min} \left\|\mathbf{U}^\top \left(\mathbf{f}(\mathbf{x}) - \tilde{\mathbf{A}}_{\mathcal{J}} \mathbf{f}'(\mathbf{x})\right)\right\|_2 \tag{262}$$

Since $\mathbf{U}^\top$ is an orthonormal matrix, it does not change the norm of a vector, so we have that

$$\sigma_{\tilde{\mathbf{L}}_{\mathcal{J}} \min} \left\|\mathbf{U}^\top \left(\mathbf{f}(\mathbf{x}) - \tilde{\mathbf{A}}_{\mathcal{J}} \mathbf{f}'(\mathbf{x})\right)\right\|_2 \tag{263}$$

$$= \sigma_{\tilde{\mathbf{L}}_{\mathcal{J}} \min} \left\|\mathbf{f}(\mathbf{x}) - \tilde{\mathbf{A}}_{\mathcal{J}} \mathbf{f}'(\mathbf{x})\right\|_2 \tag{264}$$

All in all, we have that for any choice of $\tilde{\mathbf{L}}_{\mathcal{J}}$ and $\mathbf{x}$:

$$\|\tilde{\mathbf{B}}_{\mathcal{J}}^{-\top}\left(\mathbf{f}(\mathbf{x}) - \tilde{\mathbf{A}}_{\mathcal{J}} \mathbf{f}'(\mathbf{x})\right)\|_2 \geq \sigma_{\tilde{\mathbf{L}}_{\mathcal{J}} \min} \left\|\mathbf{f}(\mathbf{x}) - \tilde{\mathbf{A}}_{\mathcal{J}} \mathbf{f}'(\mathbf{x})\right\|_2 \tag{265}$$

which means that

$$\frac{\|\tilde{\mathbf{B}}_{\mathcal{J}}^{-\top}\left(\mathbf{f}(\mathbf{x}) - \tilde{\mathbf{A}}_{\mathcal{J}} \mathbf{f}'(\mathbf{x})\right)\|_2}{\sigma_{\tilde{\mathbf{L}}_{\mathcal{J}} \min}} \geq \left\|\mathbf{f}(\mathbf{x}) - \tilde{\mathbf{A}}_{\mathcal{J}} \mathbf{f}'(\mathbf{x})\right\|_2 \tag{266}$$

And since $\sigma_{\min} \leq \sigma_{\tilde{\mathbf{L}}_{\mathcal{J}} \min}$ we get the second inequality.

We can make a similar argument by defining $\mathbf{D}_{\max}$ as the diagonal matrix with the maximum singular value of $\tilde{\mathbf{L}}_{\mathcal{J}}$, $\sigma_{\mathbf{L} \max}$, in all the diagonal entries, and go through the steps from Eq. (258) with the inequality reversed.

$\square$

We are now ready to prove the main result:

**Theorem I.4.** *Let $p, p'$ be as before, and $m$ the dimension of the representations and $k$ the number of labels. Let $C = \sqrt{\frac{2m}{k-1}}$. Let $\sigma_{\tilde{\mathbf{L}}_{\mathcal{J}} \min}$ be the smallest singular value of $\tilde{\mathbf{L}}_{\mathcal{J}}$, and let $\sigma_{\min}$ be the smallest singular value of all $\tilde{\mathbf{L}}_{\mathcal{J}}$. Then $d_{\mathrm{logit}}$, bounds the linear identifiability dissimilarity, $d_{\mathrm{rep}}$, in the following way:*

$$C \frac{d_{\mathrm{logit}}(p, p')}{\sigma_{\min}} \geq d_{\mathrm{rep}}((\mathbf{f}, \mathbf{g}), (\mathbf{f}', \mathbf{g}')) \tag{267}$$

*And similarly, if we let $\sigma_{\tilde{\mathbf{L}}_{\mathcal{J}} \max}$ be the largest singular value of $\tilde{\mathbf{L}}_{\mathcal{J}}$, and $\sigma_{\max}$ be the largest singular value of all $\tilde{\mathbf{L}}_{\mathcal{J}}$, we also have*

$$C \frac{d_{\mathrm{logit}}(p, p')}{\sigma_{\max}} \leq d_{\mathrm{rep}}((\mathbf{f}, \mathbf{g}), (\mathbf{f}', \mathbf{g}')) \tag{268}$$

*Proof.* Let $N = \binom{k-2}{m-1}$ and $J = \binom{k-1}{m}$. We recall from Thm. I.1 that

$$\|\mathbf{u}(\mathbf{x}) - \mathbf{u}'(\mathbf{x})\|_2^2 = \frac{1}{2kN} \sum_{\tilde{y} \in \mathcal{Y}} \sum_{\mathcal{J} \subseteq \mathcal{Y} \setminus \{\tilde{y}\}} \left\|\tilde{\mathbf{B}}_{\mathcal{J}}^{-\top}\left(\mathbf{f}(\mathbf{x}) - \tilde{\mathbf{A}}_{\mathcal{J}} \mathbf{f}'(\mathbf{x})\right)\right\|_2^2 . \tag{269}$$

Also, from Eq. (253) from Lemma I.3, we get have that

$$\frac{\|\tilde{\mathbf{B}}_{\mathcal{J}}^{-\top}\left(\mathbf{f}(\mathbf{x}) - \tilde{\mathbf{A}}_{\mathcal{J}} \mathbf{f}'(\mathbf{x})\right)\|_2}{\sigma_{\tilde{\mathbf{L}}_{\mathcal{J}} \min}} \geq \|\mathbf{f}(\mathbf{x}) - \tilde{\mathbf{A}}_{\mathcal{J}} \mathbf{f}'(\mathbf{x})\|_2 \tag{270}$$

which means that since $\sigma_{\min} \leq \sigma_{\tilde{\mathbf{L}}_{\mathcal{J}} \min}$ we have that

$$\frac{\|\mathbf{u}(\mathbf{x}) - \mathbf{u}'(\mathbf{x})\|_2^2}{\sigma_{\min}^2} \geq \frac{1}{2kN} \sum_{\tilde{y} \in \mathcal{Y}} \sum_{\mathcal{J} \subseteq \mathcal{Y} \setminus \{\tilde{y}\}} \|\mathbf{f}(\mathbf{x}) - \tilde{\mathbf{A}}_{\mathcal{J}} \mathbf{f}'(\mathbf{x})\|_2^2 . \tag{271}$$

Taking the expectation over $\mathbf{x}$, we get

$$\mathbb{E}_{\mathbf{x} \sim p_x} \frac{\|\mathbf{u}(\mathbf{x}) - \mathbf{u}'(\mathbf{x})\|_2^2}{\sigma_{\min}^2} \geq \frac{1}{2kN} \sum_{\tilde{y} \in \mathcal{Y}} \sum_{\mathcal{J} \subseteq \mathcal{Y} \setminus \{\tilde{y}\}} \mathbb{E}_{\mathbf{x} \sim p_x} \|\mathbf{f}(\mathbf{x}) - \tilde{\mathbf{A}}_{\mathcal{J}} \mathbf{f}'(\mathbf{x})\|_2^2 . \tag{272}$$

Which means we have that

$$\frac{d_{\text{logit}}^2(p, p')}{\sigma_{\min}^2} \geq \frac{J}{2N} \frac{1}{kJ} \sum_{\tilde{y} \in \mathcal{Y}} \sum_{\mathcal{J} \subseteq \mathcal{Y} \setminus \{\tilde{y}\}} \mathbb{E}_{\mathbf{x} \sim p_x} \|\mathbf{f}(\mathbf{x}) - \tilde{\mathbf{A}}_{\mathcal{J}} \mathbf{f}'(\mathbf{x})\|_2^2 \tag{273}$$

$$= \frac{J}{2N} d_{\text{rep}}^2((\mathbf{f}, \mathbf{g}), (\mathbf{f}', \mathbf{g}')) , \tag{274}$$

and therefore

$$\frac{d_{\text{logit}}(p, p')}{\sigma_{\min}} \geq \frac{J}{2N} \frac{1}{kJ} \sum_{\tilde{y} \in \mathcal{Y}} \sum_{\mathcal{J} \subseteq \mathcal{Y} \setminus \{\tilde{y}\}} \mathbb{E}_{\mathbf{x} \sim p_x} \|\mathbf{f}(\mathbf{x}) - \tilde{\mathbf{A}}_{\mathcal{J}} \mathbf{f}'(\mathbf{x})\|_2^2 \tag{275}$$

$$= \frac{\sqrt{J}}{\sqrt{2N}} d_{\text{rep}}((\mathbf{f}, \mathbf{g}), (\mathbf{f}', \mathbf{g}')) . \tag{276}$$

We now see that

$$\frac{2N}{J} = \frac{2(k-2)! m! (k-m-1)!}{(m-1)!(k-m-1)!(k-1)!} \tag{277}$$

$$= \frac{2(k-2)! m(m-1)!(k-m-1)!}{(m-1)!(k-m-1)!(k-1)(k-2)!} \tag{278}$$

$$= \frac{2m}{(k-1)} . \tag{279}$$

So for $C = \sqrt{\frac{2m}{(k-1)}}$ we get

$$C \frac{d_{\text{logit}}(p, p')}{\sigma_{\min}} \geq d_{\text{rep}}((\mathbf{f}, \mathbf{g}), (\mathbf{f}', \mathbf{g}')) , \tag{280}$$

which was what we wanted to prove. If we let $\sigma_{\tilde{\mathbf{L}}_{\mathcal{J}} \max}$ be the largest singular value of $\tilde{\mathbf{L}}_{\mathcal{J}}$, and $\sigma_{\max}$ be the largest singular value of all $\tilde{\mathbf{L}}_{\mathcal{J}}$, we can go through the same steps with the reversed inequality. $\square$

## J. Proof of Theorem 4.3

In this Section we show that when $d_{\text{logit}}(p, p')$ between two models is small, then concepts in one model are approximately represented in another model, in particular we prove Theorem 4.3. We recall that $\mathbf{L}$ is the matrix of unembeddings and that a concept $\mathbf{h}$ defines a conditional distribution $p_{\mathbf{h}}(c \mid \mathbf{x})$, over $C \geq 2$ values.

Consider a model $(\mathbf{f}, \mathbf{g})$, such that $\mathbf{h}$ is linearly represented in $\mathbf{f}$. For every $c \in \{1, \ldots, C\}$, we write the concept vectors $\mathbf{w}_c \in \mathbb{R}^m$ as $\mathbf{L}\boldsymbol{\alpha}_c$ for some $\boldsymbol{\alpha}_c \in \mathbb{R}^k$ which we call *concept weights* as they are the weights when writing $\mathbf{w}_c = \sum_{i=1}^k \mathbf{g}(y_i)(\alpha_c)_i$. Since we assume that $\mathbf{L}$ has full rank, this is possible for all $\mathbf{w}_c$. Moreover, this is a natural way to parametrize the concepts as it is invariant to linear transformations of the embedding space and concepts of interest are generally aligned with the unembeddings $\mathbf{g}(y)$. We also define the matrix $\mathbf{A} = (\boldsymbol{\alpha}_c)_{c \in \{1, \ldots, C\}} \in \mathbb{R}^{k \times C}$ and this is related to the concept weight matrix through $\mathbf{W} = \mathbf{L}\mathbf{A}$.

We now show that these concepts can be approximately realized on the embedding space of another model $(\mathbf{f}', \mathbf{g}') \in \Theta$ when the logit distance to $(\mathbf{f}', \mathbf{g}')$ is small. Indeed, we consider the concept matrix

$$\mathbf{W}' = \mathbf{L}'\mathbf{A}' \in \mathbb{R}^{m \times C} \tag{281}$$

and therefore the individual concept vectors are given by

$$\mathbf{w}'_c = \mathbf{L}'\boldsymbol{\alpha}'_c. \tag{282}$$

We can then define (the biases remain invariant $b_c = b'_c$)

$$p'_{\mathbf{h}}(c|\mathbf{x}) = \frac{\exp\left(\mathbf{w}'^{\top}_c \mathbf{f}'(\mathbf{x}) + b_c\right)}{\sum_{j=1}^{C} \exp\left(\mathbf{w}'^{\top}_j \mathbf{f}'(\mathbf{x}) + \mathbf{b}_j\right)} \tag{283}$$

We then get the following result.

**Theorem J.1.** *For two models $(\mathbf{f}, \mathbf{g}), (\mathbf{f}', \mathbf{g}') \in \Theta$ and a concept $\mathbf{h} : \mathcal{X} \to \Delta_{[C]}$ with $C \geq 2$, if $\mathbf{h}$ is linearly encoded in $\mathbf{f}$, then*

$$\min_{\mathbf{W}' \in \mathbb{R}^{C \times m}, \mathbf{b}' \in \mathbb{R}^C} d_{\mathrm{KL}}\left(p_{\mathbf{h}}(\cdot \mid \mathbf{x}), \mathrm{softmax}(\mathbf{W}'\mathbf{f}'(\mathbf{x}) + \mathbf{b}')\right) \leq \frac{1}{2}\|\mathbf{A}\|_{op}^2 d_{\mathrm{logit}}^2(p, p'). \tag{284}$$

*where $\mathbf{A}$ is determined from $\mathbf{W} = \mathbf{L}\mathbf{A}$, $\|\cdot\|_{\mathrm{op}}$ denotes the operator norm, where $\mathbf{W} \in \mathbb{R}^{m \times C}$ is the matrix of weights $\mathbf{w}_c$ from Def. 4.2 for $\mathbf{f}$ linearly encoding $\mathbf{h}$.*

*Proof.* Since $\mathbf{h}$ is linearly encoded in $\mathbf{f}$, we have $\mathbf{W} \in \mathbb{R}^{m \times C}$ and $\mathbf{b} \in \mathbb{R}^C$ as weights and biases that gives:

$$\mathrm{softmax}\left(\mathbf{W}^{\top}\mathbf{f}(\mathbf{x}) + \mathbf{b}\right) = p_{\mathbf{h}}(c \mid \mathbf{x}). \tag{285}$$

Let

$$p'_{\mathbf{h}}(c \mid \mathbf{x}) = \mathrm{softmax}\left(\mathbf{W}'^{\top}\mathbf{f}'(\mathbf{x}) + \mathbf{b}'\right). \tag{286}$$

We bound the minimum of $d_{\mathrm{KL}}$ starting from KL divergence. We bound for fixed $\mathbf{x}$ using Lemma E.9 (last line)

$$\min_{\mathbf{W}' \in \mathbb{R}^{m \times C}, \mathbf{b}' \in \mathbb{R}^C} \mathsf{KL}(p_{\mathbf{h}}(\cdot|\mathbf{x})||p'_{\mathbf{h}}(\cdot|\mathbf{x})) = \tag{287}$$

$$= \min_{\mathbf{W}' \in \mathbb{R}^{m \times C}, \mathbf{b}' \in \mathbb{R}^C} \mathsf{KL}\left(\mathrm{softmax}(\mathbf{W}^{\top}\mathbf{f}(\mathbf{x}) + \mathbf{b})||\mathrm{softmax}(\mathbf{W}'^{\top}\mathbf{f}'(\mathbf{x}) + \mathbf{b})\right) \tag{288}$$

$$\leq \min_{\mathbf{W}' \in \mathbb{R}^{m \times C}, \mathbf{b}' \in \mathbb{R}^C} \frac{1}{2}\|\mathbf{W}^{\top}\mathbf{f}(\mathbf{x}) - \mathbf{W}'^{\top}\mathbf{f}'(\mathbf{x}) + \mathbf{b} - \mathbf{b}'\|^2 \tag{289}$$

Next, we bound the minimum in Eq. (289) setting $\mathbf{b}' = \mathbf{b}$. We use that $\mathbf{W} := \mathbf{L}\mathbf{A}$ and $\mathbf{W}' = \mathbf{L}'\mathbf{A}'$, so we turn minimization of $\mathbf{W}'$ into a minimization of $\mathbf{A}'$:

$$\min_{\mathbf{W}' \in \mathbb{R}^{m \times C}} \frac{1}{2}\|\mathbf{W}^{\top}\mathbf{f}(\mathbf{x}) - \mathbf{W}'^{\top}\mathbf{f}'(\mathbf{x})\|^2 = \min_{\mathbf{A}' \in \mathbb{R}^{k \times C}} \frac{1}{2}\|\mathbf{A}^{\top}\mathbf{L}^{\top}\mathbf{f}(\mathbf{x}) - \mathbf{A}'^{\top}\mathbf{L}'^{\top}\mathbf{f}'(\mathbf{x})\|^2 \tag{290}$$

$$\leq \frac{1}{2}\|\mathbf{A}^{\top}\mathbf{u}(\mathbf{x}) - \mathbf{A}^{\top}\mathbf{u}'(\mathbf{x})\|^2 \tag{291}$$

$$\leq \frac{1}{2}\|\mathbf{A}^{\top}\|_{\mathrm{op}}^2 \cdot \|\mathbf{u}(\mathbf{x}) - \mathbf{u}'(\mathbf{x})\|^2. \tag{292}$$

where, in the second line, we used that $\min_{\mathbf{A}'} \|\mathbf{A}^{\top}\mathbf{z} - \mathbf{A}'^{\top}\mathbf{z}'\|^2 \leq \|\mathbf{A}^{\top}\mathbf{z} - \mathbf{A}^{\top}\mathbf{z}'\|^2$ for any $\mathbf{z}, \mathbf{z}' \in \mathbb{R}^k$ and, in the last line, we used that the $L_2$ norm for a matrix vector product satisfies $\|\mathbf{A}\mathbf{v}\| \leq \|\mathbf{v}\| \cdot \|\mathbf{A}\|_{op}$. We end the proof by taking the expectation over $\mathbf{x} \sim p_{\mathbf{x}}$. $\square$

## K. $L_1$ Based Loss Bounds Distribution Distance

We here present the proof of the following statement:

**Proposition K.1.** *Assume that $p$, $p'$ are two models which both satisfy Assumption 3.2 for some $\tau > 0$. Then*

$$\mathbb{E}_{\mathbf{x} \sim p_{\mathbf{x}}}\|\mathbf{u}(\mathbf{x}) - \mathbf{u}'(\mathbf{x})\|_2^2 \leq 2|\log(\tau)|\mathbb{E}_{\mathbf{x} \sim p_{\mathbf{x}}}\|\mathbf{u}(\mathbf{x}) - \mathbf{u}'(\mathbf{x})\|_1. \tag{293}$$

*Proof.* Note that $1 \geq p(y|\mathbf{x}) \geq \tau$ and thus $0 \geq \log(p(y|\mathbf{x})) \geq \log(\tau)$ which also implies $0 \geq \overline{\log(p(\cdot|\mathbf{x}))} \geq \log(\tau)$ denoting again by $\overline{\cdot}$ the average over the labels. Then we can apply (164) and get

$$
\begin{aligned}
(\mathbf{u}(\mathbf{x}) - \mathbf{u}'(\mathbf{x}))_i^2 &= \left| \log(p(y_i|\mathbf{x})) - \overline{\log(p(\cdot|\mathbf{x}))} - \log(p'(y_i|\mathbf{x})) + \overline{\log(p'(\cdot|\mathbf{x}))} \right| \cdot |(\mathbf{u}(\mathbf{x}) - \mathbf{u}'(\mathbf{x}))_i| \\
&\leq 2|\log(\tau)| \cdot |(\mathbf{u}(\mathbf{x}) - \mathbf{u}'(\mathbf{x}))_i|.
\end{aligned}
\tag{294}
$$

We conclude that

$$
\|\mathbf{u}(\mathbf{x}) - \mathbf{u}'(\mathbf{x})\|_2^2 \leq 2|\log(\tau)| \cdot \sum_{i=1}^{k} |(\mathbf{u}(\mathbf{x}) - \mathbf{u}'(\mathbf{x}))_i| = 2|\log(\tau)| \cdot \|\mathbf{u}(\mathbf{x}) - \mathbf{u}'(\mathbf{x})\|_1.
\tag{295}
$$

Taking the expectation over $p_{\mathbf{x}}$ we get the claim. $\qquad\square$

## L. Additional Results Details on Experiments

Code for all experiments and plots can be found on github:
https://github.com/bemigini/logit-distance-bounds-rep-sim.

### L.1. Ablation on representation dimensionality

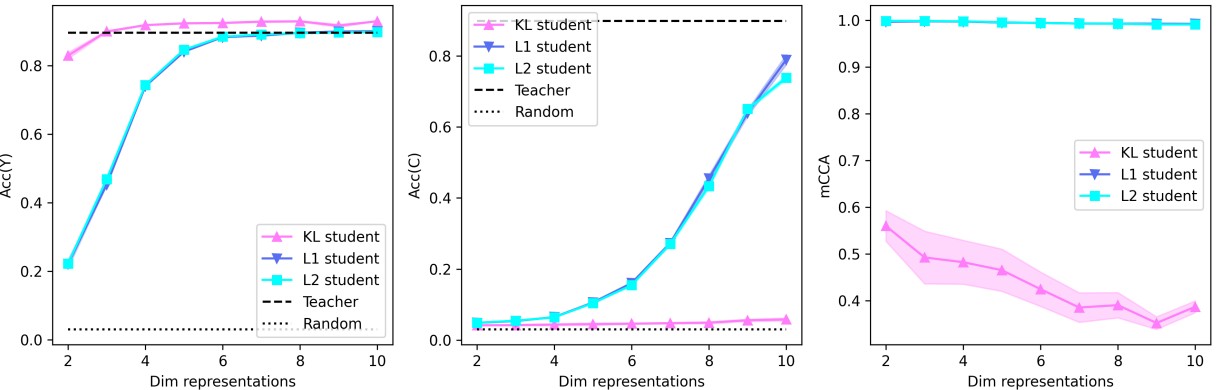

*Figure 8.* **Ablation with increasing representational dimensionality for students in SUB**. We variate the dimension of representations of students on SUB (the teacher model has representation dimensionality $m = 10$). As we increase the representational dimensionality of the students (denoted as $m'$), we observe two distinct trends for KL students on one side, and L1 and L2 students on the other. KL students reach teacher Acc(Y) with $m' = 3$, with an upward trend as $m'$ is increased. On the other hand, L1 and L2 students match teacher performance only with $m' \geq 6$. KL students perform poorly in Acc(C), with almost negligible improvemens when $m'$ is increased. L1 and L2 instead considerably increase their performance as $m'$ grows, showing that they bettter preserve the linearly encoded concepts of the teacher. For each student, we measure mCCA on a subset of representations of the teacher, considering the $m'$ most aligned subspace to student representations. L1 and L2 students always fare optimally, while KL degrades as $m'$ increases, showing bad linear alignment to teacher representations. Mean and standard deviations are evaluated on a single teacher for 5 seed runs per student.

### L.2. Dataset specifics

**Synth** consists of 7 classes distributed over 4 spiral rays. The dataset is created to ensure that, in input space, there are not two classes that are always close to each other and that input data are not linearly separable.

We generate $2d$ input data as follows: for each of the seven classes, we define four angles, each capturing the starting angle of one of the spirals. Then, for each of these rays with angle $\theta_0$, varying the radius $\rho \in [1, 10]$, we randomly sample a point near the spirals with an angle

$$
\theta_i = \theta_0 + 2\pi/T \cdot \rho/10 + \epsilon
\tag{296}
$$

where $T$ denotes the period of the spiral and $\epsilon \sim \text{Unif}(-\sigma(\rho), \sigma(\rho))$ is sampled so to ensure that it never crosses other spiral rays, by regulating the interval $[-\sigma(\rho), \sigma(\rho)]$ on the value of $\rho$. The dataset is visualized in Fig. 2.

Training data amount to $14k$ instances, while the validation and test dataset each comprise $7k$ data points.

**CIFAR.** We use the CIFAR100 (Krizhevsky et al., 2009) loaded from `torchvision` with the original split with 500 images per class for training and 100 images per class for testing. So all in all 50,000 images for training and 10,000 images for test. We do not make a validation split for this dataset.

**SUB** is a regenerated version of `CUB200` (Wah et al., 2011), a dataset for bird classification with annotation on salient bird features. In `SUB`, data points are annotated with a label $Y$ capturing the bird specie, and a concept $C$ indicating the attribute which was modified during generation.

The dataset contains a total of 38.4k images, splitted in 21.5k for training, 5.3k for validation, and 11.5k for test. It comprises 33 annotated class labels. Each image is also annotated with a concept reflecting a salient feature of the bird that has been modified during generation, such as the color of the breast or the shape of the beak, for a total of 33 different concept attributes.

We use both these annotations to train teacher models. This enforces the teacher to both linearly encode concepts and labels in the embeddings.

### L.3. Architectures and hyperparameters details

For `Synth`, we adopt an MLP constituted of 2 hidden layers, each with 512 units and ReLU activations. Teachers and students use this architecture and map to a ($m = 2$) representation space.

Teachers are trained for 1500 epochs, while students are trained for 250 epochs on the training set annotated with teacher's probabilities. For all runs, we use a learning rate of $10^{-3}$ with exponential decay on each epoch set to $\gamma = 0.995$, and batch size of 512.

For `CIFAR` 100, we use ResNet(He et al., 2016) models for the embedding part of both teacher and student, but of different sizes. For the teacher, we use a ResNet50 and for the student a ResNet18 with the same architecture as implemented by torchvision[7]. The final linear layer of the embedding model for both student and teacher goes to 50 dimensions. For the unembeddings we use a single linear layer, giving us 100 (one for each label) 50-dimensional unembeddings. We initialize the teacher model with weights trained on ImageNet taken from torchvision (ResNet50_Weights.IMAGENET1K_V2) and train the teacher for 10 epochs on the train split of the `CIFAR` dataset. Each student is then trained for 50 epochs on the training set with either $\mathcal{L}_{\text{logit}}^1$ ($L_1$ student), $d_{\text{logit}}^2$ ($L_2$ student), or KL divergence (KL student) between student and teacher distributions as a loss. For all runs on `CIFAR`, we use a learning rate of $10^{-3}$ with exponential decay on each epoch set to $\gamma = 0.995$, and batch size of 32.

For `SUB`, we first preprocess all images with DINOv2 (Oquab et al., 2024), mapping them to 768 dimensional embeddings. Then, we train a shallow neural network with 1024 units, with ReLU activations, dropout set to $p = 0.1$ and BatchNorm. Both teachers and students have the same architecture, and the representations for all models are ($m = 10$)-dimensional.

Teachers and students are trained for 500 epochs to ensure convergence and, for all runs, we use a learning rate of $5 \cdot 10^{-3}$ with exponential decay on each epoch set to $\gamma = 0.995$, and batch size of 512.

### L.4. Metrics Evaluation

We use `sklearn.evaluation` for label and concept accuracy, while we implement with `torch` the evaluation of $d_{\text{KL}}$, $d_{\text{logit}}$, and use `sklearn.cross_decomposition.CCA` to evaluate $m_{\text{CCA}}$.

To evaluate $d_{\text{rep}}$, we adopt the following procedure. Depending on the number of labels, evaluating all possible choices of pivots and subsets of labels can be unfeasible. We thus consider selecting at random the pivots and the subset of labels to reduce the computational cost. For `Synth`, we evaluate $d_{\text{rep}}$ for all possible labels and pivots. For `CIFAR` we select all possible pivots and for each of them we consider 200 subsets of labels selected at random among the

$$\binom{100 - 1}{50} \approx 5 \cdot 10^{28}$$

possible combinations. For `SUB`, we select all possible pivots and for each of them we consider 250 subsets of labels selected

---

[7]`docs.pytorch.org/vision/main/models/generated/torchvision.models.resnet50.html`

at random among the

$$\binom{33-1}{10} \approx 6 \cdot 10^7$$

possible combinations. This number of samples is very small compared to the total number of combinations possible, but the most important thing is that we get information from all possible labels and with $200(250)$ samples of $50(10)$ labels, we can be relatively sure to draw each of the $100(33)$ labels at least once.

**Aggregation of measures.** Reported values on teachers are aggregating by evaluating the mean and standard deviations over 5 runs. For each teacher, we run 5 different seeds for both the KL student and the $L_1$ student, and evaluate the mean value $\mu_i$ and variance $\sigma_i^2$, for $i \in \{1, \ldots, 5\}$. We then aggregate all mean and variances by considering the inverse-variance weighting. The mean is given by

$$\omega_i = \frac{1}{\sigma_i^2}, \quad \bar{\mu} = \frac{\sum \omega_i \mu_i}{\sum_i \omega_i}, \tag{297}$$

whereas the standard deviation is obtained as

$$\bar{\sigma} = \sqrt{\frac{1}{\sum_i \omega_i}}. \tag{298}$$

The final results are reported as $\bar{\mu} \pm \bar{\sigma}$ for all metrics.

## L.5. Extended results

We report here the full result of our experimental investigation on the three dataset we consider. Precision is set to 3 for all tables and metrics.

*Table 3.* **Full results on `Synth` and `CIFAR` 100**. We notice that on CIFAR, $L_1$ and $L_2$ students have very similar $d_{\text{logit}}$, but the $L_2$ student has much smaller $d_{\text{rep}}$. This might be because $L_1$ students result in models with unembeddings which lead to lower singular values for the $\mathbf{L}$ matrices on this dataset.

| | | Acc(Y)($\uparrow$) | $d_{\text{KL}}(\downarrow)$ | $d_{\text{logit}}(\downarrow)$ | $d_{\text{rep}}(\downarrow)$ | $m_{\text{CCA}}(\uparrow)$ |
|---|---|---|---|---|---|---|
| Synth | teach | $0.999 \pm 0.001$ | $-$ | $-$ | $-$ | $-$ |
| | KL | $0.999 \pm 0.001$ | $(4.57 \pm 0.43) \cdot 10^{-2}$ | $(7.70 \pm 0.27) \cdot 10^1$ | $(6.07 \pm 0.31) \cdot 10^0$ | $0.580 \pm 0.048$ |
| | $L_1$ | $0.999 \pm 0.001$ | $(5.95 \pm 1.40) \cdot 10^{-3}$ | $\mathbf{(2.85 \pm 0.09) \cdot 10^0}$ | $\mathbf{(1.69 \pm 0.04) \cdot 10^{-1}}$ | $\mathbf{0.999 \pm 0.001}$ |
| | $L_2$ | $0.999 \pm 0.001$ | $\mathbf{(1.72 \pm 0.14) \cdot 10^{-3}}$ | $(3.24 \pm 0.08) \cdot 10^0$ | $(2.04 \pm 0.05) \cdot 10^{-1}$ | $\mathbf{0.999 \pm 0.001}$ |
| CIFAR | teach | $0.540 \pm 0.010$ | $-$ | $-$ | $-$ | $-$ |
| | KL | $0.480 \pm 0.001$ | $\mathbf{1.29 \pm 0.004}$ | $(23.5 \pm 0.02)$ | $(8.76 \pm 1.37) \cdot 10^1$ | $0.515 \pm 0.001$ |
| | $L_1$ | $0.492 \pm 0.001$ | $1.33 \pm 0.003$ | $\mathbf{17.0 \pm 0.02}$ | $(4.56 \pm 0.60) \cdot 10^0$ | $\mathbf{0.767 \pm 0.001}$ |
| | $L_2$ | $\mathbf{0.493 \pm 0.001}$ | $1.32 \pm 0.005$ | $17.1 \pm 0.02$ | $\mathbf{(6.58 \pm 1.71) \cdot 10^{-1}}$ | $0.763 \pm 0.001$ |

*Table 4.* **Full results on `SUB`**.

| | Acc(Y)($\uparrow$) | Acc(C)($\uparrow$) | $d_{\text{KL}}(\downarrow)$ | $d_{\text{logit}}(\downarrow)$ | $d_{\text{rep}}(\downarrow)$ | $m_{\text{CCA}}(\uparrow)$ |
|---|---|---|---|---|---|---|
| teach | $0.915 \pm 0.010$ | $0.923 \pm 0.011$ | $-$ | $-$ | $-$ | $-$ |
| KL | $\mathbf{0.929 \pm 0.001}$ | $0.055 \pm 0.001$ | $0.645 \pm 0.007$ | $(3.73 \pm 0.05) \cdot 10^2$ | $(3.10 \pm 0.17) \cdot 10^3$ | $0.417 \pm 0.001$ |
| $L_1$ | $0.922 \pm 0.001$ | $\mathbf{0.747 \pm 0.009}$ | $\mathbf{0.151 \pm 0.002}$ | $(3.40 \pm 0.01) \cdot 10^1$ | $(1.00 \pm 0.09) \cdot 10^1$ | $0.993 \pm 0.001$ |
| $L2$ | $0.919 \pm 0.001$ | $0.724 \pm 0.006$ | $0.173 \pm 0.003$ | $\mathbf{(3.20 \pm 0.02) \cdot 10^1}$ | $\mathbf{(1.58 \pm 0.19) \cdot 10^0}$ | $\mathbf{0.994 \pm 0.001}$ |

