# OpenReview forum: "Logit Distance Bounds Representational Similarity"
_ICML.cc/2026/Conference — ICML 2026 regular_

### Official Review · Reviewer_hB1K · 2026-02-16

**Soundness:** 3
**Presentation:** 3
**Significance:** 2
**Originality:** 2
**Overall Recommendation:** 5
**Confidence:** 4

**Summary:**

Given two models, their final layer representations are known to be equivalent up to a linear transformation if their KL is zero. This work studies what happens when the KL, or other measures of functional similarity, are nonzero but low. They find that the distance between logits and a new metric they define is a better measure of similarity, with implications for knowledge distillation and interpretability.

**Compliance With Llm Reviewing Policy:**

Affirmed.

**Final Justification:**

The authors have addressed my main concerns. The paper is well executed and decently interesting, so it deserves to be shared with the community. The rebuttal clarified some confusions I had, especially regarding applications and whether they are applicable to language models, but also that there were no nontrivial insights/applications I missed. I recommend accepting this work.

**Key Questions For Authors:**

Questions are ordered from most to least significant. Unless there are surprising improvements I do not expect to change my score, so I recommend the authors focus on more pessimistic reviewers. If all given questions are answered, and new surprisingly interesting results/implications are shared, I will raise my score.
1. Do these results hold for language models? If so, do they have any nontrivial implications or insights?
2. Are there any nontrivial insights or implications stemming from the current results?
3. Are there reasons to believe that representations are in any way interesting beyond their utility for downstream performance? Note that interpretability tasks also constitute a task with some downstream performance.

**Limitations:**

Limitations are discussed but narrowly, pertaining to this work’s scope, not to its relation with the wider field.

**Strengths And Weaknesses:**

# Strengths
## Soundness
* The paper seems very rigorous, with definitions and assumptions well presented, well laid out proofs, good background that holds the readers hand, and experiments testing some of their claims.

## Presentation
* The paper has a nice flow, good explanations, and good visuals.
* More generally, I appreciate how the paper is very honest about which bounds are meaningful, which are more illustrative, etc.

## Significance
* This paper gives an interesting analysis of approximate representational similarity, a timely topic in the community.

## Originality
* The paper clearly extends existing work and has some interesting connections to certain methods, like knowledge distillation.

# Weaknesses
## Soundness
* Although this is clear from the paper’s details, it would be good to explicitly mention that the equivalent representations are only those of the last layer, or post-embedding representations, depending on the setup. The paper, justifiably, does not comment on deeper model representations, which are much harder to theoretically reason about.
* It is arguable whether it is unclear what “good” representations are (start of intro) - many would argue that good representations are those that yield good downstream performance. See _Representation Learning: A Review and New Perspectives, Bengio et al., 2013_, top right of page 1.
* Following this, it is not deeply discussed why having convergence in representations and not in functions is interesting/relevant. I believe there are many answers for this, the simplest being that the community is currently interested in this topic, although even that is still a partial answer.
* (more minor) An experiment(s), even a very simple artificial one, testing the theory would be valuable. Proofs speak in and of themselves but, as they can contain minute wrong details, supporting empirical evidence is valuable. The evidence in the final experiments is roundabout, as many parts of the theory - e.g. the bounds - are not shown.
* Isn’t d_rep very expensive to compute in practice? Due to the combinatorial number of subsets. It’s not used directly in the paper so this is, in this instance at least, fine, but this should still be mentioned.
* In Theorem 3.9 is L-tilde positive definite, do we know for sure that sigma_min is nonzero?
* (minor) In the analysis succeeding equation 15 I would appreciate a mention of how mu_m is also relevant, as while making d_logit^2 small one could inadvertently change mu_m as well, so this maximization is tricky.
* (minor) Was the KD temperature tuned? This isn’t mentioned.

## Presentation
* The presentation is generally solid, with all comments in this section being minor. Some comments in the other weakness sections also pertain to presentation, specifically missing discussions/comments.
* It’s a bit jarring referring to Figure 1 again on the last page, but alas space is tight.
* The shifted unembedding notation (line 144) is ambiguous.
* I don’t understand where’s the root in equation 8 (line 177).
* Equation 9 has a missing tag on one of the “f”s.
* Lines 238-244, convoluted sentence.
* Section 3.3 could have a better motivation, I liked the one in the intro more.

## Significance
* The paper’s biggest drawback is its limited setting. Some of the connections to other methods are certainly interesting, as is the theory, but the connection with common wisdom (where/if appropriate), current curiosities in the community, etc. are missing. Essentially, especially from reading the intro and conclusion, the paper doesn’t give a good answer to “okay this is interesting, but so what?”. I believe at least a somewhat better answer might simply stem from a better presentation, although I am unsure if a tremendously better one would unless there’s a link I missed.
* The settings demonstrated are all semi-artificial and small scale. A more realistic setting, using even a small language model, would be interesting. Language models should fulfill the theory’s desiderata out of the box, at least with their vocabulary almost always being much larger than their hidden dim.

## Originality
* Following the previous point, the paper is original but I’m unsure if its novelty is significant - how would it affect follow up work, are any of the insights deeply surprising, etc.

# Miscellaneous comments/suggestions
* In case there is a lack of space, some theorems can be informally stated.
* The condition in lines 187-188 is reasonable, I recommend mentioning this.

---

> ### Author Rebuttal · Authors · 2026-03-30
>
> We thank the reviewer for their thorough feedback and appreciate that they found our work rigorous.
>
> Because of the character limit, there are some minor comments we do not answer individually. As a general answer to the presentation comments, we will use this feedback to polish the text.
>
>
> > many would argue that good representations are those that yield good downstream performance
>
> We agree that downstream performance is an important aspect of representations, and this motivates us to consider the identifiability class (i.e., equivalent models with equal downstream performance). Other works advance that “good” representations should both generalize and capture interpretable structures [Koh et al. 2020], or invariant features [Arjovsky et al. 2019]. We interpret this as there being no agreed-upon definition of what makes representations “good” and prefer an agnostic position.
>
> Our work consolidates the relation between measures of closeness in model distributions and similarities of representations. If “good” representations are only expected to yield accurate predictions, KL is a valid measure of closeness. If “good” representations are meant to carry other properties and the identifiability class of the teacher matters, distillation with KL is not sufficient for this purpose.
>
>
> > why having convergence in representations and not in functions is interesting
>
> The convergence in representations we are most interested in implies a convergence in functions. For our definition of similar (close to being in the same equivalence class), similar representations give you similar distributions. We will emphasize this more in the text.
>
>
> > explicitly mention that the equivalent representations are only those of the last layer
>
> Yes, we will do this.
>
> > experiment(s)
>
> Please see the reply to the first question of reviewer **i5vc** for two new plots, where one shows the variation of representation dimensionality of students in SUB and the other illustrates the bound in Theorem F.1.
>
> > Isn’t d_rep very expensive to compute in practice?
>
> Yes, it can be expensive to compute exactly. In future work, we hope to find a way to approximate this better.
>
> > do we know for sure that sigma_min is nonzero?
>
> Since the $\tilde{\mathbf{L}}\_{\mathcal{J}}$ matrices are invertible, the minimal singular value $\sigma_{min}$ is positive. We will clarify this in the text.
>
> > I don’t understand where’s the root in equation 8 (line 177)
>
> In equation 8, we define the squared logit distance, $d_{logit}^2$, so taking the square root gives $d_{logit}$. We will make this more clear.
>
>
> > connection with common wisdom (where/if appropriate), current curiosities in the community, etc. are missing.
>
> Please see the reply to the first question of reviewer **PSzv**, for how to connect our work with other works on representational similarity.
>
> >  settings demonstrated are all semi-artificial and small scale
>
> Testing distillation setups on larger models, while relevant, is out of our reach at the moment. We look forward to overcoming this limitation in future work.
>
> > 1. Do theoretical results hold for language models?
>
> All the stated assumptions are reasonable for language models, see also [Roeder et al. 2021] and [Nielsen et al. 2025].
>
>
> > 2. any nontrivial insights or implications stemming from the current results? 3. reasons to believe that representations are in any way interesting beyond their utility for downstream performance?
>
> In section 4 we present applications of our results.
>
> 1. Knowledge distillation: Obtaining similar representations to a teacher model. KL-based losses are commonly employed in distillation, however KL does not guarantee similar representations. This can matter if students are meant not only to preserve classification accuracy of the teacher, but also capture similar rank orderings [Grivas et al. 2024] or uncertainty in the predictions, see eg., [Wang 2023].
>
> 2. Transferring interpretable representations. This can be useful in relation to mechanistic interpretability practice [Bereska and Gavves 2024]. For linear properties [Park et al., 2024], such as linear concepts we describe in Section 4.2, distillation with KL can fail to the purpose of transferring teacher’s linear properties to the student. This can be relevant for (distilled) models where it is expected to have an interpretable behavior in relation to concepts, see e.g. [Koh et al. 2020], or for linear steering the model in generation [Wu et al. 2025].
>
> We will highlight both points in the text.
>
>
>
> Arjovski et al. 2019, Invariant Risk Minimization
>
> Grivas et al. 2024, Taming the Sigmoid Bottleneck: Provably Argmaxable Sparse Multi-Label Classification
>
> Wang 2023 Calibration in Deep Learning: A Survey of the State-of-the-Art
>
> Bereska and Gavves 2024, Mechanistic Interpretability for AI Safety -- A Review
>
> Koh et al. 2020, Concept Bottleneck Models
>
> Wu et al. 2025, AxBench: Steering LLMs? Even Simple Baselines Outperform Sparse Autoencoders

---

> > ### Author Rebuttal · Reviewer_hB1K · 2026-04-02
> >
> > Thanks for the thorough response. My concerns have been addressed and I will raise my score. I will not raise it further as I still believe the implications are limited, as the answers to Q2+Q3 clarify connections to already mentioned applications but do not present new ones. Still, the paper is interesting and well executed, so it deserves to be shared with the wider community.

---

> > > ### Author Response · Authors · 2026-04-08
> > >
> > > Thank you for your reply and for raising your score.
> > >
> > > We also appreciate your suggestion to further explore the implications and applications of the theory, which we agree is a promising direction for future work.

---

### Official Review · Reviewer_i5vC · 2026-03-11

**Soundness:** 4
**Presentation:** 4
**Significance:** 4
**Originality:** 3
**Overall Recommendation:** 5
**Confidence:** 4

**Summary:**

The paper investigates the "approximate identifiability" of discriminative models (e.g. autoregressive LLMs, classifiers). It builds on recent findings that models can have nearly identical ouput distributions (low KL divergence) yet possess fundamentally different internal linear structures. The authors propose a "logit distance" metric and prove that closeness in this metric guarantees linear representational similarity under strong conditions. They introduce a new dissimilarity measure, d_{rep}, and demonstrate through distillation experiments on synthetic and image data that matching logits preserves a teacher's final representation (right before the linear prediction head) up to a linear map.

**Compliance With Llm Reviewing Policy:**

Affirmed.

**Final Justification:**

See my rebuttal acknowledgment

**Key Questions For Authors:**

* Standard distillation often uses a combination of KL and other loses. If one were to use a symmetric or "hybrid" KL, would the poor scaling in \tau identified in Eqn (14) persist?
* Is it possible to include a discussion on the tightness of the bounds derived? Specifically in Theorem 3.3.
* In cases where the student is in a different model class and the logit distance cannot reach zero, is there a risk that the logit metric forces student to "over-fit" to high-energy logit noise, whereas KL would naturally allow the student to ignore the "problematic" low-probability regions?

**Limitations:**

* The result are restricted to distillation between models of the same family
* There is no discussion on the tightness of the bounds
* The motivation mentions autoregressive LLM models, but no empirical study was conducted for this problem.
* There are no empirical experiments to suggest the theory extends to classic distillation settings between a large teacher model and small student model.

**Strengths And Weaknesses:**

Strengths:

* The scaling analysis of \tau in Theorem 3.3 is a highlight. By showing the KL-to-logit bound scales as \tau^{-1}, the authors provide a clear mathematical explanation for why KL-based distillation can fail to preserve linear structure in practice, where probabilities for "tail" tokens/classes are often infinitesimal. To further solidy this relationship, the authors should specify if the bound is tight, or explore a tight bound to confirm this failure in the KL divergence.
* The relationship between distributional distance and logit distance is intuitive, and the metric easy and efficient to compute during training.
* The relationship between each metric (especially with respect to the KL divergence) is robustly identified and proved.
* The implication of the findings for distillation is clear.

Weaknesses:
* A significant limitation is the requirement that models belong to the same family. In real-world distillation, we often move from a very large teacher model to a smaller student model. The paper lacks discussion or empirical test on how these bounds degrade when the student's architecture lacks the capacity to even represent the teacher's linear equivalence class.
* The assumption that probabilities are uniformly bounded away from 0 is hard to satisfy. In LLMs with a vocabulary size of 50k+ tokens, the vast majority of tokens have near-zero probability. While the authors use this to explain KL's failure, the reliance on \tau to make the logit metric "proper" may limit the theorem's reach in sparse-label regimes.

---

> ### Author Rebuttal · Authors · 2026-03-30
>
> We thank the reviewer for their positive assessment on our work and for calling our mathematical explanation clear. Below we address the issues raised by their review.
>
> > A significant limitation is the requirement that models belong to the same family.
>
> It is true that our theory requires models to belong to the same model class (eq. 1), a restriction common to works in identifiability, however, our theory is flexible enough to treat models which can have a different number of layers or leverage different neural architecture for their (un)embedding functions. To illustrate how student models with smaller dimension than the teacher behave, we have run additional experiments (5 students on one teacher) on SUB and made a figure ([``link here``](https://figshare.com/s/07512cd2ffb3677c3b02)) showing three panes:
>
> 1. Accuracy on the main task increases for all students with dimension of representations.
>
> 2. Accuracy on the concepts increases with dimension for the logit students, but not for the KL student.
>
> 3. Mean canonical correlation is in general high for the logit students and low for the KL student.
>
> This means the KL student is often able to solve the main task well, but it does not preserve the representational structure of the teacher.
>
> We also have an initial theoretical result in Theorem F.1. bounding canonical correlations, which allows the representational dimension of one model to be smaller than the other. To illustrate this bound, we have made a figure [(``link here``)](https://figshare.com/s/40e20eb96fdc36e09b05) , where we see for five students (dim 25) on a CIFAR100 teacher model (dim 50) how the values of the bound evolve over training epochs. We see that for all students at all measured epochs the upper bound holds.
>
> > The assumption that probabilities are uniformly bounded away from 0
>
> We remark that the logit distance does not strictly require $\tau$-lower bounded distributions and is valid for all models in our family (they always have non-zero probabilities). We only claim that $\tau$-lower bounded probabilities constitute a sufficient condition to avoid singularities with $d\_{logit}$.
>
> > If one were to use a symmetric or "hybrid" KL, would the poor scaling in \tau identified in Eqn (14) persist?
>
> Good question. Symmetry of $d\_{logit}$ implies that the bound in Theorem 3.3 holds for the smaller of forward and backward KL-divergence. The poor scaling in $\tau$ would persist and this cannot be avoided (see next question). We will include this in the appendix.
>
> > Is it possible to include a discussion on the tightness of the bounds derived?
>
> The bound in Theorem 3.3 is tight up to the logarithmic term in $\tau$. We discuss this in Remark E.3 and we will add a short discussion to the main text. The example there also shows that the same scaling in $\tau$ is necessary for the symmetric KL divergence (see previous question). In Remark E.5 we also show that relaxing the lower bound on one of the models results in even worse scaling in $\tau$. Tightness of the bound in Theorem 3.4 depends on the eigenvalue structure of the logit covariance matrix. However, in Theorem F.1 we state a more general bound which controls weighted averages of the canonical correlations by the (squared) logit distance. This bound is tight.
>
> > In cases where the student is in a different model class and the logit distance cannot reach zero, is there a risk that the logit metric forces student to "over-fit"?
>
> Good question. This relates to an interesting trade-off between capturing the structure of teacher representations and ignoring information about unlikely labels which might be noisy. If we ignore the unlikely labels, as KL does, then we are not necessarily capturing the structure of teacher representations. On the other hand, if the structure of the teacher representations is important, e.g. we know the teacher has interpretable concepts we want to preserve, then we must also consider the unlikely labels, as the logit distance does. In the case where the structure of unlikely labels of the teacher is not meaningful, the logit distance may steer students to learn a “noisy” behavior. We will add this discussion to the limitations section.
>
> For the case when the student dimension m’ is smaller than the teacher dimension m, we refer again to theorem F.1 as an initial exploration of this setting, which we hope to expand upon in future work. Moreover, the results in varying representational dimensionality on SUB, show high mCCA, highlighting that L1 and L2 students learn representations that are linear invertible transformations of a subspace of teacher representations.

---

> > ### Author Rebuttal · Reviewer_i5vC · 2026-04-02
> >
> > I appreciate the authors for addressing my questions and concerns and I will raise my score. However, I maintain that the paper would be stronger if it would add more solid empirical experiments showing the logit based loss leads to a meaningful gain in distillation performance on more large-scale real world data and models.

---

> > > ### Author Response · Authors · 2026-04-08
> > >
> > > Thank you for replying to our rebuttal and for raising your score.
> > >
> > > In this work we focused on the theoretical aspects of how KL divergence and logit distance might connect to representations, but a more thorough empirical investigation will be interesting future work.

---

### Official Review · Reviewer_PSzv · 2026-03-13

**Soundness:** 3
**Presentation:** 2
**Significance:** 2
**Originality:** 3
**Overall Recommendation:** 4
**Confidence:** 3

**Summary:**

This paper studies representational similarity under invariance to invertible linear transformations, particularly in the context of the relationship between the similarity of output distributions and the representations. They ask if networks with similar in output distributions are necessarily close together in terms of representational similarity (as measured by a metric invariant to invertible linear transformations).  This is not generally true and depends on the metric, so the authors introduce the logit distance and show that under some conditions, closeness in this distance implies closeness in mean CCA and a metric they introduce to measure how linearly identifiable the two representations are.  They apply this idea to knowledge distillation, and argue that replacing the KL divergence term to align the teacher and student distributions with the logit distance produces more linearly similar representations and student representations that retain the teacher's ability to linearly classify human-annotated concepts from the embeddings.

**Compliance With Llm Reviewing Policy:**

Affirmed.

**Final Justification:**

The author response was compelling and addressed my concerns, so I am tipped over to recommend acceptance (despite there being some limitations for how broadly this framework applies to ML tasks).

**Key Questions For Authors:**

- The introduction of the matrix Ltilde, constructed from a pivot label and a subset of m labels such that Ltilde is invertible, doesn't seem sufficiently explained.  Throughout I was often confused about how the pivot and subset of m labels are chosen.  Is the choice arbitrary, or does it require specific properties?  I assume m needs to equal the embedding dimension?  It seems to be written that way but not discussed.  In the linear identifiability theorem, you state that A “can be” set to Atilde. Is this true for any pivot and subset choice?  At present, the role of Ltilde and Atilde is under-motivated and under-explained, especially since it appears again in the definition of the linear identifiability dissimilarity they define.

**Limitations:**

Yes

**Strengths And Weaknesses:**

The core idea of this paper is interesting and their theoretical bounds lead to a very clear actionable result in knowledge distillation. However, the presentation makes the contributions difficult to fully assess. The mathematical exposition lacks clarity in key places, several definitions are introduced in an opaque way, and the broader context within representational similarity research is insufficiently discussed.

Below is a summary of my main concerns and weaknesses:

* There is a broad literature on representation similarity in the machine learning community (RSA, CKA, Procrustes alignment, etc.), and this paper didn't adequately connect/contrast this work with existing similarity metrics and more commonly used invariances (e.g., orthogonal invariance in CKA).  However, I think that the connection between the logit distance and the linear identifiability dissimilarity between the representations and the application to knowledge distillation is compelling and absent in that literature.

* A  foundational concept of this paper is invariance to invertible linear transformations. However, as noted in [Kornblith et al. (2019)](https://proceedings.mlr.press/v97/kornblith19a.html), any similarity index invariant to arbitrary invertible linear transformations yields degenerate behavior when representation width exceeds dataset size: any two representations of sufficient width can be mapped into one another via an invertible linear transform.  The paper does not clearly address this limitation; it only considers the case where the number of labels exceeds the dimensionality of the representation. This would seem to preclude many cases of interest (particularly wide hidden layers) and is limited to rather narrow circumstances (e.g. low-dimensional bottleneck layers, or LLMs in certain circumstances).

* The empirical results are rather thin (~1 page) and limited (toy synthetic dataset, CIFAR-100 with small architecture, etc.). Particularly for the knowledge distillation results, it would be helpful to investigate larger scale models and datasets.

* The results don't say much about representational similarity between two networks that are trained on different task objectives. They also don't say much about models trained on self-supervised objectives since (unless I'm misunderstanding something) the logit distance isn't so neatly defined in the self-supervised setup.

---

> ### Author Rebuttal · Authors · 2026-03-30
>
> We thank the reviewer for their valuable feedback and for finding the idea interesting.
> Below we address the issues raised by their review.
>
> > …connect/contrast this work with existing similarity metrics…
>
> In our work, we take an identifiability perspective, under which invariance to linear transformations is the most natural choice [Roeder et al., 2021]. Our point is not that this is the right similarity measure in general, but that it is a relevant one in our setting, one for which we can prove bounds, and one that is directly connected to the preservation of linear properties such as those discussed in Sec. 4.
>
> CKA [Kornblith et al., 2019] and orthogonal Procrustes alignment [Schönemann, 1966], by contrast, are based on invariance up to orthogonal transformations, motivated by considerations different from identifiability, e.g. invariance of gradient-descent dynamics [LeCun et al., 1991]. For RSA [Kriegeskorte et al., 2008], the precise invariances depend on the choice of similarity functions. These are reasonable choices in other settings, and we will add discussion of them in Appendix A, where we already mention CKA. However, they are less directly tied to the identifiability perspective that motivates our choice here. Whether theory analogous to ours can, under suitable assumptions, be established for such similarity measures is an open question.
>
> > …when representation width exceeds dataset size: any two representations of sufficient width can be mapped into one another via an invertible linear transform.
>
> It is true that it is difficult to verify linear identifiability when the number of points is less than the number of dimensions in the representations. In our theory we work with expectations taken over a potentially infinite number of samples. In our experiments, we make sure that the number of points we use to test similarity metrics is large enough to avoid this pathological behavior. So in both cases we avoid this problem.
>
> >This would seem to preclude many cases of interest (particularly wide hidden layers)
>
> We would like to clarify that the models we study can have any number of hidden layers of any dimension in the embedding function $\mathbf{f}$, we only need the final dimension (the one before the final dot product and softmax) to be smaller than the number of tokens. This is satisfied for any language model which has more tokens (usually order of 100k) than representational dimension (usually around 1k neurons). So the assumptions are fair for most modern language models e.g. GPT-2, LLaMa-3, Mistral 7B. We will clarify this in the article.
>
> > The empirical results are rather thin
>
> We agree that a broader experimental evaluation would be valuable. That said, our empirical results are designed to directly validate the distinction between using KL and $d\_{logit}$ based losses in different settings. In all our settings, we consistently observe that the proposed metric aligns well with the predicted representational similarity, whereas standard KL divergence does not exhibit the same behavior. For some additional empirical analysis see the reply to the first question of reviewer **i5vc**.
> While extending these experiments to larger-scale distillation setups is an important direction, it is currently beyond our computational scope. We view our present results as a focused but reliable illustration of the theory, and we look forward to expanding the empirical analysis in future work.
>
>
> > The results don't say much about representational similarity between two networks that are trained on different task objectives. They also don't say much about models trained on self-supervised objectives.
>
> We agree that the current theory does not cover the case where models are trained on different data (due to different modalities or input distributions). We also remark that no other works can sensibly reach that point under the identifiability lens.
> Nonetheless, the model class (eq. (1)) we consider is broad, see [Roeder et al. 2021]. For example, the model family captures self-supervised pre-training approaches such as contrastive predictive coding [Oord et al., 2018; Hénaff et al., 2019]. This implies that, if our theoretical conditions are met, the same results hold for that class of models as well. We will clarify this in the appendix.
>
> > The introduction of the matrix Ltilde
>
> The matrix $ \tilde A$ is constructed by considering the matrices $L$ and $L’$ of the two models for $m$ (equal to representational dimensionality) choices of labels and of a pivot point $\tilde y$.
> The exact choice of pivot and labels does not matter because:
>
> 1. Theorem 2.2 does not depend on this choice, which we show in proposition C.1 and mention below the theorem.
>
> 2. When conditional distributions do not coincide, we average over all possible choices of $m$ labels and pivot points (Def 3.7) and all choices lead to invertible L matrices because of assumption 3.5.
>
> We will be clearer about this in Section 3.3.

---

> > ### Author Rebuttal · Reviewer_PSzv · 2026-04-01
> >
> > Thanks for the responses. I'm happy to raise my score. I still think there are some limitations &mdash; e.g. does not apply to cases where the number of output labels is less than the final layer dimension (which is common in image classification models), so I would be hesitant to raise it further unless there is strong reason.

---

> > > ### Author Response · Authors · 2026-04-08
> > >
> > > Thank you for reading our rebuttal and for raising your score.
> > >
> > > We agree that the case with fewer labels than dimensions is important. We focused on the regime with more labels since it already covers many cases of interest, including autoregressive LLMs. Based on our results and those by Marconato et al. (2025) that relax the diversity condition, we expect that KL-based distillation may still fail to preserve similarity in the case of fewer labels, especially when teacher logits lie in a low-dimensional subspace. Vice versa, we expect d_logit distillation will give similar guarantees, as some of our results do not require considering the same dimensionality of representations for teacher and students (see Theorem F.1).
> > >  Working out the precise details is an important direction for future work.

---

### Decision · Program_Chairs · 2026-04-30

**Decision:**

Accept (regular)

**Comment:**

This paper studies representational similarity in discriminative models under invariance to linear transformations, and its relationship to differences in output distributions and introduces a logit-based distance that guarantees similarity of representations when output distributions are close. Reviewers agree that the paper is technically solid and provides a clear and original theoretical contribution, with relevant implications for knowledge distillation.

The main strengths are in the rigor of the analysis and the insight that KL divergence may fail to preserve representational structure, while the proposed logit distance provides meaningful guarantees. The paper has some limitations in scope and clarity, as the results rely on specific assumptions and focus on final-layer representations. and the connection to prior work on representational similarity could be better explained. The rebuttal addressed the main concerns, clarifying assumptions and technical details, and reviewers acknowledged these clarifications and maintained or improved their positive evaluations, as confirmed in the internal discussion.

Overall, the paper is technically sound and makes a useful contribution to the ICML community, both from a theoretical perspective and for its potential practical implications (e.g., in distillation in LLMs). Therefore I recommend acceptance.

In the revision, the authors should clarify the assumptions and scope (especially dimensionality and focus on final-layer representations) and improve the discussion of related work and limitations.